# RL4SBDD: Reinforcement Learning for Preference Alignment in Structure-based Drug Design

## Abstract

Generative models have been proven effective in designing novel ligands conditioned on protein target structures, which is a fundamental task in drug discovery. Recent research has explored aligning generative models with practical requirements for certain molecular properties through gradient guidance or direct preference optimization. In this paper, we introduce RL4SBDD, a reinforcement learning framework that unifies existing alignment methods and analyzes their limitations, including biased value estimation, inefficient data utilization, and off-policy distribution shift. Building on our framework, we propose RL4SBDD-M, a novel approach to address these problems and facilitate preference alignment for arbitrary verifiable rewards in structure-based drug design. We introduce a policy model conditioned on rewards to avoid value estimations and fully leverage training data, and perform iterative reinforcement learning to mitigate off-policy distribution shift. Moreover, we introduce a sampling process with mixed guidance and noise reduction to improve generation quality and efficiency. By jointly optimizing binding affinity and synthesizability, RL4SBDD-M achieves a state-of-the-art success rate of 58.1% on the CrossDocked2020 benchmark, which is further boosted to 93.0% within 50 generations with oracle evaluations. Experiment analysis on a broader combination of rewards further confirms its efficacy in balancing multiple objectives, underscoring its potential for practical drug design.

## 1 Introduction

Structure-based drug design (SBDD) (Anderson, 2003) is a pivotal task in pharmaceutical discovery, aiming to create molecules that selectively bind with a protein pocket. Recent advances in machine learning leverage generative models ranging from auto-regressive models (Luo et al., 2021; Peng et al., 2022), diffusion models (Guan et al., 2023a;b), flow matching (Zhang et al., 2024), and Bayesian flow networks (Qu et al., 2024) to solve SBDD as a conditional generation problem based on 3D pocket structures. However, generative models essentially mimic the distribution of training molecules, thereby inheriting their limitations such as moderate binding affinities, suboptimal synthesizability, unsatisfactory selectivity, and suboptimal ADME properties (Zhou et al., 2024; Cheng et al., 2024; Gao et al., 2024). This misalignment with the demands of drug design largely compromises the application of generative models in real-world SBDD practices.

Recently, several pioneering works have introduced preference alignment techniques to address this problem within generative models. For example, Dorna et al. (2024); Qiu et al. (2025) train regression models to predict alignment scores for noisy molecules and utilize the gradients of these models to guide sampling. Gu et al. (2024); Cheng et al. (2024); Huang & Zhang (2025) apply direct preference optimization (DPO) (Rafailov et al., 2023) by annotating winning and losing molecules and enhancing the likelihood to generate winners while suppressing losers.

Motivated by the success of reinforcement learning with verifiable rewards (RLVR) (Yu et al., 2025; Mroueh, 2025), we introduce **RL4SBDD**, a **R**einforcement **L**earning framework illustrated in Fig. 1(a) that unifies existing preference alignment methods in **SBDD**. Specifically, we interpret alignment goals as verifiable rewards and the iterative sampling procedure of generative models as a Markov decision process (MDP), and show that gradient guidance (Dorna et al., 2024; Qiu et al.,

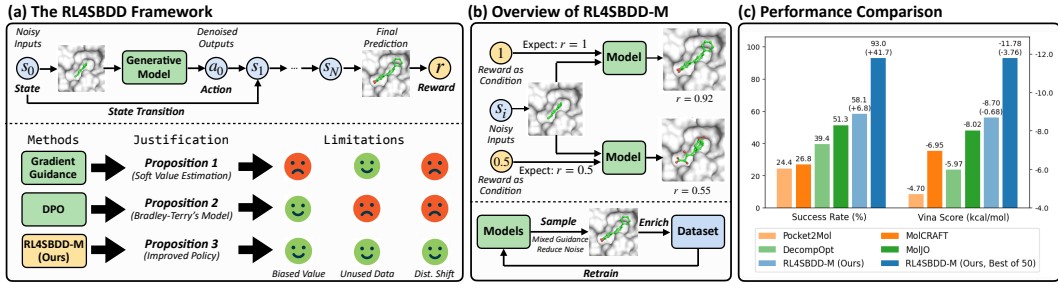

Figure 1: **Overview. (a) The RL4SBDD framework.** We formulate the alignment goals as verifiable rewards and the sampling process as an MDP, unify existing alignment methods, and analyze their limitations. **(b) An overview of the proposed model RL4SBDD-M.** We introduce a reward-conditioned policy model, iterative RL, and mixed guidance with reduced noise. **(c) Performance comparison on CrossDocked2020.** Our model achieves the best success rate of 58.1% within a single generation and 93.0% within 50 generations by jointly optimizing binding affinity and synthesizability.

2025) and DPO (Gu et al., 2024; Cheng et al., 2024; Huang & Zhang, 2025) can be viewed as alternatives for optimizing the KL-constrained reward objective in soft RL (Haarnoja et al., 2017). This unified perspective not only clarifies their conceptual foundations but also reveals three key limitations. (1) **Biased Value**: Gradient guidance depends on soft value estimation (Schulman et al., 2017a), which is unreliable in the early stages of generation when the inputs are noisy. (2) **Unused Data**: Current DPO implementations in SBDD select only the best and worst among multiple molecules for a pocket as a preference sample to avoid likelihood displacement (Razin et al., 2025; Chen et al., 2025), which breaks the pairwise comparative structure of Bradley-Terry's model (Bradley & Terry, 1952) and leaves the majority of data unexploited. (3) **Distribution Shift**: Both gradient-based and DPO-based models are trained on the original dataset but sampled from a distribution reshaped by alignment objectives, leading to an off-policy distribution shift (Kumar et al., 2019) that can degrade sampling quality.

Building upon our framework, we propose RL4SBDD-M, a novel approach shown in Fig. 1(b) that addresses these limitations by incorporating insights from recent RL advances (Uehara et al., 2025; Frans et al., 2025). Specifically, we introduce a reward-conditioned policy model with theoretical grounding from the RL perspective. The model (1) **Bypasses Biased Value Estimations** by directly proposing advantageous actions, *i.e.*, high-reward molecules during early sampling, and (2) **Leverages Supervision from All Training Data** by learning to generate molecules *w.r.t.* the desired rewards. Moreover, we adopt an iterative RL paradigm to (3) **Mitigate Off-Policy Distribution Shift** by continuously augmenting the training set with generated samples, which bridges training with the evolving sampling distribution. Finally, we design a sampling strategy that integrates mixed guidance and reduced noise, improving both the quality and efficiency of molecule generation.

As shown in Fig. 1(c), using binding affinity and synthesizability as optimization objectives, RL4SBDD-M achieves a median Vina Score of -8.70 kcal/mol and a success rate of 58.1% on the CrossDocked2020 dataset (Francoeur et al., 2020), outperforming MolJO (Qiu et al., 2025) by 6.8% absolute gains in success rate while being 4.6 times faster in sampling. A test-time scaling (Zhang et al., 2025b) analysis further boosts the success rate to 93.0% by selecting the best molecule from 50 generations. We also validate that RL4SBDD-M generalizes across a broader range of combined reward objectives, underscoring its potential in assisting real-world drug design.

Our contributions are summarized as follows:

- We present RL4SBDD, a systematic RL framework that unifies and analyzes preference alignment methods in SBDD.

- We propose RL4SBDD-M, a novel approach to address the limitations of prior works with reward conditioning, iterative RL, and sampling with mixed guidance and reduced noise.

- We show that RL4SBDD-M achieves state-of-the-art performance on CrossDocked2020 with an average success rate of 58.1% for a single generation and 93.0% for the best of 50 generations by jointly optimizing binding affinity and synthesizability.

## 2 RELATED WORK

**Generative Models in Structure-based Drug Design** Most existing studies formulate SBDD as a conditional generation problem that models the distribution of ligands given a protein pocket. Early works adopt sequence models to generate 1D SMILES strings (Bjerrum & Threlfall, 2017) or 2D molecular graphs (Segler et al., 2018). Subsequent research advances to 3D geometry modeling to better capture the atomic interactions within protein-ligand complexes. For example, Ragoza et al. (2022) develops a variational autoencoder to model the atomic density grids of 3D molecules. Luo et al. (2021); Peng et al. (2022) employ auto-regressive models to sequentially choose and place atoms or molecular fragments within the pocket. More recent methods leverage denoising diffusion models (Guan et al., 2023a;b), flow matching (Zhang et al., 2024), or Bayesian flow networks (BFN) (Qu et al., 2024) that progressively construct atom coordinates and types. In this work, we adopt BFN (Graves et al., 2023) as our backbone due to its efficiency and flexibility in molecular design.

**Reinforcement Learning for Preference Alignment** Recently, reinforcement learning with human feedback (RLHF) (Ziegler et al., 2019; Ouyang et al., 2022) and verifiable rewards (RLVR) (Mroueh, 2025; Guo et al., 2025; Wen et al., 2025) has emerged as a popular paradigm for enhancing the instruction-following and reasoning capabilities of large language models (LLMs) by enforcing the models to learn from their own generations. Similar techniques have been successfully extended to other domains, including text-to-image generation (Xiao et al., 2024; Wallace et al., 2024), inverse folding (Ektefaie et al., 2025), and protein design (Stocco et al., 2024; Wang et al., 2025). In SBDD, several works explore preference alignment for generative models using gradient guidance (Dorna et al., 2024; Qiu et al., 2025) or direct preference optimization (Gu et al., 2024; Cheng et al., 2024; Huang & Zhang, 2025), or GFlowNet (Shen et al., 2024; 2025). To our knowledge, we are the first to unify these approaches under an RL framework and to provide a systematic analysis of their limitations.

## 3 THE RL4SBDD FRAMEWORK

### 3.1 PROBLEM FORMULATION

**Structure-Based Drug Design** In SBDD, generative models aim to produce a molecule $\mathbf{m} = \{(\mathbf{x}_{\mathcal{M}}^{(i)}, \mathbf{v}_{\mathcal{M}}^{(i)})\}_{i=1,2,\cdots,N_{\mathcal{M}}} \in \mathcal{M}$ consisting of $N_{\mathcal{M}}$ atoms conditioned on a protein pocket $\mathbf{p} = \{(\mathbf{x}_{\mathcal{P}}^{(i)}, \mathbf{v}_{\mathcal{P}}^{(i)})\}_{i=1,2,\cdots,N_{\mathcal{P}}} \in \mathcal{P}$ with $N_{\mathcal{P}}$ atoms. Here, $\mathbf{x}_{\mathcal{M}}^{(i)}, \mathbf{x}_{\mathcal{P}}^{(i)} \in \mathbb{R}^3$ represent the 3D coordinates of atoms within the molecule and pocket, and $\mathbf{v}_{\mathcal{M}}^{(i)} \in \mathbb{R}^{K_{\mathcal{M}}}, \mathbf{v}_{\mathcal{P}}^{(i)} \in \mathbb{R}^{K_{\mathcal{P}}}$ denote their corresponding atomic features. In the remainder of the paper we omit the condition $\mathbf{p}$ for notational brevity.

**Preference Alignment as KL-Constrained Reward Maximization** In real-world drug design, the ligand is expected to satisfy multiple objectives such as strong binding affinity with the pocket, high synthesizability, high selectivity, and favorable ADME properties (Hughes et al., 2011). If such alignment goals can be quantified by an oracle function $r : \mathcal{M} \to \mathbb{R}$, the preference alignment problem could be formulated as optimizing the following KL-constrained reward maximization objective (Haarnoja et al., 2017):

$$\pi^* = \underset{\pi:\mathcal{M}\to\Delta(\mathcal{M})}{\arg\max} \left( \underset{\mathbf{m}\sim\pi}{\mathbb{E}} [r(\mathbf{m})] - \alpha \cdot \mathcal{D}_{KL}(\pi||\pi_{\text{ref}}) \right), \quad (1)$$

where $\pi_{\text{ref}}$ denotes the output distribution of a reference model $\Phi_{\text{ref}}$ trained to approximate data distribution, $\Delta(\mathcal{M})$ is the probability simplex over the molecular space $\mathcal{M}$, and $\alpha$ is a hyperparameter.

Following (Haarnoja et al., 2017), we obtain the closed-form solution of Eq. 1 as:

$$\pi^*(\mathbf{m}) = \frac{\pi_{\text{ref}}(\mathbf{m}) \exp\left(r(\mathbf{m})/\alpha\right)}{\mathbb{E}_{\mathbf{m}\sim\pi_{\text{ref}}}[\exp\left(r(\mathbf{m})/\alpha\right)]}. \quad (2)$$

Since $\pi_{\text{ref}}$ is guaranteed SE(3)-invariant by the design of generative models and $r(\mathbf{m})$ is typically invariant under 3D roto-translations, it follows that $\pi^*$ also preserves SE(3)-invariance.

**Markov Decision Process within Generative Models** As illustrated in Fig. 1(a), most generative models for SBDD perform sampling by starting from a noisy prior and refining the molecule with

multiple denoising steps. By interpreting the noisy sample or parameters at time $t_i$ as the state $\mathbf{s}_i \in \mathcal{S}$, the model output $\mathbf{a}_i = \Phi(\mathbf{s}_i; t_i) \in \mathcal{A}$ as the action, and the denoising step as the state transition kernel $p_T$, the generative process could be formed as a Markov decision process $(\mathcal{S}, \mathcal{A}, p_T, r, \gamma)$ (Sutton et al., 1998), where the reward $r$ is 0 for all intermediate states and is non-zero only upon the generation of the final molecule, and the decay factor $\gamma = 1$. Specifically:

$$\pi(\mathbf{m}|\mathbf{s}_N) = p_O(\mathbf{m}|\mathbf{s}_N), \quad \pi(\mathbf{m}|\mathbf{s}_i) = \mathop{\mathbb{E}}_{\mathbf{s}_{i+1} \sim p_T}[\pi(\mathbf{m}|\mathbf{s}_{i+1})], \quad \pi(\mathbf{m}) = \mathop{\mathbb{E}}_{\mathbf{s}_0 \sim p_I}[\pi(\mathbf{m}|\mathbf{s}_0)],$$

$$p_T(\mathbf{s}_{i+1}|\mathbf{s}_i, \mathbf{a}_i) = \text{Combine}(\mu_i \mathbf{s}_i, \rho_i \mathbf{a}_i, \sigma_i \mathbf{w}), \quad \mathbf{w} \sim \mathcal{N}(0, \mathbf{I}) \tag{3}$$

where $N$ is the number of sampling steps, $p_O$ decodes the final state into a valid molecule, $p_I$ represents the prior distribution over initial states, $\text{Combine}(\cdot, \cdot, \cdot)$ is a combination operator, and $\mu_i, \rho_i, \sigma_i$ are hyperparameters derived from the noise schedule of the generative model.

## 3.2 Unification and Analysis for SBDD Preference Alignment Strategies

**Analysis for Gradient Guidance** Innovated by classifier guidance (Dhariwal & Nichol, 2021) in image diffusion models, Dorna et al. (2024); Qiu et al. (2025) define a modified transition kernel $p_T^V(\mathbf{s}_{i+1}|\mathbf{s}_i, \mathbf{a}_i) \propto p_T(\mathbf{s}_{i+1}|\mathbf{s}_i, \mathbf{a}_i) \exp(-E(\mathbf{s}_{i+1})/\lambda)$, where $E$ is a latent energy function with normalizing factor $\lambda$. The negative energy $-E$ is parameterized by a neural regression model $\Phi_V$ which learns to estimate the reward $r(\mathbf{m})$ from noisy states $\mathbf{s}_i$, as shown in Fig. 2(a). The gradient of $\Phi_V$ is then used to guide the transition via a first-order approximation of the energy term as follows:

$$p_T^V(\mathbf{s}_{i+1}|\mathbf{s}_i, \mathbf{a}_i) \stackrel{\text{apx}}{\propto} p_T(\mathbf{s}_{i+1}|\mathbf{s}_i, \mathbf{a}_i) \exp(-E(\mathbf{s}_i)/\lambda) \exp(\Delta \mathbf{s}_i \nabla_{\mathbf{s}_i} \Phi_V(\mathbf{s}_i; t_i)/\lambda), \tag{4}$$

where $\Delta \mathbf{s}_i$ is a constant indicating the difference between $\mathbf{s}_i$ and $\mathbf{s}_{i+1}$. In practice, the guidance is implemented via a specifically designed combination operation in Eq. 3 that yields $p_T^V$ exactly.

In light of Q-learning (Watkins & Dayan, 1992), we interpret the energy model $\Phi_V$ as learning from a one-pass Monte Carlo estimation of the soft value function under the reference policy:

$$\Phi_V(\mathbf{s}_i; t_i) \approx V_{\pi_{\text{ref}}}(\mathbf{s}_i) = \lambda \log \mathop{\mathbb{E}}_{\mathbf{m} \sim \pi_{\text{ref}}(\cdot|\mathbf{s}_i)}[\exp(r(\mathbf{m})/\lambda)]. \tag{5}$$

Using the isomorphism of values and policies in soft RL (Schulman et al., 2017a), we can derive the following proposition, with detailed proofs in Appendix B.1:

**Proposition 1.** *If $p_I$ is a Dirac distribution, then using the transition kernel $p_T^V$ in Eq. 4 to perform sampling in Eq. 3 is equivalent to sampling from the optimal policy $\pi^*$ in Eq. 2. Furthermore, the normalizing factor $\lambda$ coincides with the hyperparameter $\alpha$.*

*Remark 1.* For Bayesian Flow Networks adopted by Qiu et al. (2025) and our model, the initial state $\mathbf{s}_0$ is fixed and $p_I$ is indeed Dirac. For diffusion models that rely on Gaussian priors, we show that the optimal $\pi^*$ can be achieved with slight methodological modifications in Appendix B.1.

Despite these theoretical guarantees, a key practical concern with gradient guidance is the reliability of its one-pass Monte Carlo (MC) estimation of the soft value function. Unfortunately, we find the estimated scores highly biased in early stages of generation when inputs are noisy and the probability mass $\pi(\mathbf{m}|\mathbf{s}_i)$ is not concentrated. As shown in Fig. 4(a), the correlations between 1-pass and 64-pass MC estimation of $V_{\pi_{\text{ref}}}$ are near zero during the first 40% sampling steps, leading to inconsistent guidance. In addition, removing gradient guidance in these steps has a negligible impact on the success rate (see definition in Sec. 5.1) of the generated molecules.

**Analysis for Direct Preference Optimization** DPO (Rafailov et al., 2023) is a fine-tuning approach based on the Bradley–Terry (BT) probabilistic model (Bradley & Terry, 1952) that aligns generative models with a pairwise preference dataset $\mathcal{D}$ by maximizing the likelihood of $\pi_\Phi$ to generate winning samples while minimizing it for losing samples. Following Diffusion-DPO (Wallace et al., 2024), existing approaches (Gu et al., 2024; Cheng et al., 2024; Huang & Zhang, 2025) approximate the probability of the overall sampling trajectory with forward probability $q(\mathbf{s}_i|\mathbf{m})$, detailed as follows:

$$\mathcal{L} = -\mathop{\mathbb{E}}_{\substack{(\mathbf{m}^+, \mathbf{m}^-) \sim \mathcal{D}, i \sim U(1,N), \\ \mathbf{s}_i^+ \sim q(\cdot|\mathbf{m}^+), \mathbf{s}_i^- \sim q(\cdot|\mathbf{m}^-)}} \log \sigma \left( \beta \left[ \log \frac{p_T(\mathbf{s}_{i+1}^+|\mathbf{s}_i^+, \mathbf{a}_i^+)}{p_T(\mathbf{s}_{i+1}^+|\mathbf{s}_i^+, \hat{\mathbf{a}}_i^+)} - \log \frac{p_T(\mathbf{s}_{i+1}^-|\mathbf{s}_i^-, \mathbf{a}_i^-)}{p_T(\mathbf{s}_{i+1}^-|\mathbf{s}_i^-, \hat{\mathbf{a}}_i^-)} \right] \right), \tag{6}$$

where $\mathbf{a}_i = \Phi(\mathbf{s}_i; t_i), \hat{\mathbf{a}}_i = \Phi_{\text{ref}}(\mathbf{s}_i; t_i)$ are obtained by the current and reference policy models, $U(\cdot, \cdot)$ is the uniform distribution, and $\sigma$ denotes sigmoid. We refer readers to Wallace et al. (2024); Gu et al. (2024) for detailed proofs for the following proposition:

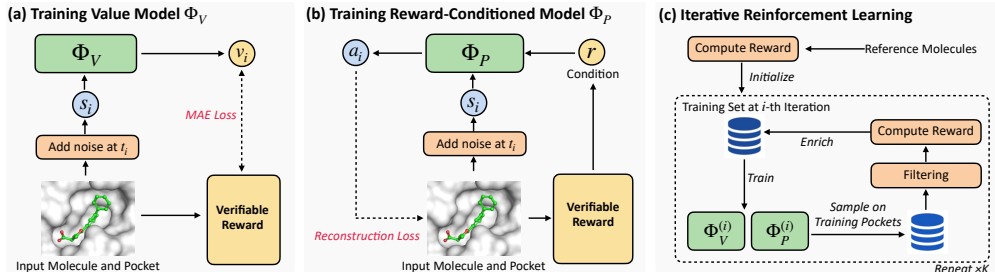

Figure 2: **The training process of RL4SBDD-M. (a) Training the value model.** The model performs one-pass MC estimation of soft values by predicting the reward of a molecule based on its noisy version. **(b) Training the policy model.** Rewards are added as extra conditions to reconstruct the molecule. **(c) The iterative RL procedure.** Generations from the previous policies are added to the dataset for subsequent training.

**Proposition 2.** *Eq. 6 is a upper bound of $\mathcal{D}_{KL}(\pi_\Phi \| \pi^*)$, where $\pi^*$ is the optimal policy in Eq. 2.*

Notably, existing DPO implementations in SBDD construct preference pairs by selecting only the best and worst molecules among the $N_K$ candidates per protein target. Such a design avoids training on preference samples with nearly identical rewards, which leads to likelihood displacement (Razin et al., 2025; Chen et al., 2025) and suboptimal results (Gu et al., 2024). However, it distorts the pairwise structure of the BT model, *i.e.*, $\Pr(\mathbf{m}_1 \succ \mathbf{m}_2) = \sigma(r(\mathbf{m}_1) - r(\mathbf{m}_2))$ for all $\binom{N_K}{2}$ molecule pairs $(\mathbf{m}_1, \mathbf{m}_2)$, by concentrating the probability mass on a single winner (Azar et al., 2024). Consequently, the majority of the data remains unused during training.

**Analysis for Off-Policy Distribution Shift** To reduce the training cost, both gradient guidance and DPO train their models by applying forward probability $q(\mathbf{s}_i|\mathbf{m})$ to obtain noisy versions of molecules from an offline reference dataset, rather than by generating $\mathbf{s}_i$ along the sampling trajectory from $\mathbf{s}_0$. However, during actual sampling, the state distribution shifts toward high-reward regions, raising concerns about off-policy distributional shift (Kumar et al., 2019), *i.e.*, the models may produce erroneous outputs on states outside the support of the training data.

## 4 METHOD

In this section, we present RL4SBDD-M, a novel approach within our framework that addresses the limitations of prior methods. First, we introduce a reward-conditioned policy model (Sec. 4.1) to bypass value estimation and enhance data utilization. Next, we present an iterative RL paradigm (Sec. 4.2) designed to mitigate off-policy distribution shift. Finally, we detail our sampling strategy with mixed guidance and noise reduction to improve sampling quality and efficiency (Sec. 4.3).

### 4.1 A POLICY MODEL CONDITIONED ON REWARDS

In light of the conditioning strategy in flow matching (Lipman et al., 2022), we envision that directly proposing an advantageous action $\mathbf{a}_i$ is more tractable than finding an accurate estimation of soft values in early sampling stages. Furthermore, we expect the model to leverage supervision from all training samples, rather than focusing solely on likelihood maximization for high-reward molecules.

To this end, we introduce a conditional policy model $\Phi_P(\mathbf{s}_i|R; t_i)$ that incorporates rewards $R$ as auxiliary condition labels (Fig. 2(b)). During training, we enforce the model to reconstruct the molecule $\mathbf{m}$ from its noisy version $\mathbf{s}_i$ given its reward $R = r(\mathbf{m})$, thereby harvesting supervision signals from every training sample. Since $\Phi_P$ has learned to generate molecules whose reward scores are consistent with the desired $R$ (Fig. 1(b)), we condition the model on a sufficiently large reward at inference time. Inspired by classifier-free guidance (Ho & Salimans, 2021), we further combine the outputs of the conditional model $\Phi_P$ and the unconditional reference model $\Phi_{\text{ref}}$ to provide gradient-free guidance in the following:

$$\tilde{\mathbf{a}}_i = \Phi_P(\mathbf{s}_i|R; t_i) + (w - 1) \cdot (\Phi_P(\mathbf{s}_i|R; t_i) - \Phi_{\text{ref}}(\mathbf{s}_i; t_i)), \tag{7}$$

where $w \geq 1$ is a hyperparameter controlling the guidance strength. The term $\Phi_P(\mathbf{s}_i|R; t_i) - \Phi_{\text{ref}}(\mathbf{s}_i; t_i)$ reflects the difference between an advantageous action and a baseline action, and Eq.7 performs extrapolation that amplifies the features that lead to higher rewards.

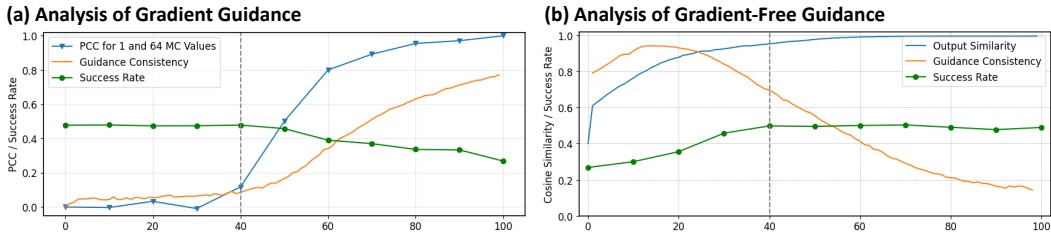

Figure 3: **The sampling procedure of RL4SBDD-M.** We apply gradient-free guidance for the first $M = \lfloor \tau N \rfloor$ steps and gradient guidance for the remaining steps and introduce a noise reduction strategy.

**(a) Analysis of Gradient Guidance**

**(b) Analysis of Gradient-Free Guidance**

Figure 4: **Analysis of mixed guidance. (a) Analysis of gradient guidance.** We evaluate the Pearson correlations between 1-pass and 64-pass Monte Carlo value estimations. We also report the success rate by removing gradient guidance in the first $k$ sampling steps, and guidance consistency measured by the cosine similarity of gradients between consecutive steps. **(b) Analysis of gradient-free guidance.** We show the cosine similarity of the predictions of $\Phi_P$ and $\Phi_{\text{ref}}$ at each step and cosine similarity of the guidance vectors $\Phi_P(\mathbf{s}_i | R; t_i) - \Phi_{\text{ref}}(\mathbf{s}_i)$ between consecutive steps. We also report the success rate by substituting $\Phi_P$ with $\Phi_{\text{ref}}$ in the last $N - k$ sampling steps. Please refer to Appendix E.1 for experiment details and more analysis.

**Proposition 3.** *Using $\tilde{\mathbf{a}}_i$ in Eq. 7 to perform sampling in Eq. 3 is an improved policy over $\pi_{ref}$, and the guidance weight $w$ balances reward maximization and adherence to the reference policy.*

The proof of the proposition is in Appendix B.2.

Notably, a concurrent work (Hu et al., 2025a) introduces a similar approach to guide diffusion models in SBDD. In comparison, we condition on any real-valued reward $R \in \mathbb{R}$, whereas Hu et al. (2025a) is limited to discrete property labels $c \in \{\emptyset, 0, 1\}$. Moreover, we provide theoretical justifications from the RL perspective and introduce additional training and sampling strategies.

### 4.2 ITERATIVE REINFORCEMENT LEARNING

To mitigate off-policy distribution shift, we draw inspiration from Tu et al. (2025); Fan et al. (2025) and adopt an iterative, semi-online RL paradigm (Lanchantin et al., 2025). As illustrated in Fig. 2(c), we initialize the policy and value models by training on the original dataset. At the $i$-th iteration, we sample $K$ molecules for each pocket using current models and the sampling procedure described in Sec. 4.3. We filter out duplicates, incomplete molecules, and PB-invalid molecules (Buttenschoen et al., 2024), and compute their rewards. These newly generated molecules are added to the training set, progressively expanding the training distribution with generations from the updated policies. The complete training process is detailed in Algorithm 1.

### 4.3 SAMPLING WITH MIXED GUIDANCE AND REDUCED NOISE

In practice, combining gradient guidance and gradient-free guidance during sampling achieves the best results. However, this comes at additional computation cost, including an extra forward pass through $\Phi_{\text{ref}}$ and a backward pass through $\Phi_V$. As shown in Fig. 4(a) and discussed in Sec. 3.2, gradient guidance is largely ineffective in early sampling steps. Besides, Fig. 4(b) shows that in the final 60% generation steps, $\Phi_P$ and $\Phi_{\text{ref}}$ produce similar predictions, leading to inconsistent guidance. As a result, removing gradient-free guidance in these steps has little impact on sampling quality, consistent with previous observations in image generation (Castillo et al., 2025; Zhang et al., 2025a). To address this, we propose a mixed guidance strategy that improves sampling efficiency without

Table 1: Performance comparison on the CrossDocked2020 test set with an average molecular size of $\sim 22.8$. (↑)/(↓) indicates the higher/lower the better. SR is short for success rate. Avg. and Med. are short for average and median. - indicates that the results are not reported in the original paper. The best and second best results are marked **bold** and underlined.

| Methods | Vina Score (↓) Avg. | Med. | Vina Min (↓) Avg. | Med. | Vina Dock (↓) Avg. | Med. | QED (↑) Avg. | SA (↑) Avg. | Clash (↓) Avg. | Div (↑) Avg. | SPM (↓) Avg. | SR (↑) % |
|---|---|---|---|---|---|---|---|---|---|---|---|---|
| Reference | -6.36 | -6.46 | -6.71 | -6.49 | -7.45 | -7.26 | 0.48 | 0.73 | 5.46 | - | - | 25.0% |
| AR | -5.75 | -5.64 | -6.18 | -5.88 | -6.75 | -6.62 | 0.51 | 0.63 | **4.18** | 0.70 | 196.59 | 6.9% |
| Pocket2Mol | -5.14 | -4.70 | -6.42 | -5.82 | -7.15 | -6.79 | **0.57** | 0.76 | 6.22 | 0.69 | 25.04 | 24.4% |
| TargetDiff | -5.47 | -6.30 | -6.64 | -6.83 | -7.80 | -7.91 | 0.48 | 0.58 | 10.67 | 0.72 | 34.28 | 10.5% |
| DecompDiff | -5.19 | -5.27 | -6.03 | -6.00 | -7.03 | -7.16 | 0.51 | 0.66 | 14.23 | 0.73 | 61.89 | 14.9% |
| MolCRAFT | -6.55 | -6.95 | -7.21 | -7.14 | -7.67 | -7.82 | 0.50 | 0.67 | 6.91 | 0.70 | **1.86** | 26.8% |
| RGA | - | - | - | - | -8.01 | -8.17 | **0.57** | 0.71 | - | 0.41 | 50.50 | 46.2% |
| DecompOpt | -5.75 | -5.97 | -6.58 | -6.70 | -7.63 | -8.02 | 0.56 | 0.73 | 16.60 | 0.63 | 89.41 | 39.4% |
| CIDD | - | - | - | - | -9.02 | - | 0.53 | 0.69 | - | 0.70 | - | 37.9% |
| TAGMol | -7.02 | -7.77 | -7.95 | -8.07 | -8.59 | -8.69 | 0.55 | 0.56 | 7.41 | 0.69 | 73.55 | 11.1% |
| MolJO | -7.52 | -8.02 | -8.33 | -8.34 | -9.05 | -9.13 | 0.56 | 0.78 | 6.72 | 0.66 | 16.10 | 51.3% |
| AliDiff | -7.07 | -7.95 | -8.09 | -8.17 | -8.90 | -8.81 | 0.50 | 0.57 | 8.51 | 0.73 | 54.31 | 10.6% |
| MolFORM | -6.16 | - | -7.18 | - | -8.13 | - | 0.50 | 0.65 | 5.80 | **0.77** | - | - |
| DecompDPO | -6.10 | -7.22 | -7.93 | -8.16 | **-9.26** | -9.23 | 0.48 | 0.64 | - | 0.62 | - | 36.2% |
| Lee et al. (2025) | -7.18 | -7.38 | -7.89 | -7.77 | -8.62 | -8.64 | 0.55 | 0.74 | 6.69 | - | - | - |
| DiffGui | -5.90 | -5.59 | -6.89 | -6.95 | -7.90 | -8.07 | 0.50 | 0.65 | 9.67 | 0.70 | 110.80 | 23.3% |
| RL4SBDD-M (Ours, $\epsilon$=1) | -7.73 | -8.58 | -8.59 | -8.83 | -9.06 | -9.26 | 0.55 | 0.81 | 6.07 | 0.68 | - | 55.5% |
| RL4SBDD-M (Ours, $\epsilon$=0.2) | **-7.97** | **-8.70** | **-8.67** | **-8.87** | -9.20 | **-9.28** | 0.54 | **0.85** | 5.89 | 0.53 | 3.49 | **58.1%** |

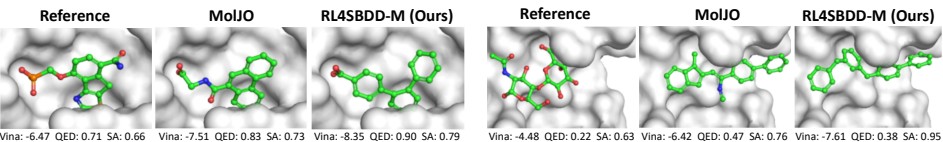

**Reference** | **MolJO** | **RL4SBDD-M (Ours)** | **Reference** | **MolJO** | **RL4SBDD-M (Ours)**

Vina: -6.47 QED: 0.71 SA: 0.66 | Vina: -7.51 QED: 0.83 SA: 0.73 | Vina: -8.35 QED: 0.90 SA: 0.79 | Vina: -4.48 QED: 0.22 SA: 0.63 | Vina: -6.42 QED: 0.47 SA: 0.76 | Vina: -7.61 QED: 0.38 SA: 0.95

Figure 5: **Visualization of the reference molecules and generations from MolJO and RL4SBDD-M** on protein 3KC1 (left) and 5JTN (right).

sacrificing performance. Specifically, we apply gradient-free guidance for the first $M = \lfloor \tau N \rfloor$ steps and gradient guidance in Eq. 4 for the last $N - M$ steps, where $\tau \in [0, 1]$ is a hyperparameter.

Moreover, by interpreting the state transition kernel $p_T(\mathbf{s}_{i+1}|\mathbf{s}_i, \mathbf{a}_i)$ in Eq. 3 as a predictor-corrector step (Song & Ermon, 2019), we note that the benefit of an improved action $\mathbf{a}_i$ in the predictor step could be diminished by the injected noise $\sigma_i \mathbf{w}$ in the corrector step. To this end, we introduce a reduction factor $\epsilon \in (0, 1]$, modifying the noise to $\epsilon \sigma_i \mathbf{w}$. This adjustment allows users to trade off between diversity and quality, either placing greater emphasis on exploitation with $\mathbf{a}_i$ or encouraging exploration with the stochastic term $\sigma_i \mathbf{w}$. The overall sampling procedure is illustrated in Fig. 3 and detailed in Algorithm 2.

## 5 EXPERIMENTS

### 5.1 EXPERIMENT SETUP

**Training Dataset** We adopt the CrossDocked2020 dataset (Francoeur et al., 2020) for training and evaluation. We follow the pre-processing steps in Luo et al. (2021); Peng et al. (2022) by filtering out poses with RMSD>1Å and duplicate proteins with >30% sequence identity, yielding 100K training poses and 100 test pockets. We sample 100 molecules for each test pocket.

**Implementation Details** While RL4SBDD-M can be applied to arbitrary verifiable rewards, we follow Qiu et al. (2025) and adopt a combination of binding affinity and synthesizability as optimization objectives in the main experiments to investigate if our model achieves superior reward maximization. Specifically, we use the normalized Vina Score and SA scores, and treat them as 2 independent dimensions for the conditional inputs to $\Phi_P$ and the prediction objectives of $\Phi_V$. For $\Phi_V$ and $\Phi_P$, we adopt the same architecture as MolCRAFT (Qu et al., 2024), an advanced Bayesian flow network in SBDD, with $N = 100$ sampling steps (detailed in Appendix A). The value model $\Phi_V$ is implemented as a 2-layer MLP with ReLU activation applied to ligand atom representations, with outputs averaged to predict values and trained using MAE loss. The gradients over multiple

objectives are averaged during sampling. For the policy model $\Phi_P$, the reward conditions are concatenated as additional input features for each ligand atom, and the training objectives are the same as Qu et al. (2024). We fix hyperparameters with guidance weight $1/\lambda = 30, w = 1.2$ and switch point $\tau = 0.4$ during sampling. We conduct 2 RL iterations with $K = 8$, expanding the training set to 826K and 1.4M samples after each iteration. Additional details are provided in Appendix C.

**Baselines** We compare our model with baselines categorized as follows: (1) pre-trained generative models, including **AR** (Luo et al., 2021), **Pocket2Mol** (Peng et al., 2022), **TargetDiff** (Guan et al., 2023a), **DecompDiff** (Guan et al., 2023b), and **MolCRAFT** (Qu et al., 2024), (2) search-based models, including **RGA** (Fu et al., 2022), **DecompOpt** (Zhou et al., 2024), and **CIDD** (Gao et al., 2025), and (3) preference-aligned models, including **TAGMol** (Dorna et al., 2024), **MolJO** (Qiu et al., 2025), **AliDiff** (Gu et al., 2024), **MolFORM** (Huang & Zhang, 2025), **DecompDPO** (Cheng et al., 2024), **DiffGui** (Hu et al., 2025a), **TacoGFN (Shen et al., 2024), 3DSynthFlow (Shen et al., 2025)**, and Lee et al. (2025). Details are available at Appendix D.

**Evaluation Metrics** We evaluate our models along the following dimensions: (1) Pocket-binding affinity calculated by AutoDock Vina (Eberhardt et al., 2021), including **Vina Score**, *i.e.*, raw energy of the generated pose, **Vina Min**, *i.e.*, energy after local energy minimization, and **Vina Dock**, *i.e.*, energy after global re-docking. (2) Molecular properties, including **QED** that measures drug likeliness, and **SA** that measures synthetic accessibility. (3) Conformational plausibility, measured by the number of **Steric Clashes**, *i.e.*, ligand atoms within a threshold distance to protein atoms. Note that steric clashes do not necessarily indicate violations of physical constraints, as pocket structures may adjust upon binding (Harris et al., 2023). (4) Sampling diversity and efficiency, with **Div** measured by Tanimoto similarity (Bajusz et al., 2015) over Morgan fingerprints of generated molecules, and **SPM** defined as the average seconds per generated molecule on a single NVIDIA A800 GPU. (5) **Success Rate**, a holistic metric that captures the percentage of molecules with Vina Dock<8.18, QED>0.25, SA>0.59, following Long et al. (2022).

## 5.2 MAIN RESULTS

The performance comparison between RL4SBDD-M and baselines is summarized in Tab. 1, with more detailed results provided in Appendix E.2 and Appendix E.4. The key findings are as follows:

**RL4SBDD-M Outperforms the Best Model by 6.8% in Success Rate with a 4.6× Speedup** As shown in Tab. 1, our model achieves the best overall performance with a median Vina Score of -8.70 kcal/mol, an average SA of 0.85, and an average success rate of 58.1%. We also provide visualizations in Fig. 5, where RL4SBDD-M generates better ligands compared to the state-of-the-art MolJO across all metrics. In addition, RL4SBDD-M enjoys high sampling efficiency and is ~4.6 times faster than MolJO, thanks to our mixed guidance strategy.

**RL4SBDD-M Effectively Maximizes the Reward Objectives** Compared to the reference model MolCRAFT, RL4SBDD-M achieves 21.7% relative gains in Vina Score and 26.9% relative gains in SA. The distribution shift of these metrics shown in Figs. 8–12 further confirms the effectiveness of our approach. Notably, using the original noise schedule ($\epsilon = 1$), RL4SBDD-M maintains a high diversity of 0.68, which is comparable with MolCRAFT (0.70).

**RL4SBDD-M Generalizes to Large OOD Molecules** As suggested by Pan et al. (2003), incorporating more atoms into the ligand can lead to higher binding affinity by forming more protein-ligand interactions, but often at the cost of reduced drug-likeness and synthetic accessibility. While we restrict the average size of generated molecules by reference molecules in our main experiments to ensure fair comparison, we investigate RL4SBDD-M's performance on larger, OOD molecules. As shown in Tab. 5, our model achieves the best Vina Dock, SA, and success rate, highlighting its robustness in optimizing reward objectives.

## 5.3 TEST-TIME SCALING ANALYSIS

Motivated by the success of reasoning LLMs (Snell et al., 2024), we conduct a test-time scaling analysis to investigate if RL4SBDD-M could benefit from virtual screening like prior search-based methods (Fu et al., 2022; Cheng et al., 2024; Reidenbach, 2024). Unlike these methods, which iteratively modify molecular components, we simply perform repeated sampling with the atom count sampled from a prior distribution based on the pocket size. Following (Zhou et al., 2024), we select

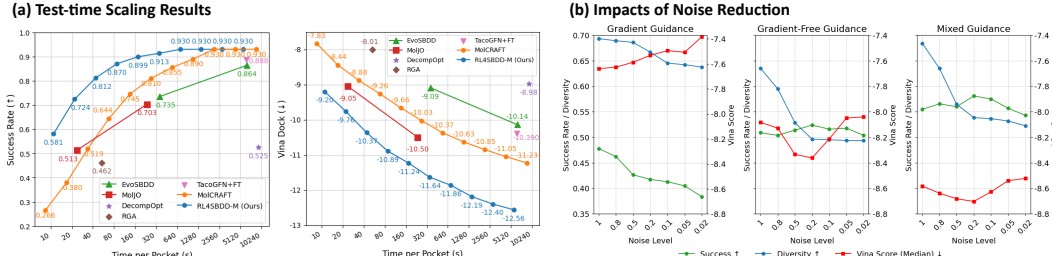

Figure 6: **Analysis of test-time scaling and noise reduction. (a) Test-time scaling performance.** We perform repeated sampling and report the success rate and Vina Dock for the best of all generated samples within a computation budget that includes sampling and oracle evaluation costs. **(b) Impacts of noise reduction.** We show the diversity, success rate, and median Vina Score for $\Phi_V$, $\Phi_P$, and mixed guidance with different $\epsilon$.

Table 2: Performance of RL4SBDD-M after RL iterations and removing different design components. (↑)/(↓) indicates the higher/lower the better. The best and second best results are marked **bold** and underlined. w/ full guide: applying both guidance in all steps. w/o $\Phi_P$: removing gradient-free guidance in early sampling and using $\Phi_{\text{ref}}$ as the policy model. w/o $\Phi_V$: removing gradient guidance in late sampling.

| #Iters | Model | Vina Score (↓) Avg. | Med. | Vina Min (↓) Avg. | Med. | Vina Dock (↓) Avg. | Med. | QED (↑) Avg. | SA (↑) Avg. | Div (↑) Avg. | SPM (↓) Avg. | SR (↑) % |
|---|---|---|---|---|---|---|---|---|---|---|---|---|
| 2 | RL4SBDD-M ($\epsilon$=0.2) | **-7.97** | -8.70 | -8.67 | -8.87 | -9.21 | -9.28 | 0.54 | **0.85** | 0.53 | 3.49 | 58.1% |
|  | RL4SBDD-M ($\epsilon$=1) | -7.73 | -8.58 | -8.59 | -8.83 | -9.06 | -9.26 | 0.55 | 0.81 | 0.68 | - | 55.5% |
|  | w/ full guide | -7.92 | **-8.80** | **-8.71** | **-8.88** | **-9.32** | **-9.31** | 0.53 | 0.83 | 0.53 | 6.62 | **58.2%** |
|  | w/o $\Phi_P$ | -7.49 | -8.18 | -8.25 | -8.38 | -9.00 | -9.07 | **0.58** | 0.76 | **0.69** | 2.60 | 47.4% |
|  | w/o $\Phi_V$ | -7.29 | -8.38 | -7.92 | -8.42 | -8.86 | -9.03 | 0.54 | **0.85** | 0.53 | 1.34 | 54.2% |
| 1 | RL4SBDD-M ($\epsilon$=0.2) | -7.74 | -8.55 | -8.55 | -8.72 | -9.15 | -9.25 | 0.55 | 0.81 | 0.58 | - | 55.7% |
|  | w/o $\Phi_P$ | -7.20 | -8.16 | -8.26 | -8.40 | -9.06 | -9.14 | 0.57 | 0.76 | 0.68 | - | 47.8% |
|  | w/o $\Phi_V$ | -7.17 | -7.98 | -8.08 | -8.17 | -8.66 | -8.92 | 0.55 | 0.81 | 0.59 | - | 47.7% |
| 0 | RL4SBDD-M ($\epsilon$=0.2) | -7.43 | -8.33 | -8.34 | -8.51 | -8.94 | -9.10 | 0.57 | 0.75 | 0.59 | - | 52.0% |
|  | w/o $\Phi_P$ | -6.61 | -7.61 | -7.74 | -8.17 | -8.70 | -8.83 | 0.56 | 0.77 | 0.68 | - | 46.6% |
|  | w/o $\Phi_V$ | -7.01 | -7.66 | -7.76 | -7.78 | -8.26 | -8.27 | 0.57 | 0.76 | 0.63 | - | 41.5% |
| - | w/o both | -6.48 | -6.84 | -7.21 | -7.15 | -7.78 | -7.91 | 0.50 | 0.68 | 0.70 | **0.93** | 24.9% |

the best molecule among $N$ generations with oracle-based Z-score, and report success rate, Vina Dock, and the overall computational cost for both generation and oracle evaluation across models. We also introduce a self-ranking strategy (Jumper et al., 2021) that selects the best molecule based on the predicted scores of $\Phi_V$. Details and analysis for this setting are available in Appendix E.3.

As shown in Fig. 6(a), both success rate and binding affinity steadily improve with increased inference computation, with RL4SBDD-M delivering the strongest performance. Notably, **RL4SBDD-M achieves a success rate of 93.0% via 50 generations**, surpassing EvoSBDD by 6.6% while requiring just 8.7% of its computational cost. The success rate plateaus with even more generations, since on several small test pockets it is impossible to achieve the affinity threshold of –8.18 kcal/mol. Interestingly, MolCRAFT also attains a success rate of 93.0% but requires 256 generations. Nevertheless, RL4SBDD-M yields substantially better Vina Dock scores across different computation budgets. These results indicate that our model does not simply concentrate the probability mass (Yue et al., 2025) but effectively explores the chemical space to discover high-reward molecules.

## 5.4 Ablation Studies

To assess the contribution of each design component, we conduct a series of ablation studies with key results summarized below (see Appendix E.5 for more analysis).

**Mixed Guidance Improves Sampling Efficiency without Sacrificing Performance** As shown in Tab. 2, removing either gradient-free guidance in early sampling or gradient guidance in later stages results in significant performance degradation. Conversely, applying both forms of guidance throughout all steps maintains performance but leads to a higher computational cost, confirming the advantage of our mixed guidance strategy. We further explore the choice of the switching point $\tau$ in Fig. 15, where the optimal value $\tau = 0.4$ aligns with our earlier observations in Fig. 4.

**Reducing Noise Trades Diversity for Quality** We experiment with different noise scales $\epsilon$ in Fig. 6(b). Interestingly, the performance of gradient guidance with $\Phi_V$ declines as noise decreases, while

$\Phi_P$ and mixed guidance achieve improved sampling performance at the expense of reduced diversity as $\epsilon$ decreases to 0.2, offering a trade-off in different practical scenarios. Notably, the generated molecules retain some diversity even as $\epsilon \to 0$, primarily due to floating-point errors accumulated during the sampling process with back correction in Qu et al. (2024).

**Iterative RL Boosts Performance** Tab. 2 reports model performance after each RL iteration. The conditional policy model $\Phi_P$ achieves substantial gains in binding affinity and SA after each iteration, while $\Phi_V$ shows limited improvements in the second iteration, potentially reflecting the greater difficulty of learning a generator compared to a verifier (Swamy et al., 2025). In addition, sampling diversity decreases over iterations only when $\epsilon = 0.2$, suggesting a joint effect from action space concentration and noise reduction.

## 5.5 ANALYSIS ON MORE REWARD FUNCTIONS

As suggested by several RL studies (Skalse et al., 2022; Karwowski et al., 2024), optimizing an imperfect proxy reward function beyond a certain threshold may lead to unintended ways to maximize the proxy. In SBDD, we observe a prevalence for generating more aromatic rings to achieve higher binding scores by forming more $\pi$-$\pi$ and $\pi$-cation interactions, which is shared across various alignment methods and may lead to high lipophilicity of molecules and undesirable ADME properties. This stems from the reward formulation for higher affinity and synthesizability, and adopts the Vina Score and SA score as proxies. We defer readers to Appendix E.6 for more analysis and discussion.

Table 3: Performance of RL4SBDD-M in optimizing different combinations of reward objectives. We report the median Vina Score and average QED, SA, and cLogP.

| Model | Objective | Vina | QED | SA | cLogP |
|---|---|---|---|---|---|
| Ref | - | -6.46 | 0.48 | 0.73 | 0.89 |
| MolJO | Affinity+SA | -8.02 | 0.56 | 0.78 | 4.20 |
| | Affinity+SA+QED | -7.47 | 0.62 | 0.73 | - |
| RL4SBDD-M | Affinity+SA | -8.70 | 0.54 | 0.85 | 5.48 |
| | Affinity+SA+QED | -8.50 | 0.66 | 0.80 | 4.26 |
| | Affinity+SA+cLogP | -7.88 | 0.58 | 0.77 | 3.08 |

Fortunately, the architectural design of RL4SBDD-M is reward-agonistic, and we explore a broader range of combined reward objectives to validate its generalization capabilities and seek to mitigate the undesired patterns by optimizing Affinity+SA only. Specifically, we incorporate drug-likeness and water-octanol partition coefficient as additional reward objectives and adopt QED and calculated LogP scores (cLogP) as proxies. We consider cLogP=1 as desirable (Arnott & Planey, 2012).

As shown in Tab. 3, RL4SBDD-M achieves a balanced trade-off between the optimization objectives. Optimizing Vina score and SA leads to a moderate increase in QED and significant increases in lipophilicity. Incorporating either QED or cLogP into reward objectives is effective but comes at the cost of a decrease in Vina score and SA at distinct levels, which underscores the complexity of multi-objective optimization. Fortunately, given the same set of optimization objectives, RL4SBDD-M consistently achieves the best results in all metrics we aim to improve.

## 6 CONCLUSION AND DISCUSSION

In this work, we introduce RL4SBDD, a systematic RL framework that unifies existing preference alignment strategies for generative models in structure-based drug design and analyzes their limitations. Drawing insights from the RL community, we introduce RL4SBDD-M, a novel approach that integrates a reward-conditioned policy model, iterative reinforcement learning, and sampling with mixed guidance and reduced noise. Experiment results show that RL4SBDD-M generates high-reward molecules, achieving state-of-the-art performance on the CrossDocked2020 benchmark while maintaining high sampling efficiency. Moreover, when equipped with oracles, our model attains outstanding results via test-time scaling.

Collectively, these results demonstrate the effectiveness and potential of our approach, and also reveals several opportunities for further exploration: (1) developing more advanced algorithms and online RL paradigms to strengthen optimization, (2) investigating a wider range of reward objectives and more realistic in silico validation for practically desirable molecules, and (3) extending our framework to other domains such as protein and material design.

# 7 ETHICS STATEMENT

This work focuses on developing in-silico tools for lead compound design, with potential positive impacts including accelerating the pharmaceutical pipeline and facilitating the development of novel therapeutics. To mitigate risks associated with the misuse of such tools in generating harmful bio-agents, we emphasize that any downstream applications should be accompanied by comprehensive wet-lab validation and thorough ethical review.

# 8 REPRODUCIBILITY STATEMENT

We ensure reproducibility through detailed descriptions of our models and algorithms provided in Appendix C, along with the experiment settings outlined in Sec. 5.1. Additionally, we provide code implementation in supplementary materials to facilitate replication of our results.

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

# A    BAYESIAN FLOW NETWORKS IN STRUCTURE-BASED DRUG DESIGN

**An Overview of Bayesian Flow Networks (BFN)** BFN (Graves et al., 2023) formulates generative modeling as an iterative exchange of messages between a sender and a receiver. The sender distribution $p_S(\mathbf{y}_i|\mathbf{m}; \alpha_i)$ generates a perturbed representation $\mathbf{y}_i$ by injecting noise to each dimension of the molecule $\mathbf{m}$ and sends it to the receiver. The amount of perturbation is defined by a noise schedule $\beta_i = \sum_{j=0}^{i} \alpha_j$ with respect to the accuracy $\alpha_i$. Instead of operating on the noisy sample space of $\mathbf{y}_i$, the receiver maintains Bayesian posterior parameters $\theta_i$ that capture the key features of molecules. Starting from a prior $\theta_0$, these parameters are progressively updated based on noisy observations $\mathbf{y}_i$ from the sender with a Bayesian update function $h(\theta_{i-1}, \mathbf{y}_i; \alpha_i)$. This process yields the Bayesian update distribution as follows:

$$p_U(\theta_i|\theta_{i-1}, \mathbf{m}; \alpha_i) = \underset{\mathbf{y}_i \sim p_S}{\mathbb{E}} \left[ \delta \left( \theta_i - h(\theta_{i-1}, \mathbf{y}_i; \alpha_i) \right) \right], \tag{8}$$

where $\delta(\cdot)$ is Dirac distribution.

According to the additive property of accuracy (Graves et al., 2023), the Bayesian flow distribution over the parameters $\theta_i$ could be derived in a closed form, thereby avoiding the need for computationally expensive simulation during training:

$$p_F(\theta_i|\mathbf{m}; \beta_i) = \underset{\theta_1, \theta_2, \cdots, \theta_{i-1} \sim p_U}{\mathbb{E}} p_U(\theta_i|\theta_{i-1}, \mathbf{m}; \alpha_i) = p_U(\theta_i|\theta_0, \mathbf{m}; \beta_i). \tag{9}$$

Given the Bayesian posterior $\theta_{i-1}$ at time $t_i$, the goal of BFN is to estimate the output distribution $p_O(\hat{\mathbf{m}}|\Phi(\theta_{i-1}, t_i))$ of the clean molecule $\mathbf{m}$ via a neural network $\Phi$. The receiver distribution is then obtained by applying the sender's corruption process and marginalizing over $\hat{\mathbf{m}}$:

$$p_R(\mathbf{y}_i|\theta_{i-1}; t_i, \alpha_i) = \underset{\hat{\mathbf{m}} \sim p_O}{\mathbb{E}} \left[ p_S(\mathbf{y}_i|\hat{\mathbf{m}}; \alpha_i) \right]. \tag{10}$$

The training objective is the KL-divergence between the sender and receiver distributions:

$$\mathcal{L} = \underset{\mathbf{m} \sim \mathcal{D}, i \sim U(1,\hat{N}), \ \theta_{i-1} \sim p_F}{\mathbb{E}} \left[ \mathcal{D}_{KL}(p_S(\mathbf{y}_i|\mathbf{m}; \alpha_i) \| p_R(\mathbf{y}_i|\theta_{i-1}; t_i, \alpha_i)) \right], \tag{11}$$

where $\mathcal{D}$ is the training dataset, $U(\cdot, \cdot)$ is uniform distribution, and $\hat{N}$ is the number of message exchange iterations.

**Joint Modeling of Continuous and Discrete Modalities in Molecular Design** As defined in Sec. 3, each molecule consists of continuous atom coordinates $\mathbf{x}$ and atom types $\mathbf{v}$ (we omit the subscript $\cdot_{\mathcal{M}}$ for brevity). A key advantage of applying BFN in molecular generation is that it models data distribution in the parameter space, enabling a unified handling of both continuous and discrete modalities. Specifically, the parameters can be decomposed as $\theta_i \overset{\text{def}}{=} [\theta_i^{\mathbf{x}}, \theta_i^{\mathbf{v}}]$. The continuous atom coordinates $\mathbf{x}$ is modeled by $\mathbf{x}_i \sim \mathcal{N}(\mathbf{x}_i|\mu_i, \rho_i^{-1}\mathbf{I})$ with $\theta_i^{\mathbf{x}} = \{\mu_i, \rho_i\}$, where $\mu_i$ is predicted by the model and $\rho_i$ is determined by the accuracy schedule $\alpha_i$. The corresponding sender and receiver distributions for $\mathbf{x}_i$ and the Bayesian update function are then defined as follows:

$$
\begin{aligned}
p_S^{\mathbf{x}}(\mathbf{y}_i^{\mathbf{x}}|\mathbf{x}; \alpha_i^{\mathbf{x}}) &= \mathcal{N}(\mathbf{y}_i^{\mathbf{x}}|\mathbf{x}, (\alpha_i^{\mathbf{x}})^{-1}\mathbf{I}) \\
p_R^{\mathbf{x}}(\mathbf{y}_i^{\mathbf{x}}|\theta_{i-1}^{\mathbf{x}}; t_i) &= \mathcal{N}(\mathbf{y}_i^{\mathbf{x}}|\Phi(\theta_{i-1}; t_i)^{\mathbf{x}}, (\alpha_i^{\mathbf{x}})^{-1}\mathbf{I}) \\
h(\{\mu_{i-1}, \rho_{i-1}\}, \mathbf{y}_i^{\mathbf{x}}; t_i) &= \left\{ \frac{\mu_{i-1}\rho_{i-1} + \mathbf{y}_i^{\mathbf{x}}\alpha_i^{\mathbf{x}}}{\rho_{i-1} + \alpha_i^{\mathbf{x}}}, \rho_{i-1} + \alpha_i^{\mathbf{x}} \right\}
\end{aligned}
\tag{12}
$$

The training objective for continuous variables is as follows:

$$\mathcal{L}_c = \underset{(\mathbf{x},\mathbf{v}) \sim \mathcal{D}, i \sim U(1,\hat{N}), \theta_{i-1} \sim p_F}{\mathbb{E}} \frac{\alpha_i}{2} \left\| \mathbf{x} - \Phi(\theta_{i-1}; t_i)^{\mathbf{x}} \right\|^2 \tag{13}$$

For discrete atom features $\mathbf{v}$, the parameters are defined on the probability simplex $\theta_i^{\mathbf{v}} \in \mathbb{R}^{K-1}$. The corresponding sender and receiver distributions and the Bayesian update function are given in the following:

$$
\begin{aligned}
p_S^{\mathbf{v}}(\mathbf{y}_i^{\mathbf{v}}|\mathbf{v}; \alpha_i^{\mathbf{v}}) &= \mathcal{N}(\mathbf{y}_i^{\mathbf{v}}|\alpha_i^{\mathbf{v}}(K\mathbf{v} - 1), \alpha_i^{\mathbf{v}}K\mathbf{I}), \\
p_R^{\mathbf{v}}(\mathbf{y}_i^{\mathbf{v}}|\theta_{i-1}^{\mathbf{v}}; \alpha_i') &= \left[ \sum_{k=1}^{K} (\Phi(\theta_{i-1}; t_i)^{\mathbf{v}})_k^{(d)} \cdot p_S^{\mathbf{v}}(\mathbf{y}_i^{\mathbf{v}}|\mathbf{e}_k; \alpha_i^{\mathbf{v}}) \right]_{d=1,2,\cdots,N} \\
h(\theta_{i-1}^{\mathbf{v}}, \mathbf{y}_i^{\mathbf{v}}; t_i) &= \frac{\exp(\mathbf{y}_i^{\mathbf{v}})\theta_{i-1}^{\mathbf{v}}}{\sum_{k=1}^{K} \exp(\mathbf{y}_i^{\mathbf{v}})(\theta_{i-1}^{\mathbf{v}})_k}
\end{aligned}
\tag{14}
$$

Table 4: Analogy between BFN and RL terms in Sec. 3.

| BFN terms | RL terms |
|---|---|
| Parameters $\theta_i$ | States $\mathbf{s}_i \in \mathcal{S}$ |
| Outputs of $\Phi(\theta_{i-1}; t_i)$ | Actions $\mathbf{a}_i \in \mathcal{A}$ |
| Bayesian updates $p_U^{\mathbf{x}}$ and $p_U^{\mathbf{v}}$ | State-transition kernel $p_T(\mathbf{s}_i|\mathbf{s}_{i-1}, \mathbf{a}_i)$ |
| Prior belief $\theta_0$ | Inital state $\mathbf{s}_0 \sim p_I$ |
| Decoding $\theta_{N-1}$ into $\mathbf{m}$ | Output step $p_O$ |
| Bayesian flow distribution $p_F(\theta_i|\mathbf{m}; \beta_i)$ | Forward distribution $q(\mathbf{s}_i|\mathbf{m})$ |

Here $\mathbf{e}_k \in \mathbb{R}^K$ is the one-hot vector for the $k$-th dimension, $(\Phi(\theta_{i-1}; t_i)^{\mathbf{v}})_k^{(d)}$ denotes the predicted probability for the $k$-th class of the $d$-th atom. The training objective is defined as follows:

$$\mathcal{L}_d = \mathbb{E}_{(\mathbf{x},\mathbf{v})\sim\mathcal{D}, i\sim U(1,\hat{N}), \theta_{i-1}\sim p_F} \left[ \mathcal{D}_{KL}(p_S^{\mathbf{v}}(\mathbf{y}_i^{\mathbf{v}}|\mathbf{v}, \alpha_i')||p_R^{\mathbf{v}}(\mathbf{y}_i^{\mathbf{v}}|\theta_{i-1}^{\mathbf{v}}, \alpha_i')) \right] \tag{15}$$

The KL divergence is calculated via Monte Carlo estimations. We refer readers to Graves et al. (2023) for detailed justifications for its modeling of continuous and discrete variables.

**Sampling with Reduced Noise, Back Correction, and Gradient Guidance** In the original BFN, the prior beliefs for continuous and discrete components are initialized as $\theta_0^{\mathbf{x}} = \mathbf{0}$ and $\theta_0^{\mathbf{v}} = [1/K]_{N_{\mathcal{M}} \times K}$ as a uniform distribution, respectively. Then, at each sampling step, the parameters are fed into the neural network to obtain the current estimation of $\hat{\mathbf{m}}_i$ in the following:

$$\hat{\mathbf{m}}_i = (\hat{\mathbf{x}}_i, \hat{\mathbf{v}}_i), \quad \hat{\mathbf{x}}_i = \Phi(\theta_{i-1}; t_i)^{\mathbf{x}}, \quad \hat{\mathbf{v}}_i \sim \text{Catgorical}(\Phi(\theta_{i-1}; t_i)^{\mathbf{v}}), \tag{16}$$

where $\text{Catgorical}(\cdot)$ refers to sampling from the categorical distribution.

Then, a noisy version $\hat{\mathbf{y}}_i$ is obtained by applying the sender distribution to the continuous and discrete variables, which is fed into the Bayesian update function $h$ to obtain the posterior beliefs $\theta_i$:

$$p_U^{\mathbf{x}}(\mu_i|\theta_{i-1}; t_i) = \mathbb{E}_{\hat{\mathbf{y}}_i^{\mathbf{x}}\sim\mathcal{N}(\hat{\mathbf{x}}, (\alpha_i^{\mathbf{x}})^{-1}\mathbf{I})} \left[ \frac{\mu_{i-1}\rho_{i-1} + \hat{\mathbf{y}}_i^{\mathbf{x}}\alpha_i^{\mathbf{x}}}{\rho_{i-1} + \alpha_i^{\mathbf{x}}} \right],$$

$$p_U^{\mathbf{v}}(\theta_i^{\mathbf{v}}|\theta_{i-1}; t_i) = \mathbb{E}_{\hat{\mathbf{y}}_i^{\mathbf{v}}\sim\mathcal{N}(\beta_i^{\mathbf{v}}(K\mathbf{e}_{\hat{\mathbf{v}}_i}-1), \beta_i^{\mathbf{v}}K\mathbf{I})} \delta \left( \theta_i^{\mathbf{v}} - \frac{\exp(\hat{\mathbf{y}}_i^{\mathbf{v}})\theta_{i-1}^{\mathbf{v}}}{\sum_{k=1}^K \exp(\hat{\mathbf{y}}_i^{\mathbf{v}})(\theta_{i-1}^{\mathbf{v}})_k} \right). \tag{17}$$

To enhance the sampling quality of molecules in SBDD, Qu et al. (2024) introduces a sampling process with noise reduction and back correction. Specifically, it bypasses the sampling of $\hat{\mathbf{v}}_i$ by directly plugging the probability $\Phi(\theta_{i-1}; t_i)^{\mathbf{v}}$ into the sender distribution. In addition, it replaces prior predictions $\hat{\mathbf{m}}_0, \hat{\mathbf{m}}_1, \cdots, \hat{\mathbf{m}}_{i-1}$ with the most recent prediction $\hat{\mathbf{m}}_i$ and re-applies Bayesian updates along the trajectory using the additive property of accuracy in Eq. 9 as follows:

$$p_U^{\mathbf{x}}(\mu_i|\theta_{i-1}; t_i) = \mathcal{N}(\mu_i|\gamma_i\Phi(\theta_{i-1}; t_i)^{\mathbf{x}}, \gamma_i(1-\gamma_i)\mathbf{I}),$$

$$p_U^{\mathbf{v}}(\theta_i^{\mathbf{v}}|\theta_{i-1}; t_i) = \mathbb{E}_{\mathbf{y}_i^{\mathbf{v}}\sim\mathcal{N}(\beta_i^{\mathbf{v}}(K\Phi(\theta_{i-1};t_i)^{\mathbf{v}}-1), \beta_i^{\mathbf{v}}K\mathbf{I})} \delta(\theta_i^{\mathbf{v}} - \text{SoftMax}(\hat{\mathbf{y}}_i^{\mathbf{v}})), \tag{18}$$

where $\gamma_i = \beta_i^{\mathbf{x}}/(1-\beta_i^{\mathbf{x}})$.

More recently, Qiu et al. (2025) introduces a regression model $\Phi_V$ to estimate the alignment scores of the molecule $\mathbf{m}$ based on the current posterior belief $\theta_{i-1}$. The gradients of the model are used to guide each sampling step as below (highlighted in red):

$$p_U^{\mathbf{x}}(\mu_i|\theta_{i-1}; t_i) = \mathcal{N}\left( \mu_i \middle| \gamma_i\Phi(\theta_{i-1}; t_i)^{\mathbf{x}} + \frac{\gamma_i(1-\gamma_i)}{\lambda} \cdot \frac{\partial\Phi_V(\theta_{i-1}; t_i)}{\partial\mu_{i-1}}, \gamma_i(1-\gamma_i)\mathbf{I} \right)$$

$$p_U^{\mathbf{v}}(\theta_i^{\mathbf{v}}|\theta_{i-1}; t_i) = \mathbb{E}_{\hat{\mathbf{y}}_i^{\mathbf{v}}\sim\mathcal{N}\left( \beta_i^{\mathbf{v}}\left[ K\left( \Phi(\theta_{i-1};t_i)^{\mathbf{v}} + \frac{1}{\lambda} \cdot \frac{\partial\Phi_V(\theta_{i-1};t_i)}{\partial\theta_{i-1}^{\mathbf{v}}} \right) - 1 \right], \beta_i^{\mathbf{v}}K\mathbf{I} \right)} \delta(\theta_i^{\mathbf{v}} - \text{Softmax}(\hat{\mathbf{y}}_i^{\mathbf{v}}))$$

$$\tag{19}$$

For detailed justification of gradient guidance in BFN sampling, we refer readers to Qiu et al. (2025).

**Formulation of RL in the Context of BFN** Recalling Sec. 3, we can accommodate BFN within the context of RL in Tab. 4.

# B  PROOFS FOR PROPOSITIONS

## B.1  PROOF FOR OPTIMALITY OF GRADIENT GUIDANCE

**Lemma 1.** *The transitional kernel in Eq. 4 is a first-order approximation of the following:*

$$p_T^V(\mathbf{s}_{i+1}|\mathbf{s}_i, \mathbf{a}_i) = \frac{p_T(\mathbf{s}_{i+1}|\mathbf{s}_i, \mathbf{a}_i)\exp(V_{\pi_{ref}}(\mathbf{s}_{i+1})/\lambda)}{\exp(V_{\pi_{ref}}(\mathbf{s}_i)/\lambda)} \tag{20}$$

*Proof.* We begin by applying a first-order Taylor expansion to $\exp(V_{\pi_{ref}}(\mathbf{s}_{i+1}))$ around $\mathbf{s}_i$:

$$\exp(V_{\pi_{ref}}(\mathbf{s}_{i+1})/\lambda) \approx \exp(V_{\pi_{ref}}(\mathbf{s}_i)/\lambda) \cdot \exp(\nabla V_{\pi_{ref}}(\mathbf{s}_i) \cdot \Delta\mathbf{s}_i/\lambda). \tag{21}$$

Recall in Eq. 4 that $p_T^V(\mathbf{s}_{i+1}|\mathbf{s}_i, \mathbf{a}_i) \overset{\text{apx}}{\propto} p_T(\mathbf{s}_{i+1}|\mathbf{s}_i, \mathbf{a}_i)\exp(-E(\mathbf{s}_i)/\lambda)\exp(\Delta\mathbf{s}_i \nabla_{\mathbf{s}_i}\Phi_V(\mathbf{s}_i; t_i)/\lambda)$. If $\Phi_V$ learns to exactly predict $V_{\pi_{ref}}$, we can substitute the gradient term with Eq. 21, which leads to $p_T^V(\mathbf{s}_{i+1}|\mathbf{s}_i, \mathbf{a}_i) = p_T(\mathbf{s}_{i+1}|\mathbf{s}_i, \mathbf{a}_i)\exp(V_{\pi_{ref}}(\mathbf{s}_{i+1})/\lambda)/Z$.

Next, we calculate the denominator $Z$ using the soft Bellman equation (Haarnoja et al., 2017):

$$
\begin{aligned}
Z &= \underset{\mathbf{s}_{i+1}\sim p_T(\cdot|\mathbf{s}_i,\mathbf{a}_i)}{\mathbb{E}}[\exp(V_{\pi_{ref}}(\mathbf{s}_{i+1})/\lambda)] \\
&= \underset{\mathbf{s}_{i+1}\sim p_T(\cdot|\mathbf{s}_i,\mathbf{a}_i)}{\mathbb{E}}\left[\underset{\mathbf{m}\sim\pi_{ref}(\cdot|\mathbf{s}_{i+1})}{\mathbb{E}}[\exp(r(\mathbf{m})/\lambda)]\right] \\
&= \underset{\mathbf{m}\sim\pi_{ref}(\cdot|\mathbf{s}_i)}{\mathbb{E}}[\exp(r(\mathbf{m})/\lambda)] \\
&= \exp(V_{\pi_{ref}}(\mathbf{s}_i)/\lambda),
\end{aligned}
\tag{22}
$$

which proves the lemma. □

**Proposition 1.** *If $p_I$ is a Dirac distribution, then using the transition kernel $p_T^V$ in Eq. 4 to perform sampling in Eq. 3 is equivalent to sampling from the optimal policy $\pi^*$ in Eq. 2. Furthermore, the normalizing factor $\lambda$ coincides with the hyperparameter $\alpha$.*

*Proof.* Assuming $p_I(\mathbf{s}_0) = \delta(\mathbf{s}_0 - \hat{\mathbf{s}}_0)$, we obtain $V_{\pi_{ref}}(\hat{\mathbf{s}}_0) = \mathbb{E}_{\mathbf{m}\sim\pi_{ref}}\exp(r(\mathbf{m})/\lambda)$ by definition. We denote the marginal distribution along the sampling trajectory with gradient guidance as $\pi_{GG}(\mathbf{m})$, and calculate it by induction in Eq. 3:

$$
\begin{aligned}
\pi_{GG}(\mathbf{m}) &= \underset{\mathbf{s}_0\sim p_I}{\mathbb{E}}[\pi_{GG}(\mathbf{m}|\mathbf{s}_0)] = \pi_{GG}(\mathbf{m}|\hat{\mathbf{s}}_0) \\
&= \underset{\mathbf{s}_1\sim p_T^V(\cdot|\hat{\mathbf{s}}_0,\mathbf{a}_0)}{\mathbb{E}}[\pi_{GG}(\mathbf{m}|\mathbf{s}_1)] \\
&= \underset{\mathbf{s}_1\sim p_T(\cdot|\hat{\mathbf{s}}_0,\mathbf{a}_0)}{\mathbb{E}}\left[\frac{\exp(V_{\pi_{ref}}(\mathbf{s}_1)/\lambda)}{\exp(V_{\pi_{ref}}(\hat{\mathbf{s}}_0)/\lambda)} \cdot \pi_{GG}(\mathbf{m}|\mathbf{s}_1)\right] \\
&= \underset{\mathbf{s}_1\sim p_T(\cdot|\hat{\mathbf{s}}_0,\mathbf{a}_0)}{\mathbb{E}}\left[\frac{\exp(V_{\pi_{ref}}(\mathbf{s}_1)/\lambda)}{\exp(V_{\pi_{ref}}(\hat{\mathbf{s}}_0)/\lambda)} \cdot \underset{\mathbf{s}_2\sim p_T(\cdot|\mathbf{s}_1,\mathbf{a}_1)}{\mathbb{E}}\left[\frac{\exp(V_{\pi_{ref}}(\mathbf{s}_2)/\lambda)}{\exp(V_{\pi_{ref}}(\mathbf{s}_1)/\lambda)} \cdot \pi_{GG}(\mathbf{m}|\mathbf{s}_2)\right]\right] \\
&= \underset{\mathbf{s}_1,\mathbf{s}_2\sim p_T}{\mathbb{E}}\left[\frac{\exp(V_{\pi_{ref}}(\mathbf{s}_2)/\lambda)}{\exp(V_{\pi_{ref}}(\hat{\mathbf{s}}_0)/\lambda)} \cdot \pi_{GG}(\mathbf{m}|\mathbf{s}_2)\right] \\
&= \underset{\mathbf{s}_1,\mathbf{s}_2\cdots,\mathbf{s}_N\sim p_T}{\mathbb{E}}\left[\frac{\exp(V_{\pi_{ref}}(\mathbf{s}_N)/\lambda)}{\exp(V_{\pi_{ref}}(\hat{\mathbf{s}}_0)/\lambda)} \cdot \pi_{GG}(\mathbf{m}|\mathbf{s}_N)\right] \quad \text{(applying induction until } \mathbf{s}_N) \\
&= \underset{\mathbf{s}_1,\mathbf{s}_2\cdots,\mathbf{s}_N\sim p_T}{\mathbb{E}}\left[\frac{\exp(r(\mathbf{m})/\lambda)}{\exp(V_{\pi_{ref}}(\hat{\mathbf{s}}_0)/\lambda)} \cdot p_O(\mathbf{m}|\mathbf{s}_N)\right] \\
&= \frac{\exp(r(\mathbf{m})/\lambda)}{\exp(V_{\pi_{ref}}(\hat{\mathbf{s}}_0)/\lambda)} \underset{\mathbf{s}_1,\mathbf{s}_2\cdots,\mathbf{s}_N\sim p_T}{\mathbb{E}}[p_O(\mathbf{m}|\mathbf{s}_N)] \\
&= \frac{\exp(r(\mathbf{m})/\lambda)\pi_{ref}(\mathbf{m})}{\mathbb{E}_{\mathbf{m}\sim\pi_{ref}}\exp(r(\mathbf{m})/\lambda)},
\end{aligned}
\tag{23}
$$

where substituting $\lambda$ with $\alpha$ yields Eq. 2. □

Notably, BFNs initialize $\mathbf{s}_0$ as a constant value, indicating that $p_I$ is indeed a Dirac distribution. For diffusion and flow matching models that rely on Gaussian priors, we have the following corollary:

**Corollary 1.** *If $p'_I(\mathbf{s}_0) \propto p_I(\mathbf{s}_0) \exp(V_{\pi_{ref}}(\mathbf{s}_0))/\lambda$, then sampling from $p'_I$ with the transition kernel $p^V_T$ is equivalent to sampling from the optimal policy $\pi^*$.*

The corollary can be proved as follows:

$$
\begin{aligned}
\pi_{\mathrm{GG}}(\mathbf{m}) &= \mathop{\mathbb{E}}_{\mathbf{s}_0 \sim p'_I} [\pi_{\mathrm{GG}}(\mathbf{m}|\mathbf{s}_0)] \\
&= \mathop{\mathbb{E}}_{\mathbf{s}_0 \sim p_I} \left[ \frac{\exp(V_{\pi_{\mathrm{ref}}}(\mathbf{s}_0))/\lambda}{\mathbb{E}_{\mathbf{s}_0 \sim p_I}[\exp(V_{\pi_{\mathrm{ref}}}(\mathbf{s}_0))/\lambda]} \cdot \pi_{\mathrm{GG}}(\mathbf{m}|\mathbf{s}_0) \right] \\
&= \frac{1}{\mathbb{E}_{\mathbf{m} \sim \pi_{\mathrm{ref}}} \exp(r(\mathbf{m})/\lambda)} \cdot \mathop{\mathbb{E}}_{\mathbf{s}_0 \sim p_I} [\exp(V_{\pi_{\mathrm{ref}}}(\mathbf{s}_0))/\lambda \cdot \pi_{\mathrm{GG}}(\mathbf{m}|\mathbf{s}_0)] \\
&= \frac{\exp(r(\mathbf{m})/\lambda)\pi_{\mathrm{ref}}(\mathbf{m})}{\mathbb{E}_{\mathbf{m} \sim \pi_{\mathrm{ref}}} \exp(r(\mathbf{m})/\lambda)} \quad \text{(Applying induction in Eq. 23)}
\end{aligned}
\tag{24}
$$

In practice, we can use $\Phi_V(\mathbf{s}_0; t = 0)$ to approximate $V_{\pi_{\mathrm{ref}}}(\mathbf{s}_0)$ and sample $\mathbf{s}_0 \sim p'_I$ by importance sampling.

### B.2 PROOF FOR OPTIMALITY OF GRADIENT-FREE GUIDANCE

**Assumption 1.** *Given a sufficiently large $R$, the likelihood of observing a reward $R$ under the reference policy from state $s_i$ is given by:*

$$
p(R|\mathbf{s}_i) = \exp[-(R - V_{\pi_{ref}}(\mathbf{s}_i))/\tau_i] \propto \exp(V_{\pi_{ref}}(\mathbf{s}_i)/\tau_i),
\tag{25}
$$

*where $\tau_i$ is a constant.*

For notational convenience, we define the induced transition distribution of a policy $\Phi$ as $\hat{\pi}_\Phi(\mathbf{s}_{i+1}|\mathbf{s}_i) = \mathbb{E}_{\mathbf{a}_i \sim \Phi(\cdot|,\mathbf{s}_i, t_i)}[p_T(\mathbf{s}_{i+1}|\mathbf{s}_i, \mathbf{a}_i)]$. By Bayes' rule, the optimal conditional distribution satisfies:

$$
\hat{\pi}_{\mathrm{opt}}(\mathbf{s}_{i+1}|R, \mathbf{s}_i) \propto \hat{\pi}_{\mathrm{ref}}(\mathbf{s}_{i+1}|\mathbf{s}_i) p(R|\mathbf{s}_{i+1})
\tag{26}
$$

The training objective of the conditional model $\Phi_P$ is to minimize the KL-divergence between $\hat{\pi}_{\mathrm{opt}}$ and $\hat{\pi}_{\Phi_P}$, and at convergence we have $\hat{\pi}_{\Phi_P} = \hat{\pi}_{\mathrm{opt}}$.

Recalling gradient-free guidance in Eq. 7, *i.e.*, $\tilde{\mathbf{a}}_i = w \cdot \Phi_P(\mathbf{s}_i|R; t_i) + (1 - w) \cdot \Phi_{\mathrm{ref}}(\mathbf{s}_i; t_i)$, we follow the assumptions in Ho & Salimans (2021) for the guided transition probability with a guidance strength of $w$ as the following:

$$
\hat{\pi}^w_{\Phi_P}(\mathbf{s}_{i+1}|R, \mathbf{s}_i) \propto \hat{\pi}_{\Phi_P}(\mathbf{s}_{i+1}|R, \mathbf{s}_i)^w \hat{\pi}_{\mathrm{ref}}(\mathbf{s}_{i+1}|\mathbf{s}_i)^{1-w}
\tag{27}
$$

Combining Eq. 25, Eq. 26 and Eq. 27 we obtain:

$$
\begin{aligned}
\hat{\pi}^w_{\Phi_P}(\mathbf{s}_{i+1}|R, \mathbf{s}_i) &= \hat{\pi}_{\mathrm{ref}}(\mathbf{s}_{i+1}|\mathbf{s}_i) \exp(w \cdot V_{\pi_{\mathrm{ref}}}(\mathbf{s}_{i+1})/\tau_i)/Z', \\
Z' &= \mathop{\mathbb{E}}_{\mathbf{s}_{i+1} \sim \hat{\pi}_{\mathrm{ref}}(\cdot|\mathbf{s}_i)} \exp(w \cdot V_{\pi_{\mathrm{ref}}}(\mathbf{s}_{i+1})/\tau_i).
\end{aligned}
\tag{28}
$$

**Lemma 2.** *The following inequality holds for any monotonic increasing function $f : \mathbb{R} \to \mathbb{R}$ and $w > 0$:*

$$
\mathop{\mathbb{E}}_{\mathbf{s}_{i+1} \sim \hat{\pi}^w_{\Phi_P}(\cdot|\mathbf{s}_i)} f(V_{\pi_{ref}}(\mathbf{s}_{i+1})) \geq \mathop{\mathbb{E}}_{\mathbf{s}_{i+1} \sim \hat{\pi}_{ref}(\cdot|\mathbf{s}_i)} f(V_{\pi_{ref}}(\mathbf{s}_{i+1})),
\tag{29}
$$

*Proof.* We first define $g(\mathbf{s}_{i+1}) = \exp(w \cdot V_{\pi_{\text{ref}}}(\mathbf{s}_{i+1})/\tau_i)/Z'$ which is a monotonic increasing function over $V_{\pi_{\text{ref}}}$ satisfying $\mathbb{E}_{\mathbf{s}_{i+1} \sim \pi_{\text{ref}}(\cdot|\mathbf{s}_i)} g(\mathbf{s}_{i+1}) = 1$. Then, we have:

$$
\mathbb{E}_{\mathbf{s}_{i+1} \sim \hat{\pi}_{\Phi_P}^w(\cdot|\mathbf{s}_i)} f(V_{\pi_{\text{ref}}}(\mathbf{s}_{i+1})) - \mathbb{E}_{\mathbf{s}_{i+1} \sim \hat{\pi}_{\text{ref}}(\cdot|\mathbf{s}_i)} f(V_{\pi_{\text{ref}}}(\mathbf{s}_{i+1}))
$$

$$
= \mathbb{E}_{\mathbf{s}_{i+1} \sim \hat{\pi}_{\text{ref}}(\cdot|\mathbf{s}_i)} [g(\mathbf{s}_{i+1})f(V_{\pi_{\text{ref}}}(\mathbf{s}_{i+1}))] - \mathbb{E}_{\mathbf{s}_{i+1} \sim \hat{\pi}_{\text{ref}}(\cdot|\mathbf{s}_i)} f(V_{\pi_{\text{ref}}}(\mathbf{s}_{i+1})) \quad \text{(Plugging Eq. 28)}
$$

$$
= \mathbb{E}_{\mathbf{s}_{i+1} \sim \hat{\pi}_{\text{ref}}(\cdot|\mathbf{s}_i)} [f(V_{\pi_{\text{ref}}}(\mathbf{s}_{i+1}))(g(\mathbf{s}_{i+1}) - 1)]
$$

$$
= \mathbb{E}_{\mathbf{s}_{i+1} \sim \hat{\pi}_{\text{ref}}(\cdot|\mathbf{s}_i)} \left[ f(V_{\pi_{\text{ref}}}(\mathbf{s}_{i+1})) \left( g(\mathbf{s}_{i+1}) - \mathbb{E}_{\mathbf{s}'_{i+1} \sim \hat{\pi}_{\text{ref}}(\cdot|\mathbf{s}_i)} g(\mathbf{s}'_{i+1}) \right) \right] \tag{30}
$$

$$
= \frac{1}{2} \mathbb{E}_{\mathbf{s}_{i+1}, \mathbf{s}'_{i+1} \sim \hat{\pi}_{\text{ref}}(\cdot|\mathbf{s}_i)} \left[ 2f(V_{\pi_{\text{ref}}}(\mathbf{s}_{i+1}))g(V_{\pi_{\text{ref}}}(\mathbf{s}_{i+1})) - 2f(V_{\pi_{\text{ref}}}(\mathbf{s}_{i+1}))g(V_{\pi_{\text{ref}}}(\mathbf{s}'_{i+1})) \right]
$$

$$
= \frac{1}{2} \mathbb{E}_{\mathbf{s}_{i+1}, \mathbf{s}'_{i+1} \sim \hat{\pi}_{\text{ref}}(\cdot|\mathbf{s}_i)} \left[ (f(V_{\pi_{\text{ref}}}(\mathbf{s}_{i+1})) - f(V_{\pi_{\text{ref}}}(\mathbf{s}'_{i+1})))(g(V_{\pi_{\text{ref}}}(\mathbf{s}_{i+1})) - g(V_{\pi_{\text{ref}}}(\mathbf{s}'_{i+1}))) \right]
$$

$$
\geq 0, \quad \text{(both } f \text{ and } g \text{ are monotonic increasing functions over } V_{\pi_{\text{ref}}})
$$

which proves the inequality. $\qquad \square$

**Proposition 3.** *Using $\tilde{\mathbf{a}}_i$ in Eq. 7 to perform sampling in Eq. 3 is an improved policy over $\pi_{ref}$, and the guidance weight $w$ balances reward maximization and adherence to the reference policy.*

*Proof.* Formally, $\pi_{\Phi_P}^w$ is an improved policy over $\pi_{\text{ref}}$ if:

$$
\mathbb{E}_{\mathbf{m} \sim \pi_{\Phi_P}^w(\cdot|\mathbf{s}_0)}[\exp(r(\mathbf{m})/\lambda)] \geq \mathbb{E}_{\mathbf{m} \sim \pi_{\text{ref}}(\cdot|\mathbf{s}_0)}[\exp(r(\mathbf{m})/\lambda)], \tag{31}
$$

holds for arbitrary $s_0 \in \mathcal{S}$.

To prove the inequality, we first define a monotonic increasing function $f(x) = \exp(x/\lambda)$ in Eq. 29, which gives:

$$
\mathbb{E}_{\mathbf{s}_{i+1} \sim \hat{\pi}_{\Phi_P}^w(\cdot|\mathbf{s}_i)} \exp(V_{\pi_{\text{ref}}}(\mathbf{s}_{i+1})/\lambda) \geq \mathbb{E}_{\mathbf{s}_{i+1} \sim \hat{\pi}_{\text{ref}}(\cdot|\mathbf{s}_i)} \exp(V_{\pi_{\text{ref}}}(\mathbf{s}_{i+1})/\lambda) = \exp(V_{\pi_{\text{ref}}}(\mathbf{s}_i)/\lambda), \tag{32}
$$

using the soft Bellman Equation in Eq. 22.

Suppose $p_O$ is a deterministic sampler that chooses the atom type based on the maximum predicted probability, we directly have $\mathbb{E}_{\mathbf{m} \sim \pi_{\Phi_P}^w(\cdot|\mathbf{s}_N)}[\exp(r(\mathbf{m})/\lambda)] = \mathbb{E}_{\mathbf{m} \sim \pi_{\text{ref}}(\cdot|\mathbf{s}_N)}[\exp(r(\mathbf{m})/\lambda)]$. Then, assume that $\mathbb{E}_{\mathbf{m} \sim \pi_{\Phi_P}^w(\cdot|\mathbf{s}_i)}[\exp(r(\mathbf{m})/\lambda)] \geq \mathbb{E}_{\mathbf{m} \sim \pi_{\text{ref}}(\cdot|\mathbf{s}_i)}[\exp(r(\mathbf{m})/\lambda)]$ holds for arbitrary $i \in \{K, K+1, \cdots, N\}$, we have the following:

$$
\mathbb{E}_{\mathbf{m} \sim \pi_{\Phi_P}^w(\cdot|\mathbf{s}_{K-1})}[\exp(r(\mathbf{m})/\lambda)] = \mathbb{E}_{\mathbf{m} \sim \pi_{\Phi_P}^w(\cdot|\mathbf{s}_K), \mathbf{s}_K \sim \hat{\pi}_{\Phi_P}^w(\cdot|\mathbf{s}_{K-1})}[\exp(r(\mathbf{m})/\lambda)]
$$

$$
\geq \mathbb{E}_{\mathbf{m} \sim \pi_{\text{ref}}(\cdot|\mathbf{s}_K), \mathbf{s}_K \sim \hat{\pi}_{\Phi_P}^w(\cdot|\mathbf{s}_{K-1})}[\exp(r(\mathbf{m})/\lambda)]
$$

$$
= \mathbb{E}_{\mathbf{s}_K \sim \hat{\pi}_{\Phi_P}^w(\cdot|\mathbf{s}_{K-1})}[\exp(V_{\pi_{\text{ref}}}(\mathbf{s}_K))/\lambda)] \tag{33}
$$

$$
\geq \exp(V_{\pi_{\text{ref}}}(\mathbf{s}_{K-1})/\lambda) \quad \text{(Plugging Eq. 32)}
$$

$$
= \mathbb{E}_{\mathbf{m} \sim \pi_{\text{ref}}(\cdot|\mathbf{s}_{K-1})}[\exp(r(\mathbf{m})/\lambda)],
$$

which proves Eq. 31 with mathematical induction.

In addition, Eq. 28 reveals that a larger $w$ places more probability mass on high-value states, while a smaller $w$ encourages behaviors closer to the reference policy. Thus, $w$ provides a natural choice to balance reward maximization and adherence to $\pi_{\text{ref}}$. $\qquad \square$

---

**Algorithm 1** Iterative Reinforcement Learning

---

**Require:** Reference dataset $\mathcal{D}$, number of rounds $R$, training steps $M$, number of samples $N_K$, filtering function $F$, reward function $r$, noise parameters $\sigma_1, \beta_1$, sampling hyperparameters $\mathbf{h}$

1: $\mathcal{D}^{(0)} \leftarrow \mathcal{D}$
2: **for** $i = 0$ to $R$ **do**
3:     **if** $i > 0$ **then**
4:         $\Phi_P^{(i)} \leftarrow \Phi_P^{(i-1)}, \Phi_V^{(i)} \leftarrow \Phi_V^{(i-1)}$
5:     **end if**
6:     **for** Step $= 0$ to $M$ **do**
7:         $(\mathbf{m}, \mathbf{p}, R) \sim \mathcal{D}^{(i)}, j \sim U(1, \hat{N}), t = \frac{j-1}{\hat{N}}$
8:         $\mu \sim p_F^{\mathbf{x}}(\mu | \mathbf{x}; \beta^{\mathbf{x}}(t) = \sigma_1^{-2t} - 1), \theta^{\mathbf{v}} = p_F^{\mathbf{v}}(\theta^{\mathbf{v}} | \mathbf{v}; \beta^{\mathbf{v}}(t) = t^2 \beta_1)$
9:         $\hat{V} \leftarrow \Phi_V^{(i)}(\mu, \theta^{\mathbf{v}}, \mathbf{p}, t), (\hat{\mathbf{x}}, \hat{\mathbf{v}}) = \Phi_P^{(i)}(\mu, \theta^{\mathbf{v}}, \mathbf{p}, t, R)$
10:         $\mathcal{L}_V \leftarrow |\hat{V} - R|, \mathcal{L}_P \leftarrow \mathcal{L}_c + \mathcal{L}_d$ in Eq. 13 and Eq. 15
11:         $\Phi_V^{(i)} \leftarrow \text{Optimize}(\Phi_V^{(i)}, \mathcal{L}_V), \Phi_P^{(i)} \leftarrow \text{Optimize}(\Phi_P^{(i)}, \mathcal{L}_P)$
12:     **end for**
13:     $\mathcal{D}^{(i+1)} \leftarrow \mathcal{D}^{(i)}$
14:     **for** $(\mathbf{m}, \mathbf{p}, r) \in \mathcal{D}$ **do**
15:         **for** $k = 1$ to $N_K$ **do**
16:             $\hat{\mathbf{m}} = (\hat{\mathbf{x}}, \hat{\mathbf{v}}) \sim \text{Sample}(\mathbf{p}, \Phi_V^{(i)}, \Phi_P^{(i)}, \Phi_{\text{ref}}, \mathbf{h})$
17:             **if** $F(\hat{\mathbf{m}}) = 1$ **then**
18:                 $\mathcal{D}^{(i+1)} \leftarrow \mathcal{D}^{(i+1)} \cup \{(\hat{\mathbf{m}}, \mathbf{p}, r(\hat{\mathbf{m}}))\}$
19:             **end if**
20:         **end for**
21:     **end for**
22: **end for**
23: Return $\Phi_V^{(R)}, \Phi_P^{(R)}$

---

## C   Implementation Details

**Featurization** For each protein atom, the feature vector $\mathbf{v}_i^{\mathcal{P}}$ involves a one-hot encoding of the atom type (H, C, N, O, S, Se), a 20-dimensional one-hot encoding of the amino acid type, and a binary indicator specifying whether the atom belongs to the protein backbone. For each ligand atom, the input feature $\mathbf{v}_i^{\mathcal{M}}$ is a 13-dimensional one-hot vector indicating the atom type (H, C, N, O, F, P, S, Cl) and if the atom is located within an aromatic ring. For the reward-conditioning model $\Phi_P$, we concatenate the ligand atom feature with two extra dimensions corresponding to the normalized values for Vina Score and SA.

**Model Architecture** Following Qu et al. (2024), we adopt PosNet3D (Guan et al., 2023b), an SE(3)-equivariant network as the backbone for $\Phi_V$ and $\Phi_P$. At the $l$-th layer, a protein-ligand graph $\mathcal{G} = (\mathcal{V}, \mathcal{E})$ is constructed on the fly by connecting each atom to its k-nearest neighbors. For each edge $(i, j) \in \mathcal{E}$, the edge features $\mathbf{e}_{ij}$ consist of the outer products of the distance embedding obtained by expanding the Euclidean distance between two atoms using radial basis functions, and a 4-dimensional one-hot vector indicating the edge type (protein-protein, protein-ligand, ligand-protein, or ligand-ligand). Then, the atom features $\mathbf{h}^{(l)}$ and coordinates $\mathbf{x}^{(l)}$ are updated as follows via message-passing:

$$
\begin{aligned}
\mathbf{h}_i^{(l+1)} &= \mathbf{h}_i^{(l)} + \sum_{j \in \mathcal{N}(i)} f_\phi \left( d_{ij}, \mathbf{h}_i^{(l)}, \mathbf{h}_j^{(l)}, \mathbf{e}_{ij}, t \right), \\
\mathbf{x}_i^{(l+1)} &= \mathbf{x}_i^{(l)} + \sum_{j \in \mathcal{N}(i)} (\mathbf{x}_i^{(l)} - \mathbf{x}_j^{(l)}) g_\varphi \left( d_{ij}, \mathbf{h}_i^{(l+1)}, \mathbf{h}_j^{(l+1)}, \mathbf{e}_{ij}, t \right) \text{ for ligand atoms,}
\end{aligned}
\tag{34}
$$

where $\mathcal{N}(i)$ is the set of neighboring nodes of $i$ in $\mathcal{G}$, $d_{ij} = \|\mathbf{x}_i^{(l)} - \mathbf{x}_j^{(l)}\|_2$ is the Euclidian distance between atoms $i$ and $j$, and $f_\phi, g_\varphi$ are attention blocks that use $\mathbf{h}_i^{(l)}$ as queries and $[d_{ij}, \mathbf{h}_i^{(l)}, \mathbf{h}_j^{(l)}, \mathbf{e}_{ij}, t]$ as keys and values. For the first layer, the inputs are defined as $\mathbf{x}^{(0)} = [\mu, \mathbf{x}^{\mathcal{P}}]$,

$\mathbf{h}^{(0)} = \text{Linear}([\theta^{\mathbf{v}}, \mathbf{v}^{\mathcal{P}}, t])$. For the value model $\Phi_V$, the scalar output is obtained by averaging the outputs from a two-layer MLP, *i.e.*, $V = \frac{1}{N_{\mathcal{M}}} \sum_{i=1}^{N_{\mathcal{M}}} \text{MLP}(\mathbf{h}_i^{(L)})$. For the policy model $\Phi_P$, the predicted coordinates are directly taken from $\mathbf{x}^{(L)}$, while the atom types are predicted by a linear classifier $\hat{\mathbf{v}} = \text{SoftMax}(\text{Linear}(\mathbf{h}^{(L)}))$.

**Reward Formulation** Following Qiu et al. (2025), the reward $r(\mathbf{m})$ is formulated as a linear combination of normalized Vina Score and SA as follows:

$$r(\mathbf{m}) = -\frac{\text{Clip}(\text{Vina\_Score}(\mathbf{m}), -16, -1) + 1}{15} + \frac{\text{SA}(\mathbf{m}) - 0.17}{0.83}, \tag{35}$$

where $\text{Clip}(\cdot, \cdot, \cdot)$ is the clipping operation.

**Training** At the $i$-th iteration, we sample molecules and pockets within the dataset $\mathcal{D}_i$ with equal probability and apply $p_F$ to generate noisy inputs. The value and policy models $\Phi_V^{(i)}$ and $\Phi_P^{(i)}$ are initialized by the checkpoints obtained in the previous iteration. Then, we use the mean absolute error (MAE) loss $\mathcal{L}_{\text{MAE}} = |\Phi_V^{(i)}(\theta_t) - r(\mathbf{m})|$ to optimize $\Phi_V^{(i)}$ and use $\mathcal{L}_c + \mathcal{L}_d$ from Eq. 13 and Eq. 15 to optimize $\Phi_P^{(i)}$. After training, we sample $N_K$ candidates per training pocket and filter duplicates whose Tanimoto similarity of Morgan fingerprints is greater than 0.6 with previous samples. Then, we remove incomplete and PB-invalid (Buttenschoen et al., 2024) molecules, calculate the reward scores for the remaining molecules, and add them to the training set $\mathcal{D}_{i+1}$. We repeat this process for 2 rounds of RL iterations. A detailed procedure is presented in Algorithm 1.

---

**Algorithm 2** Sampling with Mixed Guidance and Noise Reduction

---

**Require:** Pocket $\mathbf{p}$, value model $\Phi_V$, conditional policy model $\Phi_P$, reference model $\Phi_{\text{ref}}$, sampling steps $N$, atom feature dimension $K$, switch point $\tau$, guidance weights $\lambda, w$, noise parameters $\sigma_1, \beta_1$, noise reduction factor $\epsilon$, number of optimization objectives $d$

1: $\mu_0 \leftarrow \mathbf{0}, \theta_0^{\mathbf{v}} \leftarrow [1/K]_{N_{\mathcal{M}} \times K}, r \leftarrow [1]_d$
2: **for** $i = 0$ to $N - 1$ **do**
3:      $t \leftarrow \frac{i}{N}, \gamma(t) \leftarrow 1 - \sigma_1^{2t}, \beta(t) \leftarrow \beta_1 t^2$
4:      $(\hat{\mathbf{x}}, \hat{\mathbf{v}}) \leftarrow \Phi_P(\mu_i, \theta_i^{\mathbf{v}}, \mathbf{p}, r, t)$
5:      **if** $i \leq \lfloor \tau N \rfloor$ **then**
6:          $(\hat{\mathbf{x}}_{\text{ref}}, \hat{\mathbf{v}}_{\text{ref}}) \leftarrow \Phi_{\text{ref}}(\mu_i, \theta_i^{\mathbf{v}}, \mathbf{p}, r, t)$
7:          $\hat{\mathbf{x}} \leftarrow w\hat{\mathbf{x}} + (1-w)\hat{\mathbf{x}}_{\text{ref}}, \hat{\mathbf{v}} \leftarrow \text{SoftMax}(w\hat{\mathbf{v}} + (1-w)\hat{\mathbf{v}}_{\text{ref}})$
8:          $\mu_{i+1} \sim \mathcal{N}(\gamma(t)\hat{\mathbf{x}}, \epsilon^2\gamma(t)(1-\gamma(t))\mathbf{I})$
9:          $\mathbf{y}^{\mathbf{v}} \sim \mathcal{N}(\beta(t)(K\hat{\mathbf{v}} - 1), \epsilon^2\beta(t)K\mathbf{I})$
10:          $\theta_{i+1}^{\mathbf{v}} \leftarrow \text{SoftMax}(\mathbf{y}^{\mathbf{v}})$
11:      **else**
12:          $(\mathbf{g}_\mu, \mathbf{g}_{\mathbf{v}}) \leftarrow \left( -\frac{1}{d}\sum_{j=1}^d \left( \frac{\partial \Phi_V(\mu_i, \theta_i^{\mathbf{v}}, \mathbf{p}, t)}{\partial \mu_i} \right)_j, -\frac{1}{d}\sum_{j=1}^d \left( \frac{\partial \Phi_V(\mu_i, \theta_i^{\mathbf{v}}, \mathbf{p}, t)}{\partial \theta_i^{\mathbf{v}}} \right)_j \right)$
13:          $\mu_{i+1} \sim \mathcal{N}(\gamma(t)(\hat{\mathbf{x}} + \mathbf{g}_\mu/\lambda), \sigma^2\gamma(t)(1-\gamma(t))\mathbf{I})$
14:          $\mathbf{y}^{\mathbf{v}} \sim \mathcal{N}(\beta(t)(K\hat{\mathbf{v}} - 1 + K\mathbf{g}_{\mathbf{v}}/\lambda), \sigma^2\beta(t)K\mathbf{I})$
15:          $\theta_{i+1}^{\mathbf{v}} \leftarrow \text{SoftMax}(\mathbf{y}^{\mathbf{v}})$
16:      **end if**
17: **end for**
18: $t \leftarrow 1, (\hat{\mathbf{x}}, \hat{\mathbf{v}}) \leftarrow \Phi_P(\mu_{N-1}, \theta_{N-1}^{\mathbf{v}}, \mathbf{p}, r, t)$
19: $\hat{\mathbf{v}} \leftarrow \text{argmax}(\hat{\mathbf{v}})$
20: Return $\hat{\mathbf{x}}, \hat{\mathbf{v}}$

---

**Sampling** We set the prior belief $\theta_0$ as $\mu_0 = \mathbf{0}$ and $\theta_0^{\mathbf{v}} = [1/K]_{N_{\mathcal{M}} \times K}$ and perform $N$ sampling steps. For the first $M = \lfloor \tau N \rfloor$ steps, we calculate the predictions from both the policy model $\Phi_P$ and the reference model $\Phi_{\text{ref}}$, the latter initialized from the official checkpoint of Mol-CRAFT (Qu et al., 2024). The estimated molecule is given by $\hat{\mathbf{m}}_i = (w\Phi_P(\theta_{i-1}; t_i)^{\mathbf{x}} + (1-w)\Phi_P(\theta_{i-1}; t_i)^{\mathbf{x}}, \text{SoftMax}(w\Phi_P(\theta_{i-1}; t_i)^{\mathbf{v}} + (1-w)\Phi_P(\theta_{i-1}; t_i)^{\mathbf{v}}))$, and the posterior belief is updated using Eq. 18. For the remaining $N - M$ steps, we calculate the gradients of $\Phi_V$ and combine them with the predictions from $\Phi_P$ to perform a Bayesian update using Eq. 19. The full procedure is summarized in Algorithm 2.

**Hyperparameters** Both $\Phi_V$ and $\Phi_P$ are comprised of 9 layers with a hidden dimension of 128, 16 attention heads, ReLU activation, and layer normalization. In constructing the protein–ligand graph, each atom is connected to its 32 nearest neighbors. For training, we follow Qu et al. (2024) and set the noise schedule as $\sigma_1 = 0.03$ for continuous variables and $\beta_1 = 1.5$ for discrete variables. The number of discrete steps used for training is $\hat{N} = 1000$. For sampling, we perform $N = 100$ steps. In each iteration, we sample $N_K = 8$ candidates per training pocket and train $\Phi_V$ and $\Phi_P$ with 500K steps using the Adam optimizer (Kingma & Ba, 2014) with a learning rate of 0.005 and batch size of 8. Training the policy and value models takes around 7.5 days on a single NVIDIA A800 GPU. The sampling procedure for data enrichment takes ~20 days on 4 NVIDIA A800 GPUs, and the postprocessing requires 6.5 days on a server with 32 CPU cores.

## D EXPERIMENT DETAILS

**Baselines** We adopt the following baselines for comparison:

- **AR** (Luo et al., 2021) is a 3D autoregressive model that constructs ligands atom by atom based on predicted voxel-level atomic density maps.

- **Pocket2Mol** (Peng et al., 2022) employs an E(3)-equivariant network that iteratively selects a frontier atom and predicts its relative 3D position, element type, and bonding to existing atoms.

- **TargetDiff** (Guan et al., 2023a) is a multi-modal diffusion model that jointly perturbs and denoises both continuous atom coordinates and discrete atom types within the binding site.

- **DecompDiff** (Guan et al., 2023b) decomposes the molecule into scaffolds and arms and utilizes these chemical priors with validity constraints within a diffusion model to generate target-binding molecules.

- **MolCRAFT** (Qu et al., 2024) adopts Bayesian flow networks with a noise reduction strategy to perform multi-modal generation of atom coordinates and types.

- **RGA** (Fu et al., 2022) is a reinforcement learning framework that treats molecules as states and fragment exchanges and replacements as actions. It starts from molecules using 1D SMILES generative models and iteratively proposes exchanges and replacements with a trainable policy model. Candidate molecules are ranked by an oracle, and the top-$K$ are retained. Notably, RGA is designed to optimize search algorithms over the valid chemical space, whereas our RL4SBDD aims at directly performing preference alignment for generative models, with distinct MDP and RL formulations.

- **DecompOpt** (Zhou et al., 2024) develops a diffusion model conditioned on molecular fragments, which is similar to Guan et al. (2023b). Then, it applies an evolutionary process by extracting oracle-favored fragments and performing resampling based on these fragments.

- **CIDD** (Gao et al., 2025) optimizes generated molecules by leveraging large language models for interaction analysis, iterative design modifications, reflective feedback, and final candidate selection.

- **TagMol** (Dorna et al., 2024) is an alignment approach for TargetDiff (Guan et al., 2023a) that applies gradient guidance to atom coordinates during sampling.

- **MolJO** (Qiu et al., 2025) introduces joint gradient guidance on both continuous and discrete variables within the BFN framework.

- **AliDiff** (Gu et al., 2024) performs exact energy preference optimization, a regularized variant of direct preference optimization, to shift the sampling distribution of diffusion models toward low-energy, high-affinity states.

- **MolFORM** (Huang & Zhang, 2025) applies direct preference optimization on Vina Score to a flow matching model.

- **DecompDPO** (Cheng et al., 2024) fine-tunes DecompOpt (Zhou et al., 2024) by direct preference optimization using both molecular-level and fragment-level preference pairs and introduces physics-informed regularization to maintain conformational plausibility.

- **DiffGui** (Hu et al., 2025a) introduces a property guidance strategy on diffusion models that exhibits certain similarity to our gradient-free guidance. However, it is constrained on discrete

property labels $c \in \{0, 1, \emptyset\}$ while our RL4SBDD-M is conditioned on real-valued rewards $R \in \mathbb{R}$. Besides, we provide theoretical justifications for gradient-free guidance and additional training and sampling strategies.

**Additional Evaluation Metrics** In addition to the evaluation metrics reported in Sec. 5.1, we incorporate the following evaluation metrics:

- **Completeness**, defined as the proportion of molecules consisting of a single connecting component.

- **Strain Energy** (Harris et al., 2023), which quantifies the internal energy of the generation. A lower energy indicates that the pose is more stable.

- **PoseBusters Validity** (Buttenschoen et al., 2024), defined as the proportion of molecules that pass all 20 validity tests proposed by PoseBuster.

# E  ADDITIONAL EXPERIMENTS AND ANALYSIS

## E.1  ANALYSIS ON GRADIENT GUIDANCE AND GRADIENT-FREE GUIDANCE

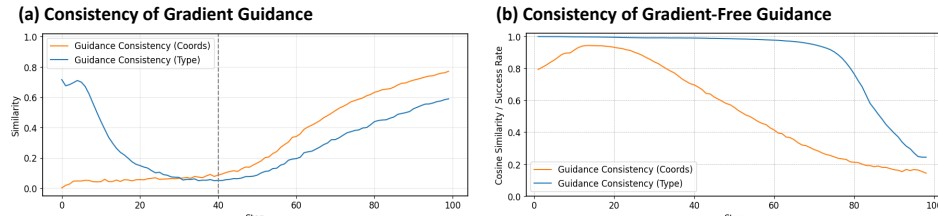

Figure 7: Analysis of guidance consistency for gradient guidance and gradient-free guidance.

**Gradient Guidance** In Fig. 4(a), we investigate the reliability of soft value estimation across different sampling steps. We first select 5 protein pockets and generate 64 sampling trajectories for each pocket using gradient guidance. For the $i$-th ($i = 10, 20, \cdots, 90$) step of each trajectory, we start from the current posterior belief $\theta_i$ and use the reference model to generate 64 molecules. We calculate the rewards of the molecules and report the average Pearson correlation between the average reward (64-pass Monte Carlo estimation) and the reward of a randomly selected molecule (1-pass Monte Carlo estimation) for the 320 trajectories. We also evaluate the effect of gradient guidance by removing it during the first $k$ steps ($k = 10, 20, \cdots, 90$) and reporting the resulting success rate of generated molecules. In addition, we evaluate guidance consistency measured by the cosine similarity of gradients between consecutive steps in Fig. 7(a):

$$C_{\text{GG}}^{(i)} = \left[ \text{CosSim}\left( \mathbf{g}_\mu^{(i)}, \mathbf{g}_\mu^{(i+1)} \right), \text{CosSim}\left( \mathbf{g}_\mathbf{v}^{(i)}, \mathbf{g}_\mathbf{v}^{(i+1)} \right) \right] \tag{36}$$

where $\mathbf{g}_\mu^{(i)}, \mathbf{g}_\mathbf{v}^{(i)}$ are the gradients on continuous and discrete variables.

As discussed in Sec. 3.2, the correlation between 1-pass and 64-pass MC estimates of the soft value function is near zero during the first 40 sampling steps, rises sharply between steps 40 to 70, and subsequently stabilizes around 1. As illustrated in Fig. 18, the BFN model first explores suitable binding modes in early generation steps, then refines the molecular structure in intermediate steps, and finally performs minor adjustments in the last steps. Therefore, estimating the value of early states is intractable due to the high noise in the inputs. As a result, removing gradient guidance has a negligible impact in the first 40 steps, but leads to significant performance drops in later steps.

Interestingly, the consistency of gradients for continuous coordinates increases along the sampling trajectory, whereas for discrete atom types it decreases during the first 40% of generation before rising again. We attribute this behavior to the early-stage shift of categorical probabilities toward the chemical prior, which encourages the selection of common atom types such as carbon, nitrogen, and oxygen.

**Gradient Free Guidance** In Fig. 4(b), we explore the effect of gradient-free guidance in different sampling steps. Specifically, we calculate the cosine similarity between the predictions of the

Table 5: Performance comparison with a larger average size. (↑)/(↓) indicates the higher/lower the better. SR is short for success rate. Avg. and Med. are short for average and median. The best and second best results are marked **bold** and underlined.

| Methods | Vina Dock (↓) | | QED (↑) | SA (↑) | Div (↑) | SR (↑) | Size |
| | Avg. | Med. | Avg. | Avg. | Avg. | Avg. | Avg. |
|---|---|---|---|---|---|---|---|
| Reference | -7.45 | -7.26 | 0.48 | 0.73 | - | 25.0% | 22.4 |
| DecompDiff | -8.39 | -8.43 | 0.45 | 0.61 | **0.68** | 24.5% | 29.4 |
| MolCRAFT | -9.25 | -9.20 | 0.46 | 0.62 | 0.61 | 36.1% | 29.4 |
| DecompOpt | -8.98 | -9.01 | 0.48 | 0.65 | 0.60 | 52.5% | 32.9 |
| MolJO | **-10.53** | -10.48 | 0.50 | 0.72 | 0.57 | 64.2% | 30.0 |
| DecompDPO | -9.90 | -10.08 | 0.48 | 0.60 | 0.61 | 52.1% | 32.9 |
| TacoGFN | -9.20 | -9.28 | 0.67 | 0.79 | 0.53 | 56.0% | 30.5 |
| 3DSynthFlow (high $\beta$) | -9.42 | -9.61 | **0.73** | - | - | - | - |
| RL4SBDD-M (Ours, $\epsilon = 0.2$) | **-10.53** | **-10.69** | 0.44 | **0.80** | 0.49 | **66.5%** | 29.8 |

reward-conditioned policy model $\Phi_P$ and the reference model $\Phi_{\text{ref}}$. Besides, we perform experiments in which $\Phi_{\text{ref}}$ is applied exclusively in the last $N - k$ steps ($k = 10, 20, \cdots, 90$) and report the resulting success rate of the generated molecules. To further assess guidance consistency, we compute the cosine similarity of the guidance vectors between consecutive steps in Fig. 7(b):

$$d_i^{\mathbf{x}} = \Phi_P(\mathbf{s}_i | R; t_i)^{\mathbf{x}} - \Phi_{\text{ref}}(\mathbf{s}_i)^{\mathbf{x}}, \quad d_i^{\mathbf{v}} = \Phi_P(\mathbf{s}_i | R; t_i)^{\mathbf{v}} - \Phi_{\text{ref}}(\mathbf{s}_i)^{\mathbf{v}}$$
$$C_{\text{GFG}}^{(i)} = \left[ \text{CosSim}\left(d_i^{\mathbf{x}}, d_{i+1}^{\mathbf{x}}\right), \text{CosSim}\left(d_i^{\mathbf{v}}, d_{i+1}^{\mathbf{v}}\right) \right]. \tag{37}$$

As discussed in Sec. 4.3, the predictions of $\Phi_P$ and $\Phi_{\text{ref}}$ converge as the generative process progresses, reflecting increased confidence in the parameter beliefs. However, the minor adjustments made in the late sampling steps are often inconsistent and fail to effectively exploit these states. Consequently, replacing $\Phi_P$ with $\Phi_{\text{ref}}$ during the final 60% steps has little impact on sampling performance, highlighting the limited utility of gradient-free guidance in late-stage generation. These findings motivate the development of our mixed guidance strategy.

### E.2 ADDITIONAL RESULTS

We report the strain energy and completeness for RL4SBDD-M and other baselines in Tab. 6. RL4SBDD-M achieves the lowest medium strain energy, indicating that the generated poses are conformationally stable. Moreover, RL4SBDD-M achieves a completeness of 95.2%, which is comparable to state-of-the-art models. Notably, enhancing binding affinity while preserving molecular connectivity remains a fundamental challenge, as over-optimization can lead to disconnected fragments that strongly bind to different regions of the pocket.

Besides, we visualize the distributions of Vina Score, Vina Min, Vina Dock, QED, and SA for RL4SBDD-M and other baselines in Figs. 8–12. Across all metrics, our model demonstrates consistent improvements over the reference model, underscoring its ability to balance binding quality with drug-likeness.

### E.3 ANALYSIS FOR SELF-RANKING

Inspired by Jumper et al. (2021), we introduce a self-ranking strategy that leverages the predicted scores of the value model $\Phi_V(\mathbf{s}_N; t = N)$ to select the best molecule among multiple generations. This setting mimics an application scenario where oracle evaluations are extremely expensive. As illustrated in Fig. 13, while self-ranking leads to moderate performance drops by prioritizing false positives, it shows a similar trend of test-time scaling as we generate more molecules.

### E.4 STRUCTURE-BASED MOLECULE OPTIMIZATION

**Task Definition** Recently, structure-based molecule optimization (SBMO) (Zhou et al., 2024; Qiu et al., 2025) has emerged as a more challenging extension of SBDD. Unlike de novo generation, SBMO focuses on modifying the reference molecule within the protein pocket to satisfy various

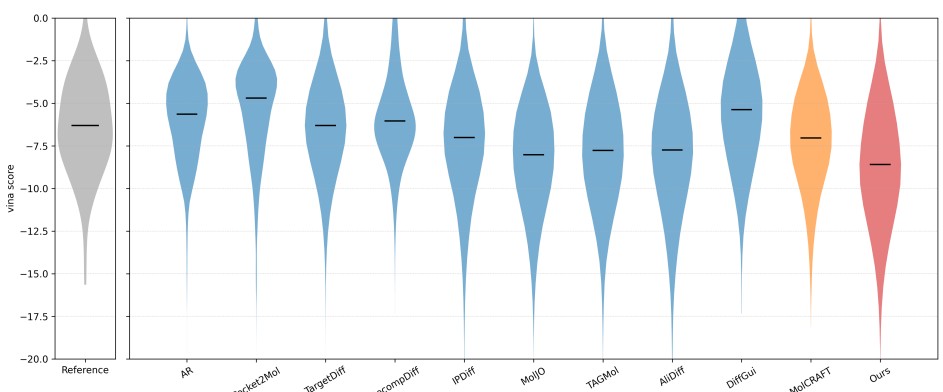

Figure 8: Violin plot of Vina Score for RL4SBDD-M and other baselines.

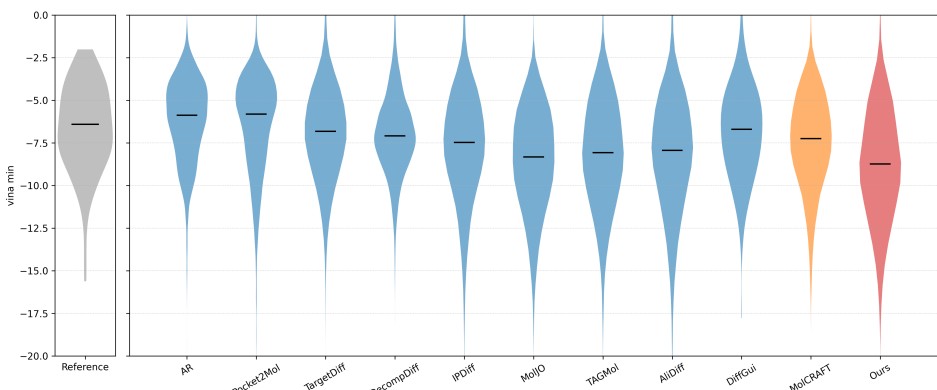

Figure 9: Violin plot of Vina Min for RL4SBDD-M and other baselines.

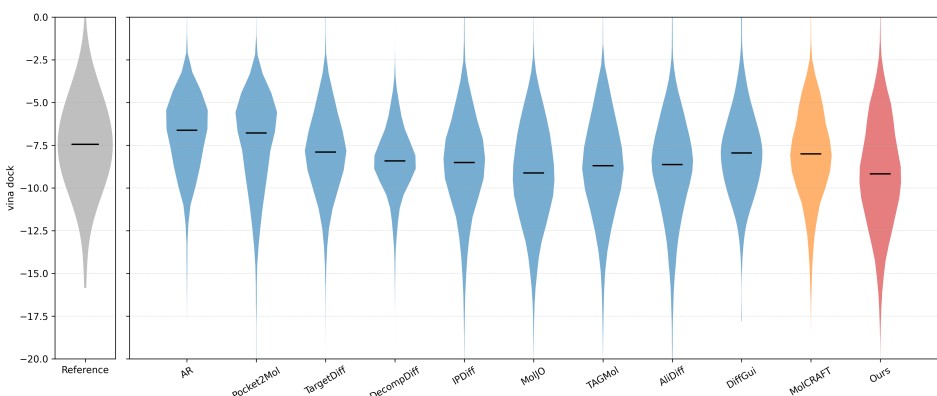

Figure 10: Violin plot of Vina Dock for RL4SBDD-M and other baselines.

alignment objectives. This task holds significant potential for real-world drug discovery, as it enables the design of *"me-better"* drugs. Following Qiu et al. (2025), we adopt three experiment settings for SBMO, described below:

- **R-group Optimization** This setting mimics the lead optimization process in drug discovery, where the Bemis-Murcko scaffold (Bemis & Murcko, 1996) of the reference molecule is fixed to preserve the binding mode, and the R-groups on side chains are substituted to improve binding affinity and drug-likeness. We follow Qiu et al. (2025) and constrain the number of atoms to match that of the reference molecule.

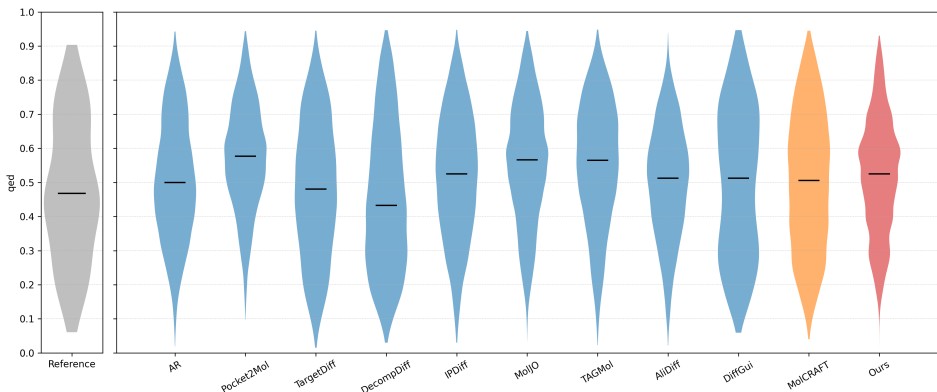

Figure 11: Violin plot of QED for RL4SBDD-M and other baselines.

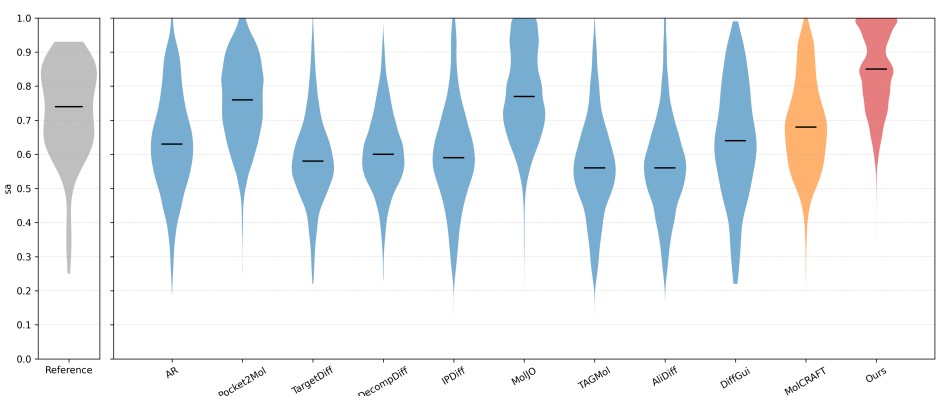

Figure 12: Violin plot of SA for RL4SBDD-M and other baselines.

Table 6: Performance comparison between RL4SBDD-M and baseline models for median strain energy, completeness, and PoseBusters validity. (↑)/(↓) indicates the higher/lower the better. The best results are marked **bold**.

| Model | Med. Strain Energy (↓) | Complete (↑) | PB-Valid (↑) |
|---|---|---|---|
| Reference | 114 | 100% | 95% |
| AR | 608 | 93.5% | 95.0% |
| Pocket2Mol | 186 | 96.3% | 59.0% |
| TargetDiff | 1208 | 90.4% | 50.5% |
| DecompDiff | 983 | 72.0% | 71.7% |
| MolCRAFT | 196 | 96.7% | 84.6% |
| RGA | - | 52.2% | - |
| DecompOpt | 861 | 2.6% | 48.0% |
| MolJO | 163 | **97.3%** | **87.1%** |
| RL4SBDD-M (Ours) | **125** | 95.2% | 85.1% |

- **Fragment Growing** This setting seeks to enlarge the molecule and enhance the desired properties. Similar to R-group optimization, the Bemis–Murcko scaffold of the reference molecule is fixed, but the model is allowed to generate additional atoms beyond the reference structure.

- **Scaffold Hopping** This setting reflects a widely used pharmaceutical strategy that maintains functional groups crucial for binding while redesigning the molecular scaffold to connect these groups and overcome limitations of the original structure. In our experiments, we fix the atoms outside the Bemis-Murcko scaffold of the reference molecule and require the model to generate new scaffold atoms.

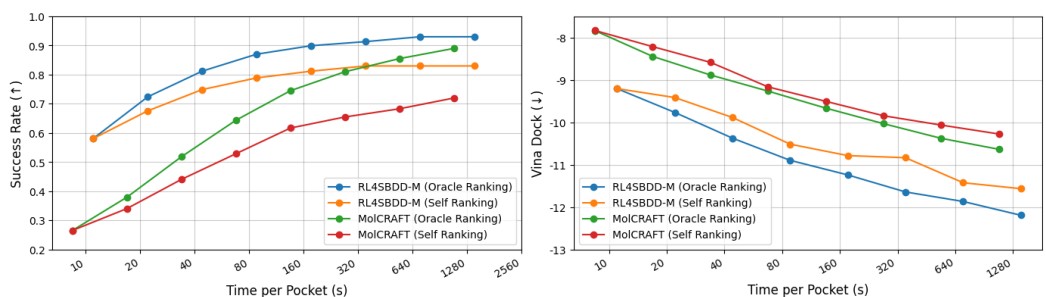

Figure 13: Inference scaling performance with self-ranking.

**Implementation Details** We perform experiments on the test set of CrossDocked2020, extract the Bemis-Murko scaffold of reference molecules using RDKit Landrum (2013), and sample 10 molecules per pocket for evaluation. For R-group optimization and fragment growing, we mask scaffold atoms in the reference molecule, while for fragment growing, we additionally introduce unmasked atoms to allow expansion. For scaffold hopping, we instead mask non-scaffold atoms. Sampling is performed directly without additional fine-tuning. At each sampling step, predicted coordinates and atom types for masked atoms are replaced with their original values from the reference molecule.

**Hyperparameters** In SBMO experiments, we set $1/\lambda = 30, w = 1.0$ for the R-group optimization and fragment growing, and $1/\lambda = 15, w = 1.0$ for scaffold hopping, as larger guidance weights are found to produce incomplete molecules. Other hyperparameters follow the SBDD settings.

**Results and Analysis** Tab. 7 compares RL4SBDD-M with baseline methods across different SBMO settings. Our model consistently achieves the best performance, improving the state-of-the-art method MolJO by 3.5%, 1.6%, and 4.2% absolute gains in success rate for R-group design, fragment growing, and scaffold hopping, respectively. We notice reduced connectivity of the generated molecules, especially for scaffold hopping. We attribute this to over-optimization, *i.e.*, our model tends to propose overly aggressive binding modes that cannot be realized given the limited number of available free atoms.

Table 7: Performance comparison of RL4SBDD-M and baselines for structure-based molecule optimization. *Redesign* is short for R-group optimization, *Growing* is short for fragment growing, and *Hopping* is short for scaffold hopping. (↑)/(↓) indicates the higher/lower the better. The best results are marked **bold**.

| Task | Methods | Vina Score (↓) Avg. | Med. | Vina Min (↓) Avg. | Med. | Vina Dock (↓) Avg. | Med. | QED (↑) Avg. | SA (↑) Avg. | Complete (↑) Avg. | SR (↑) % |
|---|---|---|---|---|---|---|---|---|---|---|---|
| - | Reference | -6.36 | -6.46 | -6.71 | -6.49 | -7.45 | -7.26 | 0.48 | 0.73 | 100% | 25.0% |
| Redesign | TargetDiff | -6.14 | -6.21 | -6.79 | -6.58 | -7.70 | -7.61 | 0.50 | 0.64 | 85.5% | 18.9% |
|  | TAGMol | -6.60 | -6.66 | -7.10 | -6.80 | -7.63 | -7.76 | 0.53 | 0.62 | 87.0% | 19.2% |
|  | MolCRAFT | -6.63 | -6.70 | -7.12 | -6.91 | -7.79 | -7.72 | 0.49 | 0.67 | **96.7%** | 22.7% |
|  | MolJO | -7.13 | -7.28 | -7.62 | -7.39 | -8.16 | **-8.20** | 0.57 | 0.68 | 95.1% | 29.0% |
|  | RL4SBDD-M(Ours) | **-7.22** | **-7.37** | **-7.64** | **-7.58** | **-8.30** | -8.19 | **0.60** | **0.69** | 90.0% | **32.5%** |
| Growing | TargetDiff | -6.73 | -7.29 | -7.60 | -7.67 | -8.89 | -8.79 | 0.39 | 0.52 | 71.6% | 11.2% |
|  | TAGMol | -7.30 | -7.70 | -8.08 | -7.81 | -8.92 | -8.78 | 0.47 | 0.53 | 78.7% | 11.8% |
|  | MolCRAFT | -6.96 | -7.47 | -7.86 | -7.73 | -8.80 | -8.65 | 0.44 | 0.59 | 91.7% | 19.9% |
|  | MolJO | -8.08 | -8.35 | -8.79 | -8.58 | -9.21 | -9.45 | 0.53 | 0.62 | **93.2%** | 32.7% |
|  | RL4SBDD-M(Ours) | **-8.70** | **-8.85** | **-9.12** | **-9.01** | **-9.63** | **-9.49** | **0.56** | **0.64** | 88.5% | **34.3%** |
| Hopping | TargetDiff | -5.72 | -5.78 | -6.00 | -5.83 | -6.31 | -6.66 | 0.39 | 0.65 | 63.3% | 6.2% |
|  | TAGMol | -6.17 | -6.10 | -6.46 | -6.07 | -7.19 | -6.80 | 0.44 | 0.62 | 68.7% | 6.9% |
|  | MolCRAFT | -6.31 | -6.17 | -6.58 | -6.40 | -7.25 | -7.15 | 0.42 | 0.67 | 89.9% | 14.6% |
|  | MolJO | -6.86 | -6.50 | -7.13 | -6.70 | -7.67 | -7.58 | 0.46 | 0.68 | **90.5%** | 23.6% |
|  | RL4SBDD-M(Ours) | **-7.03** | **-6.95** | **-7.19** | **-7.01** | **-7.79** | **-7.85** | **0.49** | **0.71** | 77.4% | **27.8%** |

## E.5 Ablation Studies

**Impacts of Switching Point** We investigate different choices of switching point $\tau$, as shown in Fig. 15. The results indicate that the optimal choice lies between 0.3 and 0.4, which corroborates our prior findings that gradient-free guidance brings more improvements in the early stages of genera-

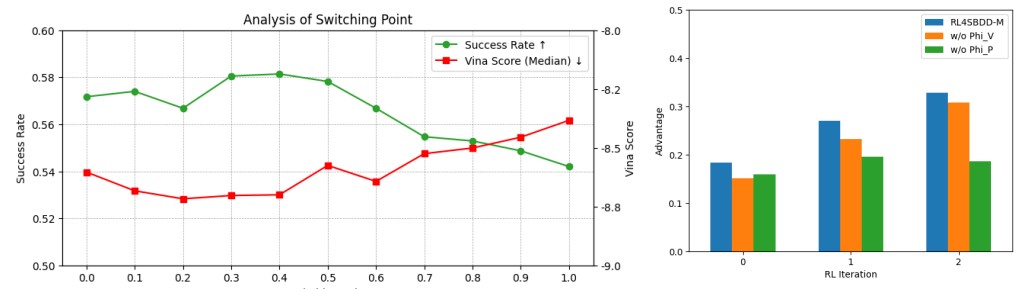

Figure 14: The success rate and Vina Score of different switching points $\tau$.

Figure 15: Reward improvement over the reference policy across RL iterations.

tion, while gradient guidance mostly benefits in later stages. Besides, using a smaller $\tau$ results in only a minor performance degradation, suggesting that the conditional policy model alone is capable of providing near-optimal actions even without gradient-free guidance.

**Impacts of Guidance Weight** We study the effects of guidance weights, $1/\lambda$ in gradient guidance and $w$ in gradient-free guidance. A large $1/\lambda$ or $w$ indicates that the sampling policy is more focused on reward maximization, while smaller values encourage stronger adherence to the reference model. To reduce computational cost, we sample 10 molecules for each pocket in the test set and report the metrics. As shown in Tab. 8, the optimal range is 20~30 for $1/\lambda$ and 1.0~1.2 for $w$, and excessively large guidance weights substantially reduce molecular completeness, suggesting over-optimization. Based on these observations, we choose $1/\lambda = 30$ and $w = 1.2$ in our main experiments. Notably, the best $w$ differs from prior design choices in image diffusion (Ho & Salimans, 2021). We attribute this discrepancy to the sensitivity of molecular design (Song et al., 2024), and the back correction step in (Qu et al., 2024) that amplifies this sensitivity by directly incorporating model predictions into Bayesian update steps.

Table 8: Analysis on different guidance scales. ($\uparrow$)/($\downarrow$) indicates the higher/lower the better. SR is short for success rate. Avg. and Med. are short for average and median. The best results are marked **bold**.

| $1/\lambda$ | $w$ | Vina Score ($\downarrow$) | | Vina Min ($\downarrow$) | | QED ($\uparrow$) | SA ($\uparrow$) | Complete ($\uparrow$) | SR ($\uparrow$) |
|---|---|---|---|---|---|---|---|---|---|
| | | Avg. | Med. | Avg. | Med. | Avg. | Avg. | Avg. | Avg. |
| 1 | 1.2 | -7.66 | -8.21 | -8.32 | -8.38 | 0.54 | 0.86 | **96.2%** | 53.2% |
| 5 | 1.2 | -7.47 | -8.36 | -8.25 | -8.51 | **0.55** | 0.86 | 96.1% | 53.5% |
| 10 | 1.2 | -7.61 | -8.43 | -8.48 | -8.57 | 0.54 | 0.86 | 95.1% | 56.4% |
| 20 | 1.2 | **-8.18** | -8.63 | -8.66 | **-8.90** | 0.53 | 0.84 | 94.7% | 58.0% |
| 30 | 1.2 | -7.97 | **-8.70** | **-8.67** | -8.87 | 0.54 | **0.85** | 95.2% | **58.1%** |
| 50 | 1.2 | -6.42 | -8.28 | -7.72 | -8.40 | 0.50 | 0.77 | 52.9% | 48.1% |
| 30 | 1.0 | -7.78 | -8.60 | -8.59 | -8.80 | 0.52 | **0.85** | **95.3%** | 57.2% |
| 30 | 1.2 | **-7.97** | **-8.70** | **-8.67** | **-8.87** | **0.54** | **0.85** | 95.2% | **58.1%** |
| 30 | 1.5 | -7.21 | -8.73 | -8.51 | -8.90 | 0.53 | 0.83 | 88.6% | 55.9% |
| 30 | 2.0 | -7.43 | -8.49 | -8.37 | -8.69 | 0.52 | 0.84 | 86.0% | 55.1% |
| 30 | 5.0 | -6.54 | -7.97 | -8.00 | -8.25 | 0.51 | 0.84 | 72.7% | 51.5% |

**Impacts of Molecular Size** Previous studies (Qu et al., 2024; Qiu et al., 2025) have shown that molecular size has a significant impact on molecular properties. Larger molecules can achieve higher binding affinity by forming more interactions with the binding pocket, but often at the cost of reduced drug-likeness and synthetic accessibility due to increased weight and structural complexity. For fair comparison, we sample the number of atoms from a prior distribution to ensure that the average atom count ($\sim 23.0$) matches that of reference molecules ($\sim 22.8$) in the main experiments. Moreover, we report the results with larger atom counts in Tab. 5, where RL4SBDD-M achieves the best success rate, highlighting its robustness in generating large out-of-distribution (OOD) molecules.

**Impacts of Conditioning Label** While the policy model $\Phi_P$ is primarily used to generate high-reward molecules conditioned on a sufficiently large $R$ during sampling, we investigate if the model could produce molecules whose reward scores align with different reward labels. Since normalized

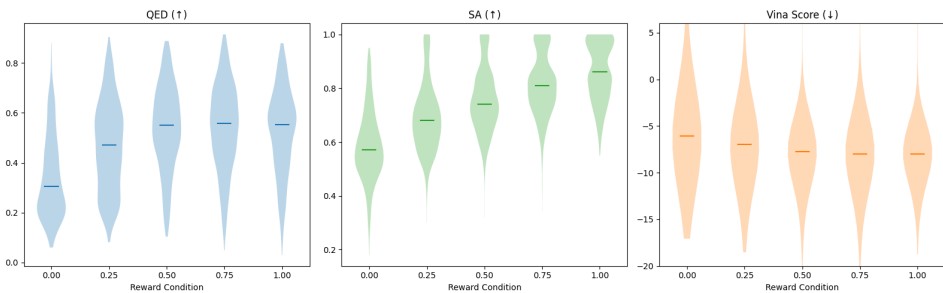

Figure 16: Violin plot of QED, SA, and Vina Score for the policy model with different reward conditions.

reward scores lies within [0, 1], we experiment with labels {0, 0.25, 0.5, 0.75, 1} and visualize the distribution of QED, SA, and Vina Score of $\Phi_P$'s generations after the second round of iterative RL. As illustrated in Fig. 16, all metrics improve consistently as the conditioning label increases up to 0.75, although a gap remains between the desired reward and the achieved average scores. Interestingly, when the conditioning label is set to 1, SA continues to improve beyond the 0.75 setting, whereas QED and Vina Score slightly degrade. We attribute this to an out-of-distribution (OOD) effect, *i.e.*, very few molecules achieve extremely high drug-likeness and binding affinity, making it difficult for the model to generalize reliably in this regime.

Table 9: Ring distribution for different models and molecular databases.

| Model | 3-ring | 4-ring | 5-ring | 6-ring | 7-ring | 8-ring |
|---|---|---|---|---|---|---|
| Ref | 0.04 | 0.00 | 0.73 | 1.63 | 0.02 | 0.00 |
| ChEMBL | 0.05 | 0.03 | 0.88 | 2.55 | 0.04 | 0.00 |
| MolCRAFT | 0.00 | 0.01 | 0.65 | 1.94 | 0.16 | 0.02 |
| MolJO | 0.00 | 0.00 | 0.45 | 3.24 | 0.18 | 0.01 |
| MolCRAFT+DPO | | | | | | |
| RL4SBDD-M (SA+Affinity) | 0.00 | 0.00 | 0.20 | 3.04 | 0.32 | 0.00 |
| RL4SBDD-M (SA+Affinity+LogP) | 0.00 | 0.01 | 0.54 | 2.65 | 0.13 | 0.01 |
| RL4SBDD-M (SA+Affinity, KL regularization) | 0.00 | 0.01 | 0.53 | 2.74 | 0.28 | 0.02 |

Table 10: Protein-ligand interactions profiles and molecular properties for different models.

| Model | HBD | HBA | $\pi$-$\pi$ | $\pi$-cation | Salt Bridges | Vina | SA | LogP |
|---|---|---|---|---|---|---|---|---|
| Ref | 2.44 | 1.09 | 0.27 | 0.08 | 0.55 | -6.46 | 0.73 | 0.89 |
| MolCRAFT | 2.68 | 1.21 | 0.34 | 0.08 | 0.55 | -6.95 | 0.67 | 1.09 |
| MolCRAFT+DPO | | | | | | | | |
| MolJO | 1.17 | 0.48 | 0.74 | 0.14 | 0.17 | -8.02 | 0.78 | 4.20 |
| RL4SBDD-M (SA+Affinity) | 0.75 | 0.18 | 0.86 | 0.12 | 0.09 | -8.70 | 0.85 | 5.48 |
| RL4SBDD-M (SA+Affinity+LogP) | 2.14 | 0.74 | 0.60 | 0.11 | 0.35 | -7.88 | 0.76 | 3.08 |
| RL4SBDD-M (SA+Affinity, KL regularization) | 1.74 | 0.41 | 0.59 | 0.09 | 0.21 | -8.38 | 0.77 | 4.30 |

### E.6 ANALYSIS OF REWARD OVER-OPTIMIZATION

While RL4SBDD-M achieves outstanding performance in optimizing a given reward, it remains unclear whether enhancing Vina scores and SA scores lead to undesirable molecular patterns in real-world drug discovery. To this end, we calculate the ring distribution of RL4SBDD generations as well as several baseline models in Tab. 9 and compare them against ChEMBL (Gaulton et al., 2017), a drug-like molecular database. We also analyze the non-covalent interactions between the generated ligands and the target protein in Tab. 10 using PLIP (Salentin et al., 2015).

In comparison to the reference molecules and the MolCRAFT model that learns the reference distribution, all preference alignment methods show a similar prevalence of introducing more 6-

membered rings (mostly aromatic rings) to formulate more $\pi$-$\pi$ stackings and $\pi$-cation interactions, which are tighter than other non-covalent interactions. Consequently, we notice a significant drop in the number of 5-membered rings, fewer hydrophobic interactions and salt bridges, and an overly high lipophilicity that may violate Lipinski's Rule of Five (Pollastri, 2010) and lead to undesirable ADME properties. As suggested by Skalse et al. (2022); Karwowski et al. (2024), such a phenomenon stems from the imperfect design of the reward and the intrinsic nature of molecules. This is evidenced by a negative correlation between Vina Score and the number of 6-membered rings (Spearman $\rho$=-0.55), and a positive correlation between LogP and the number of 6-membered rings (Spearman $\rho$=0.63).

Fortunately, RL4SBDD is a general framework built upon soft RL that can be applied to arbitrary reward functions. Based on it, we develop two strategies to control the excessive prevalence of 6-membered rings, detailed as follows:

- **Reward Redesign**. Specifically, we incorporate the water-octanol partition coefficient (LogP) into the reward objective. In our experiments, we calculate LogP of the molecules within the dataset at the 2nd round RL with RDKit (Landrum, 2013), clip the value within range [-3,7], normalize it into [0,1], and consider LogP=1 as the most desirable (Arnott & Planey, 2012). For gradient guidance, the reward is calculated as $|\text{LogP}_{\text{norm}} - 0.4|$. For reward conditioning, we set the conditioning label $\text{LogP}_{\text{norm}}$ as 0.4. As shown in Tab. 9, the number of aromatic rings is effectively controlled and reduced to an acceptable range when compared with ChEMBL. While the average LogP does not exactly match the desired value, this is potentially due to the overall distribution of our enriched dataset via iterative RL that solely optimizes the Vina score and SA score.

- **KL Regularization**. Since reference molecules typically originate from biologically plausible candidates, reducing guidance weights ($1/\lambda$ for gradient guidance and $w$ for reward conditioning) can alleviate over-optimization of the reward and encourage stronger adherence to the reference policy. As shown in Tab. 9, when we reduce these weights by half ($1/\lambda = 15, w = 0.5$), the prevalence of 6-membered rings is effectively mitigated.

Overall, developing genuinely desirable molecules in practical SBDD remains a fundamental challenge that requires joint advances in reward formulation, multi-objective optimization, and robust evaluation protocols. Our contribution is a theoretical unification of existing approaches that reveals shared limitations and provides insights into reward maximization, along with a simple and effective methodology that improves multi-objective optimization. As stated in Sec. 6, we leave the exploration of more reward objectives and realistic validation for future works.

## F  MORE VISUALIZATIONS

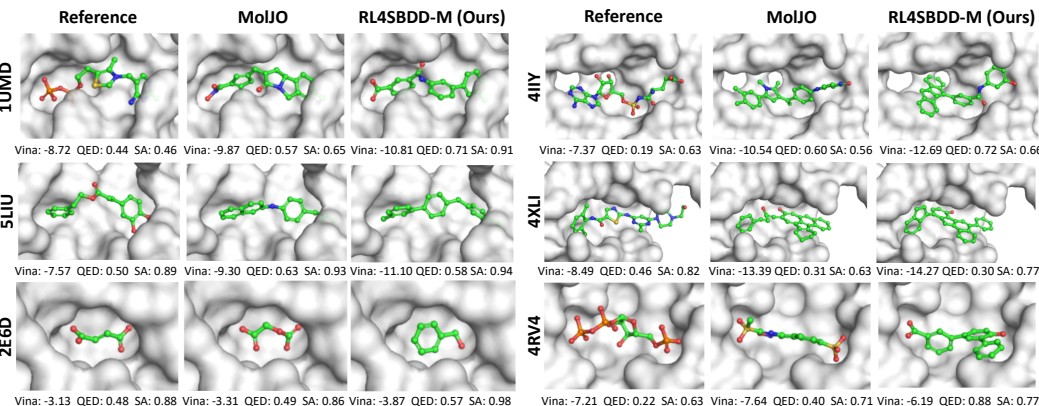

Figure 17: **Visualization of the reference molecules and generations from MolJO and RL4SBDD-M** on protein 1UMD, 4IIY, 5LIU, 4XLI, 2E6D, and 4RV4.

We provide additional visualizations of reference and generated molecules for 6 test pockets in Fig. 17. We observe that our model consistently generates pocket binders with high-affinity and favorable molecular properties across diverse molecular and pocket sizes. However, we note a limitation

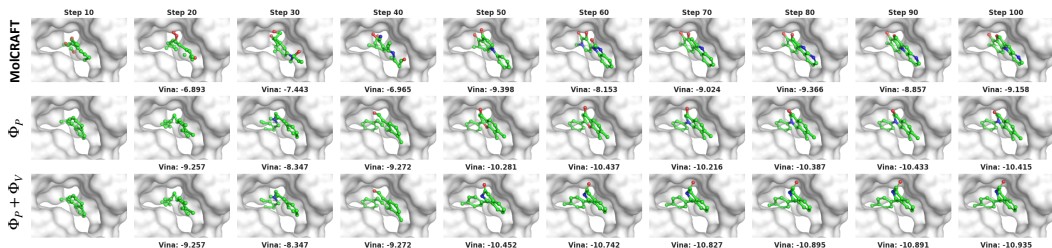

Figure 18: **Visualization of the estimated molecules with the sampling trajectory** on protein 2V3R. We perform sampling with MolCRAFT, the conditional policy model $\Phi_P$, and RL4SBDD-M that incorporates gradient guidance in late generation steps.

on 4RV4, where our model falls short in binding affinity but improves QED and SA significantly, underscoring the challenge of balancing multiple objectives.

In addition, we visualize the sampling trajectory in Fig. 18 by showing the predicted molecules at different sampling steps. The first row depicts the sampling process of MolCRAFT (Qu et al., 2024) that constructs the molecule in a coarse-to-grain manner. The second row shows the trajectory of the reward-conditioned policy model, where the predicted pose shifts deeper into the pocket in the early sampling steps, resulting in tighter binding. In the third row, we follow the second-row trajectory for the first 40% steps and apply gradient guidance in the remaining steps to assess its contribution. Notably, the oxygen atom attached to the pyridine ring is repositioned toward the upper-right region of the pocket, forming a stronger hydrogen bond and further improving affinity. These results validate the complementarity between $\Phi_P$, which defines a desirable global structure in early generation, and $\Phi_V$, which provides fine-grained refinements in later stages.

## G   MORE RELATED WORKS

**Search-based Models for Structure-based Drug Design** Recently, several research works propose to search for desirable molecules within the valid chemical space with evolutionary algorithms. Specifically, they start from the outputs of generative models, iteratively propose new candidates, and preserve the best molecules according to an oracle function. For example, RGA (Fu et al., 2022) defines cross-over and mutation operations based on chemical heuristics and develops an RL-based policy network to propose these operations. DecompOpt (Zhou et al., 2024) proposes new candidates by generating molecules conditioned on fragments favored by the oracle to enhance exploration efficiency. CIDD (Gao et al., 2025) leverages large language models (LLMs) to suggest molecular modifications. EvoSBDD (Reidenbach, 2024) directly optimizes the prior noise of generative models with black-box optimization to obtain improved molecular proposals. While these approaches have achieved promising results, they rely heavily on oracle feedbacks and lack direct preference alignment to generative models. In contrast, we show that a properly aligned model can benefit from oracle-based screening and outperform search-based methods with repeated sampling.

**Reinforcement Learning for Molecule Design** Reinforcement learning (RL) has become a widely adopted paradigm for optimizing non-differentiable objectives in goal-directed molecular design. Most existing works treat valid molecules as states and structural modifications as actions, and generate molecules without explicitly considering the pocket structure. Molecular structures are represented as molecular fingerprints (Zhou et al., 2019), 1D SMILES strings (Blaschke et al., 2020; Loeffler et al., 2024), 2D graphs (You et al., 2018; Yang et al., 2021), or even transformed into natural language (Zholus et al., 2025; Hu et al., 2025b) and are optimized using policy gradients (Silver et al., 2014), Q-learning (Watkins & Dayan, 1992), or PPO (Schulman et al., 2017b) algorithms. While these approaches could be adapted to structure-based drug design using docking scores as rewards, they typically require separate training for each pocket on the fly, and the learned policies cannot be transferred to novel pockets.

# H    THE USE OF LARGE LANGUAGE MODELS (LLMs)

Large language models (LLMs) are used exclusively to polish grammar, refine phrasing, and improve overall coherence in this work. They do not contribute to the formulation of scientific ideas, data analysis, or any other aspects of the research process.

