# OpenReview forum: "RL4SBDD: Reinforcement Learning for Preference Alignment in Structure-based Drug Design"
_ICLR.cc/2026/Conference — Submitted to ICLR 2026_

### Official Review · Reviewer_3QpG · 2025-10-15

**Soundness:** 2
**Presentation:** 3
**Contribution:** 1
**Rating:** 2
**Confidence:** 4

**Summary:**

While my primary expertise is not in the specific generative methodologies employed (diffusion models, Bayesian flow networks), this review is based on extensive experience in the domain of AI-driven drug discovery and molecular design, coupled with a deep understanding of reinforcement learning for molecular optimization.

This paper introduces RL4SBDD, a framework utilizing advanced reinforcement learning (RL) to shift the sample distribution, p(x∣target), of distribution learning-based structure-based drug design (SBDD) generative models. The central aim is to guide the generation process toward molecules that optimize a given reward function, such as binding affinity, thereby exploring chemical space beyond the distribution of the original training data.

A noteworthy aspect of this work is its claim to improve docking scores while maintaining molecular sizes comparable to those of reference compounds. This approach commendably attempts to avoid the common pitfall of artificially inflating atom counts to exploit scoring functions like AutoDock Vina, representing a more principled approach to validation.

Nevertheless, the all generation examples raise significant doubts about the practical utility of the proposed method and reliability of evaluation. **The high prevalence of aromatic groups in the generated molecules** indicates a form of reward hacking, **reminiscent to how early RL methods achieved SOTA on optimization benchmarks by generating simple carbon chains**. This concern is substantiated by the fact that the reported chemical diversity of the generated molecules is remarkably lower than that of the baseline methods, casting doubt on their effectiveness and real-world applicability.

Methodologically, the paper combines existing RL techniques to a new domain rather than proposing a fundamentally new algorithm. In the context of AI4Science, such an application can be a valuable contribution. However, in these cases, I believe the evaluation criteria should shift from pure algorithmic novelty to how effectively the method addresses critical requirements defined by domain experts. From this standpoint, the validation presented in the manuscript does not sufficiently demonstrate that proposed method produces molecules with the desirable properties for a practical drug discovery campaign.

---
**Usage of LLM**: I wrote the entire review myself and only used the LLM to correct the grammar and improve readability.

**Strengths:**

1. RL4SBDD achieves the noticeable performance improvement in benchmark table by combining advanced RL methods.
2. The authors conduct a detailed analysis of how various parameters (guidance and noise level, etc) affect performance.
3. The motivation and key differences from existing methodologies are well visualized.

**Weaknesses:**

**1. Insufficient Baseline Comparisons**

While RL4SBDD conducts online training, its performance was not compared to pure RL-based SBDD methods with online training, such as TacoGFN [1] and 3DSynthFlow [2]. In particular, 3DSynthFlow has demonstrated state-of-the-art (SOTA) Vina docking scores and QED ensuring high synthesizability, while maintaining diversity comparable to distribution-learning based models and generating molecules 5x faster than MolCRAFT.

**2. Low Chemical Diversity**

The reported diversity of generated molecules is markedly low. Given that non-flexible docking is known to have a high false-positive rate [3,4], generating a chemically diverse set of candidates is essential for increasing the probability of finding viable hit compounds [5,6].
Though authors show that it is possible to control diversity and potency of generated molecules, they do not report the performance of high-diversity setting in Benchmark Table 1.

**3. Concern about reward hacking**

Figures 5 and 15 reveals that the protein-ligand interaction patterns of generated molecules from RL4SBDD-M differ substantially from the reference ligands, indicating the high probability to be false-positive.
**Specifically, the generated molecules exhibit a high prevalence of aromatic rings, even for hydrophilic pockets (e.g., 4RV4).**
This is in stark contrast to MolCRAFT, which report a balanced number of H-Bond donors/acceptors and hydrophobic groups, even though RL4SBDD-M shares the same architecture to MolCRAFT.

This observation leads to three critical concerns:
- 3.1: Reward hacking by increasing aromatic rings.
- 3.2: If the model over-samples aromatic rings, which are structurally simple, the RMSD < 2Å metric becomes less meaningful. It is much easier to predict the structure of planar benzene ring than other druggable functional groups.
- 3.3: Highly hydrophobic molecules with numerous aromatic rings often have poor ADME properties, limiting their real-world application.

To substantiate the model's performance and allay concerns of reward hacking, please consider the following additional analyses:
- A comparative analysis of physicochemical property distributions (e.g., logP, TPSA, number of HBD/HBA/aromatic rings) between the generated molecules, reference molecules, and a drug-like molecular database like ChEMBL.
- An analysis of protein-ligand interaction profiles using a tool like PoseCheck [7], with a comparison to other baselines.
- Calculation of the Delta Score [8] to verify that the model is optimizing for genuine protein-ligand interactions rather than hacking Vina scoring function.

**4. Ambiguity in Multi-Objective Optimization**

Figure 2 presents three molecular properties (Vina score, QED, SA score), yet the reward function defined in Equation 34 incorporates only the Vina and SA scores. This discrepancy is potentially misleading.

In drug discovery, the ability of an RL method to balance multiple objectives is a critical determinant of its utility. The authors are encouraged to report performance using a reward function that simultaneously optimizes all three properties, following previous optimization methods like TacoGFN [1] and MOOD [9].

---
**References**
1. Shen, Tony, et al. "TacoGFN: target-conditioned GFlowNet for structure-based drug design." TMLR (2024).
2. Shen, Tony, et al. "Compositional Flows for 3D Molecule and Synthesis Pathway Co-design." ICML (2025).
3. Passaro, Saro, et al. "Boltz-2: Towards accurate and efficient binding affinity prediction." BioRxiv (2025): 2025-06.
4. Furui, Kairi, and Masahito Ohue. "Boltzina: Efficient and Accurate Virtual Screening via Docking-Guided Binding Prediction with Boltz-2." arXiv preprint arXiv:2508.17555 (2025).
5. Lyu, Jiankun, et al. "Ultra-large library docking for discovering new chemotypes." Nature 566.7743 (2019): 224-229.
6. Bengio, Emmanuel, et al. "Flow network based generative models for non-iterative diverse candidate generation." Advances in neural information processing systems 34 (2021): 27381-27394.
7. Harris, Charles, et al. "Benchmarking generated poses: How rational is structure-based drug design with generative models?." arXiv preprint arXiv:2308.07413 (2023).
8. Gao, Bowen, et al. "Rethinking specificity in sbdd: Leveraging delta score and energy-guided diffusion." arXiv preprint arXiv:2403.12987 (2024).
9. Lee, Seul, Jaehyeong Jo, and Sung Ju Hwang. "Exploring chemical space with score-based out-of-distribution generation." ICML (2023)

**Questions:**

1. In the abstract, the authors use the term "desirable molecules." Could the authors please clarify the specific criteria used to define "desirability" in this context? In drug discovery, a molecule is typically considered desirable only if it satisfies a complex profile of properties, including potency, selectivity, and favorable physicochemical characteristics (ADME). The manuscript currently seems to equate desirability primarily with the **success rate**, and a clarification would be beneficial for the reader.

2. The paper describes performing two iterative RL steps. To better illustrate the learning process and the efficacy of this iterative approach, would it be possible to include a plot showing the evolution of key performance metrics (e.g., average reward, Vina score) across the RL iterations? This would provide valuable insight into the online training.

---

> ### Author Response · Authors · 2025-11-20
> **Response by Authors (Part 1)**
>
> We thank the reviewer for carefully examining our work and providing constructive feedback from the perspective of practical utility and domain expertise. We are pleased to hear that you recognize our control over molecular sizes as *“noteworthy and commendable”*, our methodology as *“a valuable contribution”*, and our experiments as demonstrating *“noticeable performance improvement”* and *“a detailed analysis”*. We hope that our responses and additional analyses address your concerns satisfactorily.
>
> > W4.1 & Q1: Ambiguity in the definition of "desirability" in the context of SBDD and multi-objective optimization
>
> We would like to first address this concern to clarify our major contributions and avoid potential misinterpretations. We agree that, in practical scenarios, a desirable molecule should satisfy a wide range of properties, including potency, selectivity, synthesizability, and ADME characteristics, as noted in Lines 135–137 of the original submission. However, the primary contribution of our work lies in **the investigation of how to maximize a given reward by combining advanced RL strategies and the sampling dynamics of generative models in SBDD**, rather than developing novel rewards and validations that best reflect if the generated molecule is practically desirable. Therefore, the intention of Figure 2 is to show that **RL4SBDD is a general framework that can be applied to arbitrary verifiable rewards, where Vina Score, QED, and SA are the most widely-adopted proxies for several optimization objectives in SBDD**. We use the "..." expression in the figure to indicate that the reward can incorporate other important molecular properties, such as ADME and selectivity.
>
> In our main experiments, we show RL4SBDD-M's superiority in reward maximization by jointly optimizing affinity and SA scores. **The choice of this reward formulation, as well as all evaluation settings, follows prior published works such as DecompOpt [1] and MolJO [2] to ensure fair comparison**. While we use Success Rate as the main metric to evaluate optimization over the Affinity + SA reward, we acknowledge that achieving a fully “desirable” molecule requires satisfying a broader set of properties.
>
> To remove ambiguity, we have adjusted the abstract (Lines 21 and 25-30), introduction (Lines 40-42, 67-68, and 93-99), experiments (Lines 370-374) and relevant illustrations in our revised manuscript to clearly distinguish between (1) designing a general methodology capable of optimizing arbitrary verifiable rewards, and (2) evaluating the method on specific reward functions for empirical validation.
>
> > W1: Comparisons with TacoGFN and 3DSynthFlow
>
> Thanks for reminding us of these advanced online-RL models in SBDD. The performance comparison is as follows (- indicates that the result is not reported in the original paper):
>
> | Model                      | Vina Dock (↓) (Med./Avg.) | QED (↑) (Avg.) | SA (↑) (Avg.) | Atom Count (Avg.) |
> | -------------------------- | ------------------------- | -------------- | ------------- | ----------------- |
> | TacoGFN                    | -8.22/-8.44               | 0.67           | 0.79          | 30.5              |
> | 3DSynthFlow (high $\beta$) | -9.42/-9.61               | **0.73**       | -             | -                 |
> | RL4SBDD-M (Reference size) | -9.20/-9.28               | 0.54           | 0.85          | 23.0              |
> | RL4SBDD-M (Large size)     | **-10.53/-10.69**         | 0.44           | **0.80**      | 29.8              |
>
> **Compared to TacoGFN [3], RL4SBDD-M achieves higher binding affinities and SA scores**, which are the optimization goals in our reward formulation. It also achieves improved Vina Dock over 3DSynthflow [4]. Notably, the average number of atoms for the generated molecules in 3DSynthFlow is not reported in the original paper, which limits the fairness of direct, head-to-head comparisons. Moreover, RL4SBDD-M is trained on reference-size molecules, and its performance could be constrained by when generalizing to out-of-distribution (larger) molecular sizes. We have incorporated these comparisons in Table 5 and Figure 5 in our revised manuscript.
>
> As a general framework, **our proposed mixed-guidance strategy can also be integrated with architectural innovations from TacoGFN and 3DSynthFlow**, which may further enhance performance.

---

> ### Author Response · Authors · 2025-11-20
> **Response by Authors (Part 2)**
>
> > W2: Lowered chemical diversity
>
> The high-diversity results are reported in the updated Benchmark Tables 1 and 2 in our manuscript and displayed as follows.
>
> | Iter | $w$ | Vina Score (↓) | QED (↑) | SA (↑) | Diversity | Success Rate |
> | ---- | --- | -------------- | ------- | ------ | --------- | ------------ |
> | 0    | 1   | -8.18          | 0.57    | 0.76   | 0.69      | 51.1%        |
> | 0    | 0.2 | -8.33          | 0.57    | 0.75   | 0.59      | 52.0%        |
> | 1    | 1   | -8.44          | 0.57    | 0.77   | 0.69      | 53.2%        |
> | 1    | 0.2 | -8.55          | 0.55    | 0.81   | 0.58      | 55.7%        |
> | 2    | 1   | -8.58          | 0.55    | 0.81   | 0.68      | 55.5%        |
> | 2    | 0.2 | -8.70          | 0.54    | 0.85   | 0.53      | 58.1%        |
>
> Interestingly, we find that the decline in diversity results from the combined effects of action-space concentration during RL iterations and the noise-reduction strategy. When noise reduction is removed, RL4SBDD-M attains diversity levels comparable to the reference model MolCRAFT (0.68 vs. 0.70). Moreover, adjusting $w$ allows users to flexibly control the trade-off between diversity and potency. We have reported the results under the high-diversity setting in Table 1 in our revised version.

---

> ### Author Response · Authors · 2025-11-20
> **Response by Authors (Part 3)**
>
> > W3: RL4SBDD-M exhibits a high prevalence of aromatic rings and risks reward hacking
>
> ### **1. Reward hacking is a shared problem for SBDD preference alignment methods, regardless of their algorithmic distinctions**
>
> As suggested, we analyze the **ring distribution**, **calculated LogP and TSPA**, and **protein-ligand interaction patterns** with PLIP [5] for reference molecules, molecules from the ChEMBL database [6], RL4SBDD-M generations, and several alignment approaches that have been unified within our framework. We also follow *Gao et al. (2024)* [7] to calculate the **Delta Score** by evaluating the Vina Docking score against a randomly selected pocket in the test set. The key observations are as follows:
>
> - **Across methods optimizing Affinity + SA, we observe a consistent increase in aromatic rings**, which strengthens $\pi$–$\pi$ and $\pi$–cation interactions.
> - Consequently, the number of 5-rings, hydrogen bonds, and salt bridges drops significantly, and **an increase in cLogP** indicates high lipophilicity, making the molecules less desirable.
> - **Compared to the reference model, improving binding affinity does not yield significant gains in selectivity**.
>
> While RL4SBDD-M shares the same model architecture with MolCRAFT, the hyperparameter choices in our main experiment mainly seek the best binding and SA scores, and reduce the adherence to the reference policy, leading to a significant distribution shift.
>
> However, we argue that **the risk of reward hacking originates primarily from the reward design itself [8, 9] and from inherent molecular properties, rather than algorithmic design**. This is supported by our empirical findings, including a negative correlation between Vina Score and the number of 6-membered rings (Spearman $\rho$=-0.55), and a positive correlation between cLogP and the number of 6-membered rings (Spearman $\rho$=0.63). This underscores **the value of our theoretical unification that facilitates researchers to identify shared limitations across diverse SBDD approaches**.
>
> ### **2. The RL4SBDD framework provides insights into mitigating reward hacking**
>
> As noted in R1, RL4SBDD is a general framework built upon soft RL and can be applied to arbitrary reward functions. Building on this foundation, we develop and empirically validate two strategies to mitigate the excessive preference for 6-membered rings, with the corresponding experiments presented in the tables above.
>
> - **Introducing more objectives into the reward**. We incorporate the cLogP value into the reward function to regularize lipophilicity and control the number of aromatic rings. Specifically, we calculate cLogP for molecules generated in the second RL iteration with RDKit [10], clip the value to the range of -3$\sim$7, normalize it into 0$\sim$1, and consider cLogP=1 as the most desirable value [11]. For gradient guidance, the reward is calculated as $|cLogP_{norm}-0.4|$. For reward conditioning, we set the conditioning label $cLogP_{norm}$ as 0.4. Experiments show that RL4SBDD-M achieves a balanced trade-off between binding affinity, synthesizability, and LogP. Importantly, the number of aromatic rings is reduced to a level comparable to that found in ChEMBL. Although the average cLogP does not exactly match the target value, this may stem from the distribution of the enriched dataset produced through iterative RL that optimizes only Vina Score + SA.
>
> - **Controlling adherence to the reference policy (KL regularization)**: Since reference molecules typically originate from biologically plausible candidates, we can moderate reward over-optimization by adjusting the guidance weights ($1/\lambda$ for gradient guidance and $w$ for reward conditioning). We conduct an experiment where these weights are reduced by half ($1/\lambda=15$, $w=0.5$), which also mitigates the prevalence of excessive 6-membered rings.
>
> Unfortunately, at present there is no reliable or explicit approach to quantify molecular selectivity without knowing the specific negative pockets to avoid. Nevertheless, our framework may provide conceptual guidance for optimizing selectivity using RL-based methods.
>
> Overall, developing genuinely desirable molecules in practical SBDD remains a fundamental challenge that requires joint advances in reward formulation, multi-objective optimization, and robust evaluation protocols. **Our contribution is a theoretical unification of existing approaches that reveals shared limitations and provides insights into reward maximization, along with a simple and effective methodology that improves multi-objective optimization**. We have incorporated the analysis of potential reward hacking modes and mitigation strategies into Section 5.5 and Appendix E.6 in the revised version of our paper.

---

> ### Author Response · Authors · 2025-11-20
> **Response by Authors (Part 4)**
>
> The experiment results for W3 are in the following tables:
> | Model                                      | 3-ring | 4-ring | 5-ring | 6-ring | 7-ring | 8-ring |
> | ------------------------------------------ | ------ | ------ | ------ | ------ | ------ | ------ |
> | Ref                                        | 0.04   | 0.00   | 0.73   | 1.63   | 0.02   | 0.00   |
> | MolCRAFT                                   | 0.00   | 0.01   | 0.65   | 1.94   | 0.16   | 0.02   |
> | MolJO                                      | 0.00   | 0.00   | 0.45   | 3.24   | 0.18   | 0.01   |
> | MolCRAFT+DPO                               | 0.00   | 0.01   | 0.61   | 2.31   | 0.21   | 0.03   |
> | RL4SBDD-M (SA+Affinity)                    | 0.00   | 0.00   | 0.20   | 3.04   | 0.32   | 0.00   |
> | RL4SBDD-M (SA+Affinity+cLogP)              | 0.00   | 0.01   | 0.54   | 2.65   | 0.13   | 0.01   |
> | RL4SBDD-M (SA+Affinity, KL regularization) | 0.00   | 0.01   | 0.53   | 2.74   | 0.28   | 0.02   |
> | ChEMBL                                     | 0.05   | 0.03   | 0.88   | 2.55   | 0.04   | 0.00   |
>
> | Model                                      | Avg. HBD | Avg. HBA | Avg. $\pi$-$\pi$ Interactions | Avg. $\pi$-cation Interactions | Avg. Salt Bridges |
> | ------------------------------------------ | -------- | -------- | ----------------------------- | ------------------------------ | ----------------- |
> | Ref                                        | 2.44     | 1.09     | 0.27                          | 0.08                           | 0.55              |
> | MolCRAFT                                   | 2.68     | 1.21     | 0.34                          | 0.08                           | 0.55              |
> | MolCRAFT+DPO                               | 2.50     | 1.01     | 0.44                          | 0.09                           | 0.46              |
> | MolJO                                      | 1.17     | 0.48     | 0.74                          | 0.14                           | 0.17              |
> | RL4SBDD-M (SA+Affinity)                    | 0.75     | 0.18     | 0.86                          | 0.12                           | 0.09              |
> | RL4SBDD-M (SA+Affinity+cLogP)              | 2.14     | 0.74     | 0.60                          | 0.11                           | 0.35              |
> | RL4SBDD-M (SA+Affinity, KL regularization) | 1.74     | 0.41     | 0.59                          | 0.09                           | 0.21              |
>
> | Model                                      | Avg. TPSA | Med. Vina Score | Avg. SA | Avg. cLogP |
> | ------------------------------------------ | --------- | --------------- | ------- | ---------- |
> | Ref                                        | 92.48     | -6.46           | 0.73    | 0.89       |
> | MolCRAFT                                   | 93.86     | -6.95           | 0.67    | 1.09       |
> | ~MolJO~ MolCRAFT+DPO                                     | 77.71     | -7.41           | 0.71    | 2.06       |
> | ~MolCRAFT+DPO~ MolJO                               | 40.46     | -8.02           | 0.78    | 4.20       |
> | RL4SBDD-M (SA+Affinity)                    | 20.75     | -8.70           | 0.85    | 5.48       |
> | RL4SBDD-M (SA+Affinity+cLogP)              | 68.55     | -7.88           | 0.76    | 3.08       |
> | RL4SBDD-M (SA+Affinity, KL regularization) | 37.84     | -8.38           | 0.77    | 4.30       |
> | ChEMBL                                     | 94.11     | -               | 0.77    | 3.31       |
>
> | Model     | Delta Score (median$\pm$std) |
> | --------- | ---------------------------- |
> | Ref       | 0.60$\pm$0.10                |
> | MolCRAFT  | 0.69$\pm$0.13                |
> | MolJO     | 0.72$\pm$0.17                |
> | RL4SBDD-M | 0.72$\pm$0.15                |

---

> > ### Author Response · Authors · 2025-11-26
> > **Fixing a Typo**
> >
> > We sincerely apologize for the typing error in the third table of our previous rebuttal. This has been corrected in the revised response.

---

> ### Author Response · Authors · 2025-11-20
> **Response by Authors (Part 5)**
>
> > W4.2: RL4SBDD-M performance in jointly optimizing affinity, QED, and SA, and how it balances between multiple objectives
>
> Thanks for your suggestion. We conduct experiments with an objective that jointly optimizes Affinity, SA, and QED. Due to computational resource constraints, we do not perform the full RL procedure; instead, we train our models using the enriched dataset obtained at the second RL iteration under the Affinity + SA optimization setup. We evaluate the following reward designs:
>
> - **Independent reward modeling (the original design of our approach)** In this setting, the value model predicts each property independently, and the corresponding gradients for each dimension are combined during sampling. The normalized values of the three properties are also encoded as additional conditioning inputs to the reward-conditioned policy.
>
> - **MOOD [12] reward**, where a single reward $r\in [0,1]$ is computed as the product of normalized Vina Score, QED, and SA values.
>
>   | Objectives                           | Med. Vina Score | Avg. QED  | Avg.SA   |
>   | ------------------------------------ | ---------- | ---- | ---- |
>   | Affinity+SA (Independent reward)     | -8.70      | 0.54 | 0.85 |
>   | Affinity+SA+QED (Independent reward) | -8.32      | 0.60 | 0.83 |
>   | Affinity+SA+QED (MOOD reward)        | -8.58      | 0.66 | 0.72 |
>
> The above results show that RL4SBDD-M could improve QED significantly at the expense of a decline in binding affinity and SA with both reward formulations. More importantly, **our approach demonstrates a more balanced trade-off**, while the MOOD reward places more emphasis on optimizing Vina Score and QED. These analyses are incorporated in Section 5.5 in our revised version.
>
> Furthermore, we investigate how RL4SBDD balances multiple objectives by adjusting the conditioning label to the reward-conditioned policy, from which we observe that optimizing one objective does not significantly compromise the overall performance on the other.
>
> | Conditioning Label | Affinity=1 | Affinity=0.5 | Affinity=0 |
> | ------------------ | ---------- | ------------ | ---------- |
> | SA=1               | -8.38/0.83 | -7.59/0.84   | -6.31/0.83 |
> | SA=0.5             | -8.21/0.71 | -7.51/0.68   | -6.30/0.67 |
> | SA=0               | -8.03/0.65 | -7.47/0.62   | -6.06/0.58 |
>
> As stated in Lines 484-485 in our initial submission (Section 6), we leave investigation on more reward functions for future work, and we encourage the SBDD community to propose more realistic evaluation dimensions that better capture the practical needs of domain experts.
>
> > Q2: A plot showing performance evolution across RL iterations
>
> Thank you for the suggestion. The performance comparisons were originally presented in Table 2 of our submission and analyzed in Lines 466–470 (Section 5.4). The key finding is that **the average reward (Vina Score + SA) consistently improves with reward conditioning and mixed guidance, while gradient guidance yields only marginal gains in the second iteration**. We include an illustration of the average advantage over the reference model across different RL iterations in Figure 15 of the revised manuscript.
>
> References
>
> [1] Zhou, Xiangxin, et al. "DecompOpt: Controllable and Decomposed Diffusion Models for Structure-based Molecular Optimization" ICLR 2024
>
> [2] Qiu K, Song Y, Yu J, et al. "Empower Structure-Based Molecule Optimization with Gradient Guided Bayesian Flow Networks" ICML 2025
>
> [3] Shen, Tony, et al. "TacoGFN: target-conditioned GFlowNet for structure-based drug design." TMLR 2024
>
> [4] Shen, Tony, et al. "Compositional Flows for 3D Molecule and Synthesis Pathway Co-design." ICML 2025
>
> [5] Salentin S, Schreiber S, Haupt V J, et al. "PLIP: fully automated protein–ligand interaction profiler" Nucleic acids research, 2015
>
> [6] Gaulton A, Hersey A, Nowotka M, et al. "The ChEMBL database in 2017" Nucleic acids research, 2017
>
> [7] Gao B, Ren M, Ni Y, et al. "Rethinking specificity in SBDD: leveraging delta score and energy-guided diffusion", ICML 2024
>
> [8] Skalse J, Howe N, Krasheninnikov D, et al. "Defining and characterizing reward gaming" NeurIPS 2022
>
> [9] Karwowski J, Hayman O, Bai X, et al. "Goodhart's Law in Reinforcement Learning", ICLR 2024
>
> [10] Bento A P, Hersey A, Félix E, et al. "An open source chemical structure curation pipeline using RDKit" Journal of Cheminformatics, 2020
>
> [11] Arnott J A, Planey S L. "The influence of lipophilicity in drug discovery and design" Expert opinion on drug discovery, 2012
>
> [12] Lee, Seul, Jaehyeong Jo, and Sung Ju Hwang. "Exploring chemical space with score-based out-of-distribution generation." ICML 2023

---

> ### Comment · Reviewer_3QpG · 2025-11-24
>
> I thank the authors for their detailed response and additional experiments. I have carefully reviewed the provided data, including the PLIP analysis and the results on multi-objective optimization.
>
> However, the additional analyses have unfortunately reinforced my initial concerns regarding the experimental setting and the validity of the generated molecules. I maintain my rating based on the following observations:
>
> **1. Lack of Trade-off between Binding Affinity and SA score.** The authors argue that RL4SBDD effectively optimizes binding affinity and SA scores. However, looking at the results provided in the response (W4.2), I observe no significant trade-off between the Vina score and the SA score. This suggests a positive correlation between these two metrics in this context, which differs from the explicit trade-off observed between Vina score and QED.
>
> This trend is consistent with findings in other literature. For instance, Table 12 in the TacoGFN paper reports that while increasing the QED constraint significantly degrades the Vina score, increasing the SA score threshold (e.g., from 0.75 to 0.8) actually leads to a slight improvement in the Vina score. Therefore, achieving SOTA performance by jointly optimizing only Vina and SA—metrics that do not inherently conflict—does not sufficiently demonstrate the model's capability to handle the complex multi-objective optimization landscape required for real-world drug discovery.
>
> **2. Confirmation of Reward Hacking** My concern regarding reward hacking was not merely about the numerical scores but the structural validity of the molecules. The authors' new analysis of protein-ligand interaction patterns (Table in response to W3) reveals that the generated molecules exhibit 2-3 times more π-π interactions and π-cation interactions compared to the reference ligands.
>
> This excessive reliance on aromatic interactions confirms that the model is indeed "hacking" the Vina scoring function by over-sampling aromatic rings, as hypothesized in my initial review. Furthermore, the fact that the Vina score drops significantly when cLogP or QED is added to the optimization objective serves as evidence that the previously reported high affinity scores were driven by this artificial over-accumulation of hydrophobic/aromatic groups rather than specific, high-quality binding interactions.
>
> **3. The Purpose of Multi-Objective Benchmarks** As the authors noted, metrics like Vina, QED, and SA are poor proxies. In fact, these metrics are less considered than simple Lipinski Ro5 in real world application. However, the reason the AI community typically benchmarks on the simultaneous optimization of all three (despite their limitations) is to demonstrate a method's ability to balance conflicting objectives—a simulation of real-world drug discovery campaigns where potency must be balanced against ADME/T properties.
>
> Demonstrating superiority in a setting that optimizes only positively correlated metrics (Vina + SA), while failing to maintain performance when a conflicting metric (QED) is introduced, significantly limits the contribution of this work. The current results suggest that the framework struggles to find a viable chemical space when genuine trade-offs are enforced.
>
> For these reasons, I believe the current experimental evidence is insufficient to support the claims of practical utility in SBDD.

---

> > ### Author Response · Authors · 2025-11-26
> > **Response by Authors**
> >
> > We thank the reviewer for carefully reading our responses and providing immediate feedback. We would like to provide additional experiment results and analysis here to clarify potential misunderstandings and further address your concerns.
> >
> > > Q1: Lack of Trade-off between Binding Affinity and SA score
> >
> > While there is a slightly positive correlation between the Vina Score and the SA score (Spearman $\rho$=0.33), we observe a clear trade-off by directly manipulating the weights within our reward formulation used for gradient guidance. Specifically, for $R=w_1\cdot r_{\text{affinity}}+w_2\cdot r_{\text{SA}}$, we show that increasing the weight of either $w_1$ or $w_2$ consistently leads to reduced performance on the other:
> >
> > | Weight    | $w_1=1$    | $w_1=0.5$  | $w_1=0$    |
> > | --------- | ---------- | ---------- | ---------- |
> > | $w_2=1$   | -8.18/0.76 | -7.74/0.77 | -6.94/0.80 |
> > | $w_2=0.5$ | -8.91/0.74 | -7.94/0.76 | -7.02/0.79 |
> > | $w_2=0$   | -9.01/0.69 | -8.34/0.71 | -6.95/0.67 |
> >
> >    (In each cell, we report the Vina Score of 1000 generated molecules on the left of / and SA on the right of /)
> >
> > Therefore, we believe **it is non-trivial to simultaneously optimize these two objectives**. Besides, this result highlights a key contribution of our work: as shown in our detailed response to Reviewer F2gE (part 2), we specifically intend to demonstrate that via **reward conditioning**, **RL4SBDD-M effectively handles multi-objective optimization** and ensures that **no single objective overwhelms the other**.
> >
> > > Q2: Confirmation of reward hacking
> >
> > We agree with the underlying premise that generating genuinely desirable molecules requires joint efforts in two distinct areas: developing **more reliable computational proxies** as optimization objectives (the "what to optimize"), and designing **more effective strategies** to maximize a given reward (the "how to optimize").
> >
> > The **main goal of our proposed framework lies in the latter area, NOT the former.** Our main experiment results on optimizing the Vina Score and SA Score serve as a proof of concept for our effective algorithmic strategy.
> >
> > Furthermore, **the additional experiments incorporating cLogP and QED further substantiate that RL4SBDD-M generalizes effectively to other reward functions** . The risk of **reward hacking** is therefore understood to originate primarily from the **reward design itself**, rather than the algorithmic design [1, 2], and we believe our contribution offers a robust mechanism for maximizing any well-defined multi-objective function.
> >
> > > Q3: Multi-Objective Optimization
> >
> > As suggested by previous studies [3, 4], introducing a conflicting metric (QED) will inevitably degrade the performance of other metrics (Vina Score and SA score). Fortunately, through grid search on the weights of each metric in the reward formulation $R=w_1\cdot r_{\text{affinity}}+w_2\cdot r_{\text{SA}}+w_3\cdot r_{\text{QED}}$, we observe that:
> >
> > - **Given the same set of optimization objectives, RL4SBDD-M achieves the best results in all metrics that we aim to improve** (Comparing rows 2-4 and rows 5-8).
> >
> > - **Even when a conflicting objective (QED) is incorporated, the Vina Score and SA score of RL4SBDD-M still outperforms state-of-the-art models** (Comparing rows 2,3 and rows 7,8).
> >
> > These observations effectively demonstrate the general power of our method for balancing multiple, even conflicting, objectives. We have incorporated these analyses in Table 3 and Section 5.5 in our revised manuscript.
> >
> > | Model | Optimization Objective | Med. Vina Score | Med. Vina Dock | Avg. QED | Avg. SA  |
> > | - | - | - | - | - | - |
> > | Ref| - | -6.46 | -7.26 | 0.48 | 0.73 |
> > | AliDiff  | Vina Score | -7.95 | -8.81 | 0.50| 0.57  |
> > | MolJO  | Vina Score+SA  | -8.02 | -9.13  | 0.56 | 0.78 |
> > | RL4SBDD-M (Ours) | Vina Score+SA | **-8.70** | **-9.28**| 0.54 | **0.85** |
> > | MolJO | Vina Score+QED+SA | -7.47 | -8.27 | 0.62 | 0.73 |
> > | TacoGFN | Vina Score+QED+SA | - | -8.44 | **0.67** | 0.79 |
> > | RL4SBDD-M (Ours, $w_1=1, w_2=1, w_3=1$)     | Vina Score+QED+SA | -8.32           | -8.84 | 0.60     | 0.83     |
> > | RL4SBDD-M (Ours, $w_1=1, w_2=0.2, w_3=1.5$) | Vina Score+QED+SA | -8.50           | -9.04 | **0.67** | 0.80     |
> >
> > References
> >
> > [1] Skalse J, Howe N, Krasheninnikov D, et al. "Defining and characterizing reward gaming" NeurIPS 2022
> >
> > [2] Karwowski J, Hayman O, Bai X, et al. "Goodhart's Law in Reinforcement Learning", ICLR 2024
> >
> > [3] Qiu K, Song Y, Yu J, et al. "Empower Structure-Based Molecule Optimization with Gradient Guided Bayesian Flow Networks" ICML 2025
> >
> > [4] Shen, Tony, et al. "TacoGFN: target-conditioned GFlowNet for structure-based drug design." TMLR 2024

---

### Official Review · Reviewer_mVUX · 2025-11-01

**Soundness:** 3
**Presentation:** 3
**Contribution:** 3
**Rating:** 6
**Confidence:** 3

**Summary:**

The paper proposes RL4SBDD, a unifying reinforcement learning view of preference alignment for structure-based drug design, and introduces RL4SBDD-M, a practical method that conditions the policy on continuous reward targets, updates the dataset with iterative RL, and uses a mixed guidance schedule with reduced noise during sampling. The authors argue that popular approaches such as gradient guidance and DPO are optimizing a KL-constrained reward objective in soft RL and diagnose three weaknesses in current practice: biased value estimates early in the trajectory, inefficient use of pairwise data in DPO, and off-policy distribution shift between training and inference. On CrossDocked2020, RL4SBDD-M reports strong improvements, including a median Vina score of −8.70 kcal/mol and a 58.1% success rate with a 4.6× sampling speedup over a recent state of the art, and further reaches 93% success by selecting the best of 50 generations under oracle screening.

**Strengths:**

The work’s strengths are the conceptual clarity of the RL framing, which helps reconcile guidance-based and preference-based methods and yields concrete propositions that tie guidance to optimal policies under soft RL assumptions. The reward-conditioned policy is a simple, scalable idea that sidesteps noisy value learning early in sampling and exploits supervision from all training examples rather than only best–worst pairs, while the iterative RL loop squarely targets distribution shift. The mixed guidance schedule is well motivated by analyses showing when gradient guidance is unreliable or redundant, and the reduced-noise corrector connects neatly to a predictor–corrector interpretation of the sampler. Empirically, the paper compares against diverse baselines with a consistent evaluation suite and reports both quality and efficiency gains, complemented by ablations that isolate the contributions of each ingredient and a test-time scaling study that is relevant for practical lead finding. The paper is clearly written, the figures are helpful, and the inclusion of implementation details and a reproducibility statement is appreciated.

**Weaknesses:**

The main weaknesses concern scope, assumptions, and external validity. Several claims of unification and improvement rest on idealized conditions such as the Dirac-prior assumption in the linkage between guided transitions and the optimal policy; more discussion of how deviations from these assumptions affect guarantees would strengthen the theoretical framing. The empirical validation is heavily anchored to docking and easy-to-compute cheminformatic metrics, which are known to be imperfect surrogates for true binding and developability; without broader property coverage or prospective wet-lab feedback there remains risk of score hacking and limited translational value. Diversity decreases across RL iterations, suggesting potential mode concentration that may impair downstream medicinal chemistry; additional safeguards or multi-objective controls could help. The critique of DPO’s pair construction would be more convincing with a direct comparison to BT-consistent training that uses all pairs under identical compute. The test-time scaling results rely on oracle screening and pocket-specific atom-count priors, so the practicality and fairness of the reported compute budgets relative to search-based baselines deserve closer scrutiny. Finally, sensitivity analyses for the guidance switch point, guidance weight, and noise scale are informative but point to nontrivial tuning; reporting statistical significance across seeds and clarifying hardware parity for timing would improve confidence.

---

For rebuttal and discussion:

1. Theoretical claims hinge on idealized assumptions such as a Dirac-prior link between guidance and the optimal policy.
2. Heavy reliance on docking and simple cheminformatic metrics risks score hacking and limits biological validity.
3. Diversity declines across RL iterations, suggesting potential mode concentration that could hinder downstream use.
4. Critique of DPO efficiency would be stronger with compute-matched, all-pairs BT-consistent comparisons.
5. Test-time scaling depends on oracle screening and pocket-specific priors, raising fairness and practicality questions.
6. Performance appears sensitive to guidance weights, switch points, and noise scales, with limited variance reporting across seeds and hardware.

**Questions:**

-

---

> ### Author Response · Authors · 2025-11-20
> **Response by Authors (Part 1)**
>
> We thank the reviewer for your appreciation of our *"conceptual clarity in RL framing"*, *"simple scalable idea"*, and *"well-motivated design"*, as well as for recognizing the *"quality and efficiency gains"* demonstrated in our experiments. We address your concerns as follows.
>
> > Q1: The Dirac prior assumption for gradient guidance
>
> Thank you for pointing this out. We provide the corresponding theoretical analysis below:
>
> - As noted in Line 917 in the original submission, **the initial state distribution for our backbone model, i.e., BFN [1], is indeed Dirac**.
>
> - For a Gaussian prior in diffusion models, if $\Phi_V$ is capable of predicting the value of initial states ($V_{\pi_{ref}}(\mathbf{s}_0)$), we can prove that **sampling from the reward-reshaped prior** $p_I'(\mathbf{s}_0)\propto p_I(\mathbf{s}_0)\Phi_V(\mathbf{s}_0;t=0)$ **by importance sampling and gradient guidance is equivalent to the soft optimal policy**.
>
>   We have incorporated these discussions into the revised version of our manuscript.
>
> > Q2: Heavy reliance on docking and simple cheminformatic metrics, risks of score hacking, and limited biological validity
>
> Thank you for the insightful suggestion. In our experiments, we formulate the reward objective as a combination of binding affinity and synthesizability, and we follow the evaluation protocols of prior works [2, 3] to ensure fair comparison. Our results demonstrate that **RL4SBDD-M effectively maximizes the specified reward objective**.
>
> By unifying SBDD preference alignment methods with soft RL, we argue that **reward misspecification is a shared and intrinsic risk across existing approaches, regardless of their algorithmic distinctions**. Empirically, we analyze the ring distribution and protein-ligand interaction patterns with PLIP [4], and observe that all existing works optimizing Affinity + SA tend to introduce an increased number of aromatic rings to achieve tighter binding via stronger $\pi$-$\pi$ and $\pi$-cation interactions. However, an excessive number of 6-membered rings increases molecular lipophilicity, potentially leading to suboptimal ADME properties [5].
>
> Fortunately, **RL4SBDD is a general RL-based framework for SBDD that can be applied to arbitrary verifiable rewards**, and we can derive several strategies to mitigate score hacking risks by drawing inspiration from our framework.
>
> - **Introducing more objectives into the reward**. As an example, we include the calculated water–octanol partition coefficient (cLogP) to regularize lipophilicity and limit aromatic ring proliferation. Specifically, we compute LogP for the molecules generated in the second RL iteration using RDKit [6], clip values to the range of -3$\sim$7, normalize them to 0$\sim$1, and consider LogP = 1 as the most desirable [7]. For gradient guidance, we compute the reward as $|cLogP_{norm} - 0.4|$. For reward conditioning, we set the conditioning label to $cLogP_{norm} = 0.4$. Experiments show that RL4SBDD-M achieves a balanced trade-off among binding affinity, synthesizability, and desirable LogP. Moreover, the number of aromatic rings is reduced to an acceptable range compared to ChEMBL [8], a widely used drug-like molecule library.
>
> - **Controlling adherence to the reference policy (KL regularization)**: Since reference molecules typically originate from biologically plausible candidates, we can moderate reward over-optimization by adjusting the guidance weights ($1/\lambda$ for gradient guidance and $w$ for reward conditioning). We conduct an experiment where these weights are reduced by half ($1/\lambda$=15, $w=0.5$), which also mitigates the prevalence of excessive 6-membered rings.
>
> We have provided a thorough discussion of this issue in Section 5.5 and Appendix E.6 in our revised manuscript.

---

> ### Author Response · Authors · 2025-11-20
> **Response by Authors (Part 2)**
>
> The experiment results for Q2 are in the following tables:
> | Model                                      | 3-ring | 4-ring | 5-ring | 6-ring | 7-ring | 8-ring |
> | ------------------------------------------ | ------ | ------ | ------ | ------ | ------ | ------ |
> | Ref                                        | 0.04   | 0.00   | 0.73   | 1.63   | 0.02   | 0.00   |
> | MolCRAFT                                   | 0.00   | 0.01   | 0.65   | 1.94   | 0.16   | 0.02   |
> | MolJO                                      | 0.00   | 0.00   | 0.45   | 3.24   | 0.18   | 0.01   |
> | MolCRAFT+DPO                               | 0.00   | 0.01   | 0.61   | 2.31   | 0.21   | 0.03   |
> | RL4SBDD-M (SA+Affinity)                    | 0.00   | 0.00   | 0.20   | 3.04   | 0.32   | 0.00   |
> | RL4SBDD-M (SA+Affinity+cLogP)              | 0.00   | 0.01   | 0.54   | 2.65   | 0.13   | 0.01   |
> | RL4SBDD-M (SA+Affinity, KL regularization) | 0.00   | 0.01   | 0.53   | 2.74   | 0.28   | 0.02   |
> | ChEMBL                                     | 0.05   | 0.03   | 0.88   | 2.55   | 0.04   | 0.00   |
>
> | Model                                      | Avg. HBD | Avg. HBA | Avg. $\pi$-$\pi$ Interactions | Avg. $\pi$-cation Interactions | Avg. Salt Bridges |
> | ------------------------------------------ | -------- | -------- | ----------------------------- | ------------------------------ | ----------------- |
> | Ref                                        | 2.44     | 1.09     | 0.27                          | 0.08                           | 0.55              |
> | MolCRAFT                                   | 2.68     | 1.21     | 0.34                          | 0.08                           | 0.55              |
> | MolCRAFT+DPO                               | 2.50     | 1.01     | 0.44                          | 0.09                           | 0.46              |
> | MolJO                                      | 1.17     | 0.48     | 0.74                          | 0.14                           | 0.17              |
> | RL4SBDD-M (SA+Affinity)                    | 0.75     | 0.18     | 0.86                          | 0.12                           | 0.09              |
> | RL4SBDD-M (SA+Affinity+cLogP)              | 2.14     | 0.74     | 0.60                          | 0.11                           | 0.35              |
> | RL4SBDD-M (SA+Affinity, KL regularization) | 1.74     | 0.41     | 0.59                          | 0.09                           | 0.21              |
>
> | Model                                      | Avg. TPSA | Med. Vina Score | Avg. SA | Avg. cLogP |
> | ------------------------------------------ | --------- | --------------- | ------- | ---------- |
> | Ref                                        | 92.48     | -6.46           | 0.73    | 0.89       |
> | MolCRAFT                                   | 93.86     | -6.95           | 0.67    | 1.09       |
> | ~MolJO~ MolCRAFT+DPO                                      | 77.71     | -7.41           | 0.71    | 2.06       |
> | ~MolCRAFT+DPO~ MolJO                               | 40.46     | -8.02           | 0.78    | 4.20       |
> | RL4SBDD-M (SA+Affinity)                    | 20.75     | -8.70           | 0.85    | 5.48       |
> | RL4SBDD-M (SA+Affinity+cLogP)              | 68.55     | -7.88           | 0.76    | 3.08       |
> | RL4SBDD-M (SA+Affinity, KL regularization) | 37.84     | -8.38           | 0.77    | 4.30       |
> | ChEMBL                                     | 94.11     | -               | 0.77    | 3.31       |

---

> > ### Author Response · Authors · 2025-11-26
> > **Fixing a Typo**
> >
> > We sincerely apologize for the typing error in the third table of our previous rebuttal. This has been corrected in the revised response.

---

> ### Author Response · Authors · 2025-11-20
> **Response by Authors (Part 3)**
>
> > Q3: Diversity decline across RL iterations
>
> We report the model performance and diversity under a high-diversity setting ($\sigma=1$) after each RL iteration. Interestingly, we observe that **diversity decline arises from a combination of action space concentration during RL iterations and the noise reduction strategy**. By removing noise reduction, RL4SBDD-M achieves a level of diversity comparable to the reference model MolCRAFT (0.68 v.s. 0.70). Moreover, by adjusting $w$, users can conveniently balance the trade-off between diversity and potency. We have reported the results and adjusted the statements in Lines 487 and 494-495 in our revised manuscript.
>
> | Iter | $w$ | Avg. Diversity | Avg. Success Rate |
> | ---- | --- | --------- | ------------ |
> | 0    | 1   | 0.69      | 51.1%        |
> | 0    | 0.2 | 0.59      | 52.0%        |
> | 1    | 1   | 0.69      | 53.2%        |
> | 1    | 0.2 | 0.58      | 55.7%        |
> | 2    | 1   | 0.68      | 55.5%        |
> | 2    | 0.2 | 0.53      | 58.1%        |
>
> > Q4: Comparison with all-pairs DPO training
>
> Borrowing Table 6 in *Gu et al. (2024)* [9], we show that all-pairs DPO training yields even inferior results to best-worst DPO training in SBDD. As suggested by *Razin et al. (2025); Chen et al. (2025)* [10, 11], this may arise from the likelihood displacement induced by preference pairs with similar rewards.
>
> | Choice     | Med. Vina Score (↓) | Med. Vina Min (↓) | Avg. QED (↑) | Avg. SA (↑) |
> | ---------- | ------------------- | ----------------- | ------------ | ----------- |
> | best-worst | -7.95               | -8.17             | 0.50         | 0.56        |
> | all-pairs  | -7.82               | -7.96             | 0.50         | 0.56        |
>
> > Q5: Fairness of test-time scaling using oracle screening and atom counts from a pocket-specific prior
>
> The justifications for the experiment setting in test-time scaling are as follows:
>
> - **Use of oracle screening**. This setting resembles practical lead discovery workflows, where designed molecules are evaluated using wet-lab experiments and selected accordingly. The oracle functions act as proxies of such experimental validation. In addition, optimization-based approaches such as DecompDPO [12] and EvoSBDD [13] rely on oracle feedback to improve their generations. For fair comparison, we therefore adopt oracle-based screening.
>
> - **Use of a pocket-specific prior**. In real-world applications, the pocket structure is known, while the number of atoms in the designed binder is typically unspecified. Using a prior distribution conditioned on available pocket information offers design flexibility and increases the likelihood of successful outcomes. Furthermore, as discussed in Lines 419–420 in our original submission, several baselines [12, 13] operate by modifying molecular components—such as substituting functional groups—during generation, which naturally changes the atom count. Thus, we sample atom counts based on pocket size.
>
> Moreover, following the **self-ranking strategy** introduced in *Jumper J et al. (2021)* [14], we use the predicted properties of $\Phi_V$ to calculate the Z-score for each molecule and perform ranking. The resulting performance **is only slightly below that of oracle-based ranking**, highlighting the potential of conducting *in silico* screening when optimizing objectives that are difficult to verify experimentally. We have incorporated the results in Lines 463-464 and Appendix E.3 in our revised manuscript.
>
> | Best of K                  | 1           | 2           | 4            | 8            | 16           | 32           | 64           | 128          |
> | -------------------------- | ----------- | ----------- | ------------ | ------------ | ------------ | ------------ | ------------ | ------------ |
> | MolCRAFT (Self-ranking)    | 0.266/-7.83 | 0.341/-8.21 | 0.441/-8.58  | 0.529/-9.16  | 0.617/-9.50  | 0.655/-9.84  | 0.683/-10.06 | 0.720/-10.27 |
> | MolCRAFT (Oracle-ranking)  | 0.266/-7.83 | 0.380/-8.48 | 0.519/-8.88  | 0.644/-9.26  | 0.745/-10.03 | 0.810/-10.37 | 0.855/-10.63 | 0.890/-10.85 |
> | RL4SBDD-M (Self-ranking)   | 0.581/-9.20 | 0.676/-9.41 | 0.749/-9.88  | 0.789/-10.51 | 0.812/-10.78 | 0.830/-10.83 | 0.830/-11.42 | 0.830/-11.56 |
> | RL4SBDD-M (Oracle-ranking) | 0.581/-9.20 | 0.724/-9.76 | 0.812/-10.37 | 0.870/-10.89 | 0.899/-11.24 | 0.913/-11.64 | 0.930/-11.86 | 0.930/-12.19 |
>
>   (In each cell, we report the average success rate on the left of / and the median Vina Dock of the selected molecule on the right of /)

---

> ### Author Response · Authors · 2025-11-20
> **Response by Authors (Part 4)**
>
> > Q6: Performance sensitivity to hyperparameters
>
> Due to computational resource constraints, we sample 10 molecules per pocket to analyze the effects of different hyperparameters. All the experiments are conducted with an identical random seed on the same server with 8 NVIDIA A800 GPUs.
>
> References
>
> [1] Graves A, Srivastava R K, Atkinson T, et al. "Bayesian flow networks", arXiv 2023
>
> [2] Zhou, Xiangxin, et al. "DecompOpt: Controllable and Decomposed Diffusion Models for Structure-based Molecular Optimization" ICLR 2024
>
> [3] Qiu K, Song Y, Yu J, et al. "Empower Structure-Based Molecule Optimization with Gradient Guided Bayesian Flow Networks" ICML 2025
>
> [4] Salentin S, Schreiber S, Haupt V J, et al. "PLIP: fully automated protein–ligand interaction profiler" Nucleic acids research, 2015
>
> [5] Ritchie T J, Macdonald S J F. "The impact of aromatic ring count on compound developability–are too many aromatic rings a liability in drug design?" Drug discovery today, 2009
>
> [6] Bento A P, Hersey A, Félix E, et al. "An open source chemical structure curation pipeline using RDKit" Journal of Cheminformatics, 2020
>
> [7] Arnott J A, Planey S L. "The influence of lipophilicity in drug discovery and design" Expert opinion on drug discovery, 2012
>
> [8] Gaulton A, Hersey A, Nowotka M, et al. "The ChEMBL database in 2017"" Nucleic acids research, 2017
>
> [9] Gu, Siyi, et al. "Aligning target-aware molecule diffusion models with exact energy optimization." NeurIPS 2024
>
> [10] Razin N, Malladi S, Bhaskar A, et al. "Unintentional unalignment: Likelihood displacement in direct preference optimization." ICLR 2025
>
> [11] Chen P, Chen X, Yin W, et al. "ComPO: Preference alignment via comparison oracles." NeurIPS 2025
>
> [12] Cheng X, Zhou X, Yang Y, et al. "Decomposed direct preference optimization for structure-based drug design" arXiv 2024
>
> [13] Reidenbach D. "EvoSBDD: Latent evolution for accurate and efficient structure-based drug design", ICLR 2024 workshop
>
> [14] Jumper J, Evans R, Pritzel A, et al. "Highly accurate protein structure prediction with AlphaFold" Nature 2021

---

### Official Review · Reviewer_F2gE · 2025-11-01

**Soundness:** 3
**Presentation:** 2
**Contribution:** 2
**Rating:** 4
**Confidence:** 3

**Summary:**

This paper introduces RL4SBDD, a reinforcement learning framework for structure-based drug design that unifies gradient guidance and direct preference optimization (DPO) under a soft policy optimization perspective. The authors identify challenges in existing SBDD methods, including biased value estimation, inconsistent reward interpretation, and off-policy training instability. They propose a reward-conditioned policy that reconstructs molecules based on their rewards, together with iterative RL training and guided sampling, to improve alignment with affinity and synthetic accessibility objectives. Experiments on CrossDocked2020 show improved docking affinity, generative quality, and inference efficiency compared to prior approaches.

**Strengths:**

- The paper presents a clear and unified theoretical framework that connects gradient-guided diffusion, direct preference optimization, and KL-regularized reinforcement learning in the context of structure-based drug design.
- The proposed reward-conditioned generative policy and iterative RL refinement strategy are implemented thoughtfully, with strong engineering execution.
- Experimental results demonstrate consistent performance gains and significant inference-time speedup compared to state-of-the-art diffusion-based molecular design models, indicating that the RL perspective not only clarifies the theory but also yields practical benefits.

**Weaknesses:**

- The three proposed components, reward conditioning, iterative RL, and mixed guidance, are all adaptations of existing techniques from diffusion modeling and RL literature rather than fundamentally new contributions. The theoretical unification, while neat, largely reuses soft-RL derivations known from Haarnoja et al. (2017) and Schulman et al. (2017).
- The method is only evaluated on one benchmark (CrossDocked2020). There is no demonstration of generalization to other protein–ligand datasets (e.g., PDBBind), which weakens claims of broad applicability.
- Although the RL formulation is mathematically sound, it doesn’t yield deeper theoretical insights or new algorithmic properties. The propositions largely restate standard equivalences between KL-regularized RL and reward-weighted likelihoods.
- While the reported 6.8% success-rate gain is notable, it is relatively modest given the methodological complexity. It remains unclear whether the improvement primarily stems from the reward-conditioning trick or the iterative data expansion.

**Questions:**

- The paper adopts normalized Vina and SA scores as independent reward dimensions. How does the proposed method trade off between binding affinity and synthesizability when the objectives conflict, and is there an explicit mechanism to avoid dominance of one reward over the other?
- The work emphasizes bypassing value-function estimation by reward conditioning and iterative dataset expansion. Could the authors provide empirical evidence showing that reward conditioning yields superior credit assignment compared to learned value models, beyond the early-stage bias argument?
- The experiments are conducted primarily on CrossDocked2020. Have you tested generalization to other SBDD datasets (e.g., PDBBind)?

---

> ### Author Response · Authors · 2025-11-20
> **Response by Authors (Part 1)**
>
> We thank the reviewer for carefully examining our work and providing insightful suggestions. We are pleased to hear that you recognize our theoretical contributions as *“clear and unified”*, our methodology as *“thoughtfully implemented”*, and the practical value demonstrated by our experiments. We address your concerns and respond to your questions below.
>
> > W1: The proposed components are adaptations of existing techniques. The theoretical unification largely reuses soft-RL derivations
>
> While the guidance approaches and RL theories we build upon originate from prior works [1, 2, 3], we argue that **the main contribution of our paper lies in unifying existing preference-alignment approaches in SBDD, analyzing their limitations, and developing a principled methodology informed by insights from both RL and the sampling dynamics of molecular generation**. Our aim is not to introduce a new diffusion modeling strategy or a novel RL theory, but rather to provide a coherent and theoretically grounded framework for SBDD, an aspect that has also been recognized by Reviewers ocYX, mVUX, and 3QPG.
>
> > W2 & Q3: Generalization to other protein-ligand benchmarks
>
> Thank you for the constructive suggestion. We conduct a zero-shot evaluation on the BindingMOAD [4] dataset. Following the split defined in *Schneuing et al. (2024)* [5], we ensure that the 130 test pockets are structurally distinct from the CrossDocked training pockets, and we sample 10 ligands for each pocket. We report the performance of our model alongside several baselines, and observe that **RL4SBDD-M achieves outstanding results on this novel benchmark**.
>
> | Model            | Med. Vina Score (↓) | Med. Vina Min (↓) | Med. Vina Dock (↓) | Avg. QED (↑) | Avg. SA (↑) |
> | ---------------- | ------------------- | ----------------- | ------------------ | ------------ | ----------- |
> | Ref              | -6.94               | -7.25             | -8.49              | 0.38         | 0.57        |
> | MolCRAFT [6]     | -7.02               | -7.43             | -7.85              | 0.53         | 0.65        |
> | MolJO [7]        | -7.47               | -7.87             | -8.65              | 0.58         | **0.69**    |
> | RL4SBDD-M (Ours) | **-8.89**           | **-9.19**         | **-9.27**          | **0.59**     | **0.69**    |
>
> > W3: The RL formulation doesn't yield deeper theoretical insights or new algorithmic properties
>
> We argue that the RL formulation provides valuable insights that guide our algorithmic design in the following ways:
>
> - **Gradient Guidance**. We offer a theoretical justification for TAGMol [6] and MolJO [7] by connecting their “energy models” with value estimation. As elaborated in Lines 184-190 and Figure 4(a), **the RL perspective reveals why these methods struggle during early sampling and motivates the design of our reward-conditioned policy model**.
>
> - **Reward Conditioning**. To the best of our knowledge, **we are the first to formally connect reward conditioning with the RL objective via Proposition 3**. The theoretical derivations clarify why reward conditioning is particularly effective during early sampling and provide a principled foundation for our mixed guidance strategy.
>
> > W4: The contribution of different components in the proposed methodology
>
> We further elaborate on the benefits of each design component presented in Table 2:
>
> - **Both gradient guidance and reward conditioning provide substantial improvements**. As shown in rows 9,10,11, gradient guidance improves the success rate by 21.7%, and reward-conditioning improves it by 16.6%.
>
> - **Combining gradient guidance and reward conditioning yields additional gains**, due to their complementary roles at different stages of sampling. By comparing row 1 with rows 3 and 4, row 5 with rows 6 and 7, and row 8 with rows 9, 10, we show that the combined policy outperforms the best individual policy by 3.9%, 7.9%, and 5.4% in success rate in each of the RL iterations.
>
> - **Mixed guidance improves the sampling efficiency while maintaining performance**, as evidenced by the comparison between rows 1 and 2.
>
> - **Iterative RL consistently improves the performance**. As shown in rows 1,5,8 and 4,7,10, the first and second RL rounds improve the reward-conditioned model by 6.2% and 6.8%, respectively, and improve the combined policy by 3.7% and 2.4%.

---

> ### Author Response · Authors · 2025-11-20
> **Response by Authors (Part 2)**
>
> > Q1: Trade-off between binding affinity and synthesizability
>
> The trade-off between the two objectives can be achieved through the following mechanisms:
>
> - **Controlling the conditioning inputs to $\Phi_P$**. As stated in Line 323 of the original submission, “we treat each metric as an independent dimension of the reward”, meaning that the normalized Vina Score and SA are used as two separate conditioning dimensions rather than being combined into a single reward value. This enables the model to regulate one property while preserving the other. We analyze different combinations of conditioning labels and observe that the reward-conditioned model can indeed optimize the two objectives independently, preventing one from dominating the other. Furthermore, when increasing either affinity or SA, we observe slight improvements in the other metric. This aligns with the findings of *Qiu et al. (2025)* [7], which report a positive correlation between binding affinity and synthesizability.
>
>   | Conditioning Label | Affinity=1 | Affinity=0.5 | Affinity=0 |
>   | ------------------ | ---------- | ------------ | ---------- |
>   | SA=1               | -8.38/0.83 | -7.59/0.84   | -6.31/0.83 |
>   | SA=0.5             | -8.21/0.71 | -7.51/0.68   | -6.30/0.67 |
>   | SA=0               | -8.03/0.65 | -7.47/0.62   | -6.06/0.58 |
>
>     (In each cell, we report the median Vina Score of 1000 generated molecules on the left of / and average SA on the right of /)
>
> - **Controlling the reward formulation for $\Phi_V$**. Instead of  Instead of assigning equal weights to both objectives in the reward function, we can define separate weights for each objective *i.e.*, $R=w_1\cdot r_{\text{affinity}}+w_2\cdot r_{\text{SA}}.$ We experiment with different combinations of weights and observe a clear trade-off between the two objectives: increasing the weight of one objective consistently leads to reduced performance on the other.
>
>   | Weight    | $w_1=1$    | $w_1=0.5$  | $w_1=0$    |
>   | --------- | ---------- | ---------- | ---------- |
>   | $w_2=1$   | -8.18/0.76 | -7.74/0.77 | -6.94/0.80 |
>   | $w_2=0.5$ | -8.91/0.74 | -7.94/0.76 | -7.02/0.79 |
>   | $w_2=0$   | -9.01/0.69 | -8.34/0.71 | -6.95/0.67 |
>
>    (In each cell, we report the median Vina Score of 1000 generated molecules on the left of / and average SA on the right of /)
>
> The above analysis underscores **the effectiveness of RL4SBDD-M in handling multi-objective optimization, ensuring that no single objective overwhelms the other**.
>
> > Q2: Empirical evidence that reward conditioning yields superior credit assignment
>
> Our original statement indicates that *“reward conditioning yields superior credit assignment in early sampling”*. To further validate this, we apply either gradient guidance or reward conditioning during sampling steps i * 20 ~ (i + 1) * 20 for i=0,1,2,3,4 while using the reference policy in all remaining steps. We report their advantages over the pure reference model (measured as the difference in final rewards), and we use a fixed noise sequence for each sampling trajectory to reduce stochasticity. The key observations are as follows:
>
> - **Reward conditioning outperforms gradient guidance in early sampling**, whereas gradient guidance becomes more effective in later stages.
>
> - **Both strategies provide consistent benefits throughout the sampling trajectory**, as indicated by their positive advantages over the reference model.
>
> | Model               | [0,20) | [20,40) | [40,60) | [60,80) | [80,100) |
> | ------------------- | ------ | ------- | ------- | ------- | -------- |
> | Gradient Guidance   | 0.0582 | 0.1029  | 0.1439  | 0.0912  | 0.0876   |
> | Reward Conditioning | 0.0608 | 0.2198  | 0.1398  | 0.0621  | 0.0473   |
>
> References
>
> [1] Tuomas Haarnoja, Haoran Tang, et al. "Reinforcement learning with deep energy-based policies." ICML 2017
>
> [2] Uehara M, Zhao Y, Wang C, et al. "Reward-guided controlled generation for inference-time alignment in diffusion models: Tutorial and review." arXiv 2025
>
> [3] Ho, Jonathan, and Tim Salimans. "Classifier-Free Diffusion Guidance." NeurIPS 2021 Workshop
>
> [4] Benson M L, Smith R D, Khazanov N A, et al. "Binding MOAD, a high-quality protein–ligand database." Nucleic acids research, 2007
>
> [5] Schneuing, Arne, et al. "Structure-based drug design with equivariant diffusion models." Science 2024
>
> [6] Qu Y, Qiu K, Song Y, et al. "MolCRAFT: structure-based drug design in continuous parameter space." ICML 2024
>
> [7] Qiu K, Song Y, Yu J, et al. "Empower Structure-Based Molecule Optimization with Gradient Guided Bayesian Flow Networks." ICML 2025
>
> [8] Dorna V, Subhalingam D, Kolluru K, et al. "Tagmol: Target-aware gradient-guided molecule generation." ICML 2024 Workshop

---

> > ### Comment · Reviewer_F2gE · 2025-11-26
> >
> > Thank you for the detailed responses and the additional experimental results. I have one more question.
> >
> > Recent findings such as PoseBuster [1,2] show that many generative SBDD models fail to produce physically valid ligand poses. Moreover, works like [3] build on these observations and explicitly evaluate physical validity as part of the optimization process.
> >
> > Could the authors clarify how physically valid the poses generated by RL4SBDD(-M) are? For example, have you assessed physical validity using criteria similar to those used in PoseBuster, or evaluated whether generated molecules exhibit issues such as steric clashes, unrealistic conformations, or violations of geometric constraints? Any quantitative or qualitative analysis would greatly strengthen the empirical claims.
> >
> > [1] Harris, Charles, et al. "Benchmarking generated poses: How rational is structure-based drug design with generative models?." arXiv:2308.07413, 2023.
> > [2] Buttenschoen, Martin, Garrett M. Morris, and Charlotte M. Deane. "PoseBusters: AI-based docking methods fail to generate physically valid poses or generalise to novel sequences." Chemical Science, 2024.
> > [3] Lee, Seungbeom, et al. "Enhancing Ligand Validity and Affinity in Structure-Based Drug Design with Multi-Reward Optimization." ICML 2025.

---

> > > ### Author Response · Authors · 2025-11-27
> > > **Response by Authors**
> > >
> > > We thank the reviewer for the insightful suggestion regarding the evaluation of conformational quality. We would like to confirm that the strain energy of the generated poses has been reported using PoseCheck [1] in our initial submission (Table 4), and the number of steric clashes (*i.e.*, the number of ligand atoms exhibiting a intermolecular distance of $\leq 0.4\text{Å}$ with respect to the protein atoms) is reported in Table 1. Moreover, we report the percentage of the generated poses passing all 20 PoseBusters validity tests [2] in the following table:
> > >
> > > | Model                   | Med. Strain Energy (↓) | Clashes (↓) | PB-Valid (↑) |
> > > | ----------------------- | ---------------------- | ----------- | ------------ |
> > > | Ref                     | **114**                | 5.46        | **95.0%**    |
> > > | AR                      | 608                    | **4.18**    | 59.0%        |
> > > | Pocket2Mol              | 186                    | 6.22        | 72.3%        |
> > > | TargetDiff              | 1208                   | 10.67       | 50.5%        |
> > > | DecompDiff              | 983                    | 14.23       | 71.7%        |
> > > | MolCRAFT                | 196                    | 6.91        | 84.6%        |
> > > | DecompOpt               | 861                    | 16.60       | 48.0%        |
> > > | MolJO                   | 163                    | 6.72        | 87.1%        |
> > > | *Lee et al. (2025)* [3] | -                      | 6.69        | 85.0%        |
> > > | RL4SBDD-M (Ours)        | 125                    | 5.89        | 85.1%        |
> > >
> > > The above results show that **RL4SBDD-M achieves outstanding performance in generating physically plausible molecules and binding poses**. We attribute this success largely to our carefully constructed RL dataset, where PB-invalid poses are filtered. The PB-validity results and comparisons with *Lee et al. (2025)* [3] have been incorporated in Table 6 and Table 1 within the revised version of our paper.
> > >
> > > We sincerely hope that our responses and additional results have addressed all your concerns satisfactorily, and we kindly ask you to re-evaluate our submission in light of these clarifications. We remain enthusiastic about our work and are willing to provide any further information you may require.
> > >
> > > References
> > >
> > > [1] Harris C, Didi K, Jamasb A R, et al. "Benchmarking generated poses: How rational is structure-based drug design with generative models?" arXiv 2023
> > >
> > > [2] Buttenschoen M, Morris G M, Deane C M. "PoseBusters: AI-based docking methods fail to generate physically valid poses or generalise to novel sequences" Chemical Science 2024
> > >
> > > [3] Lee S, Jo M, Ok J, et al. "Enhancing Ligand Validity and Affinity in Structure-Based Drug Design with Multi-Reward Optimization." ICML 2025

---

> > > > ### Comment · Reviewer_F2gE · 2025-11-27
> > > >
> > > > Thank you for the comprehensive response. I appreciate the additional analyses on physical validity. I also acknowledge the substantial revisions prompted by the other reviewers’ comments. Given the improved manuscript, I am updating my evaluation accordingly and raising my overall score.

---

> > > > > ### Author Response · Authors · 2025-11-27
> > > > > **Thank you**
> > > > >
> > > > > Thanks again for your favorable comments on our responses and revision. We are glad to have addressed all your concerns. We are willing to provide additional information if you have any further questions.

---

### Official Review · Reviewer_ocYX · 2025-11-01

**Soundness:** 3
**Presentation:** 3
**Contribution:** 3
**Rating:** 6
**Confidence:** 4

**Summary:**

This paper proposes RL4SBDD, a unified reinforcement-learning framework for preference alignment in structure-based drug design, which bridges gradient-based and gradient-free methods under a single KL-regularized objective, and emphasize the limitation including biased value estimation, insufficient use of training data, and off-policy distribution shift of current preference alignment methods. The authors introduce a reward-conditioned policy model $\Phi_{P}$ with value model $\Phi_{V}$, and a mixed-guidance sampling strategy that combines gradient-based and gradient-free methods for better performance. The method achieves state-of-the-art results on CrossDocked2020 with improved efficiency.

**Strengths:**

- Provides a clear theoretical unification of preference alignment methods under a principled RL objective.
- Offers solid theoretical derivations and clean algorithmic formulation
- Introduces a reward-conditioned generative policy, novel in SBDD and well-grounded
- Comprehensive experiments with consistent performance improvements and ablations

**Weaknesses:**

Please see questions below.

**Questions:**

1. Lines 205–206 and 209–210 state that current DPO data selection leads to suboptimal results. However, this appears not to be an intrinsic limitation of DPO itself, but rather a data efficiency issue, since DPO must iterate over all possible preference pairs to fully utilize training data.
2. The rationale for the noise reduction design requires further justification. When noise level decreases from 1 to 0.2, the diversity drops sharply to 0.53, while the success rate increases only marginally. This suggests that the current setting may lead to over-regularization.
3. It remains unclear to me whether the reward-conditioned policy model alone suffices for generation. What is the performance when sampling directly from $\Phi_{P}$ with high-reward conditioning only? Why is additional mixing with the reference model $\Phi_{ref}$ necessary?
4. Is there any quantitative analysis of the reward model $\Phi_{V}$? Its predictive accuracy and stability would help clarify how much it contributes to the overall performance.
5. The authors only use the normalized Vina Score and SA score as reward signals, neglecting QED. Given that the improvement in QED is relatively small, it would be helpful to explain this choice and whether the framework generalizes to other reward functions.

---

> ### Author Response · Authors · 2025-11-20
> **Response by Authors (Part 1)**
>
> We thank the reviewer for thoroughly examining our work and providing constructive feedback. We are delighted to hear that our framework offers *“a clear theoretical unification, derivation, and clean algorithmic formulation”*, and that the novelty, solid methodological grounding, and comprehensive experiments are appreciated. We hope that our responses and additional analyses address your concerns satisfactorily.
>
> > Q1: Critiques about the limitations of DPO-based approaches in SBDD
>
> We apologize for the confusing statements regarding DPO. Since the scope of our framework is restricted to SBDD, our intention was to **critique the existing implementations of DPO within the SBDD context** (as referenced in Lines 71 and 205 in our original submission), **rather than the DPO algorithm itself**. Although training on all preference pairs may alleviate data-efficiency concerns, experiments indicate that this approach still leads to suboptimal performance. Furthermore, we acknowledge that recent works [1, 2] have found that training on all-pair DPO, especially those with similar rewards, can lead to likelihood displacement. We have incorporated these discussions in Lines 77-78 and 232-234 in our revised manuscript.
>
> (The table below is borrowed from Table 6 in AliDiff [3])
>
> | Choice     | Med. Vina Score (↓) | Med. Vina Min (↓) | Avg. QED (↑) | Avg. SA (↑) |
> | ---------- | ------------------- | ----------------- | ------------ | ----------- |
> | best-worst | -7.95               | -8.17             | 0.50         | 0.56        |
> | all-pairs  | -7.82               | -7.96             | 0.50         | 0.56        |
>
> > Q2: Further justification for noise reduction
>
> Thank you for your insightful comments. To further demonstrate the benefits of noise reduction, we conduct experiments using three random seeds for both $\epsilon=0.2$ and $\epsilon=1$ and generating 10 molecules per pocket for each run. We report both the mean and standard deviation across runs, and the results clearly indicate that **the improvements in success rate are statistically significant**.
>
> Furthermore, the intention of the proposed noise reduction strategy is to **facilitate the trade-off between optimality and diversity**. Hence, $\epsilon$ is not a fixed choice and can be flexibly adjusted in different practical scenarios. We have adjusted our statements in Lines 357-359 and 486 in our revised version.
>
> | $\epsilon$ | Avg. Diversity    | Avg. Success Rate (↑) |
> | ---------- | ------------ | ---------------- |
> | 0.2        | ~0.68$\pm$0.1~  0.53$\pm$0.0 | 58.0%$\pm$0.9%   |
> | 1.0        | ~0.53$\pm$0.0~  0.68$\pm$0.1 | 55.5%$\pm$1.4%  |
>
> > Q3: Performance of the conditional policy model and additional mixing with the reference model
>
> The motivation for incorporating additional mixing with the reference model originates from the concept of the “unconditional model” in classifier-free guidance [4]. From an RL perspective, **the reference model serves as a baseline policy, and the deviation between it and the reward-conditioned model helps identify a more advantageous region of the action space via extrapolation**. We have provided explanations for this design in Lines 266-269 in our revised paper.
>
> In our original submission (Table 6), we also evaluated sampling using only the reward-conditioned policy (i.e., $w$=1). We observe that **introducing mild mixing with the reference model ($w$=1.2) leads to modest performance gains**.
>
> | Setting             | Med. Vina Score (↓) | Med. Vina Min (↓) | Avg. QED (↑) | Avg. SA (↑) | Avg. Success Rate (↑) |
> | ------------------- | ------------------- | ----------------- | ------------ | ----------- | --------------------- |
> | w/o mixing ($w=1$)  | -8.60               | -8.80             | 0.52         | 0.85        | 57.2%                 |
> | w/ mixing ($w=1.2$) | -8.70               | -8.87             | 0.54         | 0.85        | 58.1%                 |

---

> ### Author Response · Authors · 2025-11-20
> **Response by Authors (Part 2)**
>
> > Q4: Qualitative analysis of the reward model
>
> Thank you for the insightful suggestion. We evaluate the accuracy and stability of the value model on the validation pockets of CrossDocked2020 and the ligands obtained after the second round of iterative RL. At each of 10 steps, we add noise to the original ligand and use the value model $\Phi_V$ to predict its reward. We then report the mean and standard deviation of the mean absolute error (MAE) with respect to the normalized ground-truth properties.
>
> | Step             | 10            | 20            | 30            | 40            | 50            |
> | ---------------- | ------------- | ------------- | ------------- | ------------- | ------------- |
> | MAE (Vina Score) | 0.0979±0.0167 | 0.0642±0.0072 | 0.0556±0.0069 | 0.0463±0.0056 | 0.0401±0.0048 |
> | MAE (SA)         | 0.1163±0.0104 | 0.0908±0.0088 | 0.0665±0.0060 | 0.0509±0.0043 | 0.0409±0.0044 |
>
> | Step             | 60            | 70            | 80            | 90            | 100           |
> | ---------------- | ------------- | ------------- | ------------- | ------------- | ------------- |
> | MAE (Vina Score) | 0.0325±0.0045 | 0.0276±0.0034 | 0.0252±0.0034 | 0.0231±0.0032 | 0.0224±0.0035 |
> | MAE (SA)         | 0.0338±0.0038 | 0.0285±0.0033 | 0.0263±0.0034 | 0.0253±0.0030 | 0.0245±0.0029 |
>
> We observe that the predictive accuracy of $\Phi_V$ improves along the sampling trajectory as the input molecule becomes less noisy. In the 40-100 steps, **the MAE is close to or below 0.05**, underscoring the reliability of the model in providing gradient guidance. The irreducible error may also explain why expanding the training set in the second RL iteration yields limited improvements.
>
> > Q5: Choice of the reward function and generalization to other reward functions
>
> We follow the same evaluation setting as MolJO [5] and use Vina Score + SA to ensure a fair comparison. Our additional justification is as follows:
>
> - **Using Vina Score + SA yields the most balanced optimization results**. Under this setting, the average QED is 0.56 for MolJO and 0.54 for RL4SBDD-M, both of which are considered favorable in practical scenarios (QED > 0.5) [6]. Moreover, our experiments show that optimizing Vina Score + SA also brings slight improvements in QED.
>
> - **Optimizing QED (enforcing drug-likeliness) restricts the exploration space and adversely affects binding affinity**. For substantiation, we train $\Phi_P$ and $\Phi_V$ on the enriched dataset from the second RL iteration to jointly optimize Vina Score, QED, and SA. The results show a notable increase in QED at the expense of declines in binding affinities.
>
> To further demonstrate the effectiveness of our framework under alternative reward functions, we conduct an additional experiment on jointly optimizing the Vina Score, SA, and calculated water-octanol partition coefficient (cLogP) to regularize lipophilicity and the number of aromatic rings. Specifically, we calculate the LogP of the molecules within the dataset at the second round RL with RDKit [7], clip the value within the range [-3, 7], normalize it into [0,1], and consider cLogP=1 as the most desirable value [8]. For gradient guidance, the reward is calculated as $|cLogP_{norm}-0.4|$. For reward conditioning, we set the conditioning label $cLogP_{norm}$ as 0.4. Experiments show that RL4SBDD-M achieves a balanced trade-off between binding affinity, synthesizability, and desired cLogP.
>
> | Model            | Alignment Objective     | Med. Vina Score (↓) | Avg. QED (↑) | Avg. SA (↑) | Avg. cLogP |
> | ---------------- | ----------------------- | ------------------- | ------------ | ----------- | ---------- |
> | Ref              | -                       | -6.46               | 0.48         | 0.73        | 0.89       |
> | MolJO            | Vina Score + SA         | -8.02               | 0.56         | 0.78        | 4.20       |
> | RL4SBDD-M (Ours) | Vina Score + SA         | **-8.70**           | 0.54         | **0.85**    | 5.48       |
> | RL4SBDD-M (Ours) | Vina Score + SA + QED   | -8.32               | 0.60         | 0.83        | 4.86       |
> | RL4SBDD-M (Ours) | Vina Score + SA + cLogP | -7.88               | 0.58         | 0.77        | 3.08       |
>
> The above results show that **RL4SBDD-M is generalizable to a broader range of reward functions and balances between multiple objectives**. We have incorporated the above analysis on more reward combinations in Section 5.5 in our revised manuscript. As stated in our initial submission (Section 6), we leave investigation of more reward functions such as ADME properties [9] for future work.

---

> ### Author Response · Authors · 2025-11-20
> **Response by Authors (Part 3)**
>
> References
>
> [1] Razin N, Malladi S, Bhaskar A, et al. "Unintentional unalignment: Likelihood displacement in direct preference optimization." ICLR 2025
>
> [2] Chen P, Chen X, Yin W, et al. "ComPO: Preference alignment via comparison oracles." NeurIPS 2025
>
> [3] Gu, Siyi, et al. "Aligning target-aware molecule diffusion models with exact energy optimization." NeurIPS 2024
>
> [4] Ho, Jonathan, and Tim Salimans. "Classifier-Free Diffusion Guidance." NeurIPS 2021 Workshop
>
> [5] Qiu K, Song Y, Yu J, et al. "Empower Structure-Based Molecule Optimization with Gradient Guided Bayesian Flow Networks." ICML 2025
>
> [6] RGDscience, "Molecular Descriptors & Ligand Efficiency Metrics." https://www.rgdscience.com/index.php/molecular-descriptors-ligand-efficiency-metrics/
>
> [7] Bento A P, Hersey A, Félix E, et al. "An open source chemical structure curation pipeline using RDKit" Journal of Cheminformatics, 2020
>
> [8] Arnott J A, Planey S L. "The influence of lipophilicity in drug discovery and design" Expert opinion on drug discovery, 2012
>
> [9] Hodgson J. "ADMET—turning chemicals into drugs." Nature biotechnology, 2001

---

> ### Comment · Reviewer_ocYX · 2025-11-28
>
> Thanks to the authors for providing the additional experiments and clarifications. They adequately address my earlier questions. In the table for Q2, it appears that the diversity values for $\epsilon = 0.2$ and $\epsilon = 1.0$ may have been reversed, please double-check this. I will maintain my positive rating. If the paper is accepted, please include these discussions into the final version.

---

> > ### Author Response · Authors · 2025-11-28
> > **Thank you**
> >
> > We thank the reviewer for the favorable comments and are glad to hear that we have addressed all your concerns. The result for Q2 in the table was indeed a typographical error, which has now been corrected. We are prepared to provide any additional information or clarification if any further questions or points of discussion arise.

---

### Author Response · Authors · 2025-11-20
**Paper Revision Overview**

We sincerely appreciate the reviewer's valuable comments and constructive feedback, which helps us to enhance the quality of our work. We have conducted a thorough revision of the manuscript, making every effort to address the stated concerns. For the reviewer's convenience, key modifications are highlighted in blue throughout the paper. We hope that the reviewers will find our revisions and responses satisfactory and reconsider their evaluation of our paper. The major changes are summarized below; please refer to the attached rebuttal for point-to-point discussions.

### Major Revisions

- Reviewers mVUX and 3QpG raised concerns on the possibility that over-optimizing the reward (*i.e.*, Vina Scores and SA scores) in our main experiments may inadvertently introduce undesirable molecular properties. While acknowledging that this issue stems primarily from the established, yet imperfect, reward design adopted from prior literature, we have implemented and validated two effective strategies to mitigate this phenomenon: (1) integrating additional objectives into the reward function, and (2) adjusting guidance weights to encourage stronger adherence to the reference policy. A thorough discussion and comprehensive analysis of these approaches are presented in Section 5.5 and Appendix E.6 of the revised manuscript, as well as in our direct responses to Reviewers mVUX and 3QpG.
- Reviewer 3QpG pointed out the ambiguity between *"desired molecules"* and *"molecules with high binding and SA scores"*. We have adjusted potentially misleading statements and explicitly delineated the distinctions between RL4SBDD-M's adaptability to *"any verifiable rewards"* and *"the specific reward employed in the main experiments"*.

### Minor Revisions

- We have included a brief discussion on the potential application of all-pair DPO training, as suggested by reviewers ocYX and mVUX.
- In response to reviewer F2gE, a remark regarding the assumption of a Dirac prior has been added.
- We provide additional technical justifications for our choice of a gradient-free guidance design, as suggested by Reviewer ocYX.
- Based on the insights from reviewers ocYX, mVUX, and 3QpG, we have refined the statements regarding the noise reduction strategy and have clarified that the application of RL does not inherently lead to a reduction in molecular diversity.
- We have conducted supplementary experiments and baseline comparisons to address the specific recommendations put forth by Reviewers F2gE and 3QpG.

---

### Meta-Review · Area_Chair_yudz · 2026-01-01

**Summary:**

This paper proposes RL4SBDD, an RL-based “unification” of preference-alignment methods for structure-based drug design, and introduces RL4SBDD-M combining reward-conditioned policies, iterative RL, and mixed-guidance/noise-reduction sampling to improve docking/SA metrics on CrossDocked2020. Despite strong empirical gains on the chosen benchmark, the submission’s core contributions are largely repackaging known soft-RL and diffusion guidance ideas, and the strongest results hinge on surrogate objectives with demonstrated reward hacking and nontrivial tuning, limiting practical and scientific value. The submission went through major revisions, adding substantial experiments and ablations that were missing from the original submission. Given the length of discussions and the significant amount of required additional experiments, the paper is not ready for acceptance at ICLR.

Pros: clear RL framing; strong engineering/ablations; competitive CrossDocked2020 numbers; added generalization/validity analyses in rebuttal (BindingMOAD, PoseBusters/PoseCheck).

Cons: limited novelty; primary optimization target (Vina+SA) is weakly conflicting and encourages “hacks”; practical utility and fairness of test-time scaling (oracle screening, priors) remain questionable; sensitivity/tuning + limited multi-seed evidence reduces confidence.

**Reviewer Concerns:**

Partially addressed: ocYX’s technical clarifications (DPO critique scope, mixing/noise rationale, value-model diagnostics, reward-choice justification) and F2gE’s requests on additional benchmarks and physical validity (PoseBusters/PoseCheck + clashes/strain) with score increase; some discussion added on assumptions (Dirac prior) and diversity/noise trade-off.

Outstanding: 3QpG’s core domain objections remain—reward hacking is empirically confirmed (interaction/ring/cLogP shifts).

**Reviewer Scores:**

* ocYX: explicitly satisfied; likely unchanged (stay marginally positive) as they stated themself.
* F2gE: already mentioned that they would raise their score from 4 to 6 after added physical-validity evaluation; likely stays high.
* 3QpG raised fundamental concerns regarding the missing baslines, low diversity and reward hacking and it's unlikely that the rebuttal would substantially change their rating.
* mVUX: may remain positive, but borderline;

---

### Decision · Program_Chairs · 2026-01-26

Reject